# The Antarctic contribution to 21st century sea-level rise predicted by the UK Earth System Model with an interactive ice sheet

Antony Siahaan[1], Robin S. Smith[2], Paul R. Holland[1], Adrian Jenkins[1,3], Jonathan M. Gregory[2,4],Victoria Lee[5], Pierre Mathiot[4,6], Antony J. Payne[5], Jeff K. Ridley[4], and Colin G. Jones[7]

[1]British Antarctic Survey, Cambridge, UK
[2]NCAS/Department of Meteorology, University of Reading, Reading, UK
[3]Now at University of Northumbria, Northumbria, UK
[4]Met Office Hadley Centre, Exeter, UK
[5]CPOM/Bristol Glaciology Centre, University of Bristol, Bristol, UK
[6]Now at CNRS/Université Grenoble Alpes, Grenoble, France
[7]NCAS/University of Leeds, Leeds, UK

*Correspondence to*: Antony Siahaan (antsia@bas.ac.uk)

**Abstract.**

The Antarctic Ice Sheet will play a crucial role in the evolution of global mean sea-level as the climate warms. An interactively coupled climate and ice sheet model is needed to understand the impacts of ice—climate feedbacks during this evolution. Here we use a two-way coupling between the U.K. Earth System Model and the BISICLES dynamic ice sheet model to investigate Antarctic ice—climate interactions under two climate change scenarios. We perform ensembles of SSP1-1.9 and SSP5-8.5 scenario simulations to 2100, which we believe are the first such simulations with a climate model that include two-way coupling of atmosphere and ocean models to dynamic models of the Greenland and Antarctic ice sheets. We focus our analysis on the latter. In SSP1-1.9 simulations, ice shelf basal melting and grounded ice mass loss from the Antarctic ice sheet are generally lower than present rates during the entire simulation period. In contrast, the responses to SSP5-8.5 forcing are strong. By the end of the 21st century, these simulations feature order-of-magnitude increases in basal melting of the Ross and Filchner-Ronne ice shelves, caused by intrusions of warm ocean water masses. Due to the slow response of ice sheet drawdown, this strong melting does not cause a substantial increase in ice discharge during the simulations. The surface mass balance in SSP5-8.5 simulations shows a pattern of strong decrease on ice shelves, caused by increased melting, and strong increase on grounded ice, caused by increased snowfall. Despite strong surface and basal melting of the ice shelves, increased snowfall dominates the mass budget of the grounded ice, leading to an ensemble-mean Antarctic contribution to global mean sea level of a fall of 22 mm by 2100 in the SSP5-8.5 scenario. We hypothesise that this signal would revert to sea-level rise on longer timescales, caused by the ice sheet dynamic response to ice shelf thinning. These results demonstrate the need for fully coupled ice—climate models in reducing the substantial uncertainty in sea-level rise from the Antarctic Ice Sheet.

# 1 Introduction

The Antarctic Ice Sheet (AIS) is a critically important component of the Earth system (Fyke et al., 2018; Noble et al., 2020). The total freshwater stored in the AIS amounts to ~58 m global mean sea level (GMSL) equivalent (Fretwell et al., 2013; Vaughan et al., 2013). Its close coupling with the surrounding Southern Ocean (Holland et al., 2020; Leutert et al., 2020) gives it a vital role in climate processes with global impacts such as sea ice growth and melting (Nadeau et al., 2019), bottom water formation (Purkey and Johnson, 2010), and carbon (Sabine et al., 2004) and heat (Marshall and Speer, 2012) uptake. Direct or indirect interactions between the AIS and other Earth system components can therefore strongly affect future sea level rise and climate change in response to anthropogenic greenhouse gas forcing.

About 75% of Antarctica's coastline is fringed by floating ice shelves (Rignot et al., 2013). While these ice shelves do not make a substantial direct contribution to sea level rise, the stability of the AIS depends on their buttressing (Dupont and Alley, 2005), since thinning of ice shelves can lead to faster ice flow and loss of grounded ice. Increased monitoring of the AIS in recent decades indicates that the ice sheet has been thinning in various basins and increasingly losing mass from grounded areas (Wingham et al., 1998; Wingham et al., 2006; Pritchard et al., 2009; Rignot et al., 2019; Shepherd et al., 2019) which contribute to sea level rise. The observed thinning of the grounded AIS, especially in the western sector, is notably associated with strong oceanic melting under ice shelves (Rignot and Jacobs, 2002; Shepherd et al., 2004; Payne et al., 2004; Jacobs et al., 2011; Pritchard et al., 2012) which reduces the ice sheet buttressing force and thus increases the ice discharge across grounding lines (Schoof, 2007; Fürst et al., 2016; Gudmundsson et al., 2019).

For this reason, it would be concerning if the projected rise of global surface temperature associated with anthropogenic emissions of greenhouse gases (GHGs) were to be replicated in Antarctic ocean properties. It has been hypothesised that anthropogenic changes in the winds have increased the transport of warm ocean waters (Spence et al., 2017) towards the ice shelves in the Amundsen Sea, increasing melting (Holland et al., 2019). However, ocean heat transport towards the Antarctic coastline is controlled by many factors, such as density structure of the ocean, wind patterns, bathymetry, local ocean circulation, and sea ice processes (Thompson et al., 2018). As parts of the Earth system, these factors will also interact with each other and may respond in complex ways to future change in regional climate. As a result, making projections of the response of Antarctic ice shelf melt to future climate warming is very challenging.

One of the objectives of the Coupled Model Intercomparison Project Phase 6 (CMIP6; Eyring et al., 2016) is the understanding of future climate change through multi-model climate projections based on alternative scenarios of future emissions and land use changes, coordinated via the Scenario Model Intercomparison Project Phase 6 (ScenarioMIP6; O'Neill et al., 2016). In addition to a core of coupled atmospheric and ocean-sea models, many CMIP6 models include components such as atmospheric chemistry, and land and oceanic biogeochemistry models which can be also used to simulate the carbon cycle. Yet none of the current CMIP6 models has included a dynamic AIS model, which makes them unable to simulate the impacts that the ice sheet evolution has on global sea-level rise and the other climate components.

The main CMIP6 efforts in assessing the ice sheet contribution to future sea level rise come through the Ice Sheet Model Intercomparison Project Phase 6 (ISMIP6; Seroussi et al., 2020). However, the forcings used in ISMIP6 are derived from atmospheric and oceanic outputs of CMIP5 and CMIP6 climate models, which do not include feedbacks with the AIS. In addition, due to the absence of ice shelf ocean cavities, none of those climate models represent the ocean physics under ice shelves, which is a key regulator of ice shelf melting (Jacobs et al., 1992) and therefore the stability of the ice sheet (Pritchard et al., 2012) and the oceanography of the nearby continental shelf and deep Southern Ocean (Foldvik et al., 2004).

Other future projections which include the AIS or ice shelf cavities vary between regional or global ocean-sea ice models (Hellmer et al., 2012; Timmermann and Hellmer, 2013; Naughten et al., 2018), stand-alone Antarctic ice sheet models (Sun et al., 2020), regional coupled ocean-ice sheet models (Timmermann and Goeller, 2017; Naughten et al., 2021) and global low resolution coupled climate-ice sheet model without ice shelves (Vizcaino et al., 2010). While these setups offer more flexibility in the model resolution due to their reduced computational resource requirement, they suffer from limitations associated with the absence of feedbacks between climate components or through open boundary conditions.

A CMIP6-class global Earth System model that now includes a two-way interaction between UKESM1.0 (Sellar et al.,2019) and the BISICLES ice sheet model (Cornford et al.,2013) has been introduced in a previous work (Smith et al., 2021). This model, called UKESM1.0-ice, includes evolving ice shelf cavities in which oceanic basal melting is explicitly simulated. Here we present the first simulations of this ice—climate coupled model for the AIS under the anthropogenic forcing scenarios (Shared Socioeconomic Pathway) SSP1-1.9 and SSP5-8.5. We believe these are the first simulations using an atmosphere-ocean general circulation model (AOGCM) with two-way coupling between both atmosphere and ocean components to dynamic models of the Greenland and Antarctic ice sheets.

In this paper we examine the most novel aspect of UKESM1.0-ice by investigating the evolution of the Antarctic ice sheet in these simulations. We will not discuss our simulation of the Greenland ice sheet here, for which indicative results are discussed by Smith et al. (2021). We focus primarily on the evolution of Antarctic ice shelf basal melting, but also consider the surface mass balance (SMB) and the dynamic response of the ice sheet, and note the implications for the AIS contribution to global mean sea-level rise over the 21st century. Our main intentions are to scientifically analyse climatic signals around the AIS over the 21st century, especially where there are clear differences between the impacts of the high and low radiative forcing scenario. In section 2, we describe the coupled model and how the simulations are set up and initialised. The results of the future scenario runs are analysed in Section 3, which covers the major features of the 21st century simulations of the AIS and its surface and basal mass balance. In Section 4 some general issues in each of the two scenarios are discussed while Section 5 concludes the main results from this work.

## 2 Methods and Experimental Setups

This section first outlines the main components of UKESM1.0-ice and how the coupling between its climate and ice sheet model component is implemented. Then we detail how the simulations are set up and initialised.

## 2.1 Model Description and Coupling

UKESM1.0-ice (UK Earth System Model-Ice Sheet) comprises a global Earth system model and an ice sheet model which are bidirectionally coupled (Smith et al., 2021). This system uses a modified version of UKESM1.0 (Sellar et al., 2019) coupled to the adaptive-mesh BISICLES ice sheet model (Cornford et al., 2013). We emphasise that the suffix 'ice' in UKESM1.0-ice is added to refer to the coupled model with ice shelf cavities and an active ice sheet, whereas UKESM1.0 (without suffix 'ice') is the coupled model with a static ice sheet and no ice shelf cavity.

UKESM1.0 is built upon various component models. Its physical core is the atmosphere-ocean climate model HadGEM3-GC3.1 (Kuhlbrodt et al., 2018; Williams et al., 2018) with some minor adjustments (Sellar et al., 2019). The additional components which are interactively coupled to HadGEM3-GC3.1 in UKESM1.0 are terrestrial carbon and nitrogen cycles, which include dynamic vegetation and representation of agricultural land use change (Harper et al., 2018), ocean biogeochemistry (Yool et al., 2013), and a unified troposphere-stratosphere chemistry model, tightly coupled to a multi-species modal aerosol scheme (Archibald et al., 2020).

The UKESM1.0 configuration of HadGEM3-GC3.1 has a global resolution of N96 (~135 km) and 85 vertical levels in the atmosphere and ORCA1 (1° longitude) and 75 vertical levels in the ocean. Its component models are the Unified Model (UM) for the atmosphere (Brown et al., 2012), Nucleus for European Modelling of the Ocean (NEMO) for the ocean (Madec and the NEMO Team, 2016), the Community Ice CodE (CICE) for the sea ice (Hunke et al., 2015), and the Joint UK Land Environment Simulator (JULES) for the land surface (Best et al., 2011). The interactions between these components are carried out through the OASIS3-MCT coupler (Craig et al., 2017).

BISICLES (Cornford et al., 2013) is an ice sheet model that implements a vertically-integrated stress balance approximation built on the adaptive-mesh Chombo framework (Adams et al., 2019). The time-evolving adaptive horizontal meshing enables BISICLES to resolve dynamically important ice sheet regions at fine resolution while using coarser resolution in the slower-moving main body of the ice sheet. The configuration in UKESM1.0-ice uses a shallow-shelf/shelfy-stream approximation (SSA) with a modified L1L2 approximation that includes vertical shear in the effective viscosity (Schoof and Hindmarsh, 2010), but neglects the L1L2 approximation in the mass flux (Cornford et al 2020). At the base, we use the basal friction physics as in Tsai et al. (2015). The ice thickness is divided vertically into 10 sigma layers which increase in resolution from 16% of thickness near the surface to 3% of thickness near the ice sheet base (Cornford et al., 2015). For reasons of computational affordability, we set our coarsest BISICLES mesh to have a grid box length of 8 km which is allowed to refine to 2km where required to resolve the flow better. Details of refinement criteria and levels of refinement are described in earlier BISICLES works (Cornford et al., 2016; Cornford et al., 2013). A steady state 3-D temperature field from a higher-order model (Pattyn, 2010) is used as the internal ice temperature of the ice sheet. The fields of effective drag coefficients and effective viscosities employed in the model are held constant over the course of simulations. Values for these coefficients are taken from Cornford et al. (2016) which used the inversion procedure in Cornford et al. (2015) to minimise the discrepancy between modelled and observed ice speeds.

Some modifications are made to UKESM1.0 to enable its coupling with BISICLES (Smith et al., 2021). The most notable modification in the NEMO ocean is the activation of ocean circulation in the cavities (with horizontal resolution about 17-22

km) under ice shelves, whose draft can evolve. This allows NEMO to simulate thermodynamic ice shelf–ocean interaction (Mathiot et al., 2017), explicitly calculating ice shelf basal melting and freezing using a fixed-thickness boundary layer under the ice shelf (Losch, 2008) by means of the 3-equation method (Holland and Jenkins, 1999) :

$$\rho C_p \gamma_T (T_w - T_b) = -L_f q - \rho_i C_{p,i} \kappa \frac{T_s - T_b}{h_{isf}} \tag{1}$$

$$\rho \gamma_S (S_w - S_b) = (S_i - S_b) q \tag{2}$$

$$T_b = \lambda_1 S_b + \lambda_2 + \lambda_3 z_{isf} \tag{3}$$

where $T_w$ and $S_w$ are the ocean cavity top boundary layer water temperature and salinity, respectively, $T_b$ and $S_b$ the temperature and salinity at the ocean-ice shelf interface, respectively, $\gamma_T$ and $\gamma_S$ the exchange coefficients for temperature and salt, respectively, $S_i$ the ice salinity, $z_{isf}$ and $h_{isf}$ the ice shelf draft and thickness, respectively, $\rho$ and $\rho_i$ the density of seawater and ice shelf, respectively, $C_p$ and $C_{p,i}$ the specific heat capacity of water and ice, respectively, $\kappa$ and $L_f$ the thermal diffusivity and

specific latent heat of ice, respectively, and $T_s$ the ice/air interface temperature (assumed to be $-20\ ^0C$) whereas $\lambda_1$, $\lambda_2$, and $\lambda_3$ are all constant.

In (1) and (2), the top boundary layer properties $T_w$ and $S_w$ are averaged over 20m thickness below the ice shelf base, whereas the exchange coefficients are velocity dependent and written as :

$$\gamma_T = \sqrt{C_d} u_w \Gamma_T \tag{4}$$

$$\gamma_S = \sqrt{C_d} u_w \Gamma_S \tag{5}$$

where $u_w$ is the ocean velocity in the top boundary layer, $C_d$ the drag coefficient and both $\Gamma_T$ and $\Gamma_S$ are constant. We use the same values for all the parameters as those in Mathiot et al. (2017). The resulting melt rate $q$ on the ocean grid is then bilinearly interpolated onto the (finer) ice sheet model grid and used by BISICLES as its basal mass balance forcing.

Ocean properties in the cavity may be adapted (Smith et al., 2021) after coupling with BISICLES in response to changes in

the ice shelf thickness. The rules for the affected variables in the cavity are:

- Tracer and velocity values in cells that are ocean before and after the change in ice shelf thickness remain unchanged even if their cell thickness has changed due to the new vertical position of the ice shelf draft.

- If a new ocean tracer cell is created to replace an ice shelf cell, tracer values are obtained by extrapolating from neighbouring ocean tracer cells. New velocity cells are initialised at rest.

- If an entire new vertical column of water is created following grounding line retreat, then sea surface height is obtained by extrapolating from neighbouring columns. Tracer and velocity values follow the rules for newly created cells.

- If an entire cell/column is closed and replaced by ice shelf, the cell/column is masked and its properties are lost.

Additionally, artificial mass fluxes are applied to the first timestep after coupling, to conserve the depth-integrated horizontal divergence of a water column under a changed ice shelf draft. This maintains the stability of the ocean computation when the

ice shelf evolves and is particularly useful for situations when grounding lines advance. The column-integrated divergence is

conserved by first applying artificial mass fluxes to conserve the horizontal divergence of each individual cell in the water column. Any cell which has been closed and replaced by ice shelf gives its artificial mass flux to the nearest unmasked cell beneath it. The heat and salt flux associated with this added mass flux is computed using the local temperature and salinity as an advective flux.

In this coupling framework, calved icebergs drift and melt in the ocean. Nevertheless, the ice shelf calving front position in NEMO is fixed because the land-sea mask cannot be easily changed in UKESM1.0. For this reason, a fixed-front calving condition is applied along the initial boundary of the Antarctic ice sheet in BISICLES. This is maintained throughout the simulation by imposing an artificial minimum ice thickness of 10 m. The flux of ice through this calving front is passed to the ocean model, where it is used to seed icebergs in the Lagrangian iceberg tracking scheme in NEMO. Although this calving

front restriction and the bilinear remapping of meltrate previously described do not maintain the mass and heat conservation of the coupled system, the errors arising from them are not likely to be significant in simulations with timescales less than a century. Since the ocean biogeochemistry component (MEDUSA) is currently technically incompatible with the ocean cavities beneath ice shelves, it is not included in UKESM1.0-ice.

The full details of these and other modifications to UKESM1.0, along with how other UKESM1.0 components are modified

as well as the complete ice-ocean coupling procedure, forcing exchange and its implementation are covered in Smith et al. (2021). Here we only describe briefly the coupling procedure in terms of flux exchange between the Earth system and ice sheet models. We use annual coupling in UKESM1.0-ice, so the boundary topography and ice calving field seen by the climate model, and the boundary forcing seen by the ice sheet model, are updated every year. One simulation year of the coupled model is run sequentially in the following order: After running the climate model for one year with a given static ice sheet

geometry and a given calving field, the coupler processes the annual average of some climate model outputs (ice shelf melt rate and SMB) and sends them to the ice sheet model to be used as its forcing data. In turn, the ice sheet model is run for one year after which its final geometrical state and calving field are processed by the coupler and then used to update the climate model setup for the following simulation year. Testing in the early stages of UKESM1.0-ice development suggested that the 1 year coupling interval that we choose is adequate for this coupled configuration where we are concerned with the ice sheet

response to slow, multi-decadal evolution of the climate. Favier et al. (2019) indicates very little sensitivity to varying the ocean-ice sheet coupling period in their model between 1 month and 1 year, while Zhao et al. (in review) indicates very little sensitivity to the coupling time interval between 0.5 days and 3 months.

**2.2 Experimental Design**

The ScenarioMIP simulations are the CMIP6 effort to coordinate future climate projections by sampling a range of emission

scenarios produced by integrated assessment models (O'Neill et al. 2016). Here we use the SSP1-1.9 and SSP5-8.5 scenarios which represent the lowest and highest anthropogenic forcing levels. Throughout this paper we compare our SSP1-1.9 and SSP5-8.5 projections and our conclusions are based on the differences between these, with the SSP1-1.9 runs effectively acting as control simulations. Since the impact of initialisation shocks, model drifts, and coupling choices will be expressed in both

scenarios, to leading order their differences are independent of these features and reveal the influences of anthropogenic
forcing, which is our main interest.

Since there is a large difference in radiative forcing between the two scenarios, we expect to clearly identify the forced changes generated by the scenarios and therefore a small ensemble of simulations is sufficient. We use a four-member initial condition ensemble for each scenario, with initial states obtained as described in the next section. For both scenarios, we follow the standard simulation years in the ScenarioMIP which cover the period from 2015 to 2100. For some runs, we extend the SSP5-8.5 scenario simulations by 15 years in order to confirm the persistence of model behaviour which only becomes apparent at the end of 21$^{st}$ century, using the SSP5-8.5-Ext scenario (O'Neill et al. 2016).

### 2.3 Initialisation

The ideal initial condition for the coupled model to start the scenario runs would be climate and ice sheet states which are consistent under modern conditions and reproduce present-day observations, but creating such states is extremely challenging. A common practice for obtaining a present-day state for a coupled atmosphere-ocean climate model is to perform a long simulation under pre-industrial climate forcing to achieve an equilibrium, and then to conduct a transient simulation under historical forcing starting from the equilibrium state. In contrast, ice sheet model projections are often initialised without any spin-up, by formally modifying model parameters to obtain the best fit to an observed present-day ice state. Performing a multi-centennial spin-up for the coupled UKESM1.0-ice model is not possible for reasons of computational cost. Moreover, neither the Antarctic nor Greenland ice sheets were likely in static equilibrium states during the preindustrial period, and reliable observational datasets for model evaluation in Antarctic regions are not available prior to the satellite era. Initialisation of coupled climate-ice sheet models is an area of active research (e.g. Lofverstrom et al. 2020) and there is no consensus on best practice.

The simulations described in this paper grew out of model development efforts and were initialised in a rather ad-hoc manner which will be improved in future research. The topic of coupled ice-climate initialisation is a challenging research question and much work is still required to develop best practice in this area. However, any undesirable features that may result from our initialisation procedure are strongly mitigated by our experimental design, which concentrates on analysis of the differences between pairs of simulations with identical initialisations but different radiative forcings. We emphasise that we believe these simulations are the first of their kind, and work on improving our coupled ice sheet–climate initialisation is ongoing.

Given that the simulation length needed by an ice sheet model to achieve an appropriate initial condition may be very different to that required by the climate model, our approach splits the initialisation process into separate climate and ice sheet parts, making use of available model restart data from UKESM and BISICLES simulations under similar forcing regimes. Overall, the process of obtaining the initial ice and climate states in this work has four sequential stages (Fig. 1): an ice sheet model inversion based on observations (Stage A), a preliminary UKESM1.0-ice simulation to generate initial ocean cavity properties (Stage B), a stand-alone ocean simulation to adjust the ocean in UKESM1.0-ice toward a state characteristic of UKESM1.0

simulations for the year 2000 and generate corresponding ice shelf melt rates (Stage C), and a stand-alone ice-sheet relaxation run (Stage D) to overcome the initial shock of forcing the observationally-derived ice sheet state from Stage A with the climate from UKESM1.0. Stage A and B were already completed in previous work (Cornford et al., 2016; Smith et al., 2021) whereas

Stage C and D are carried out in this work.  We outline each of these stages below.

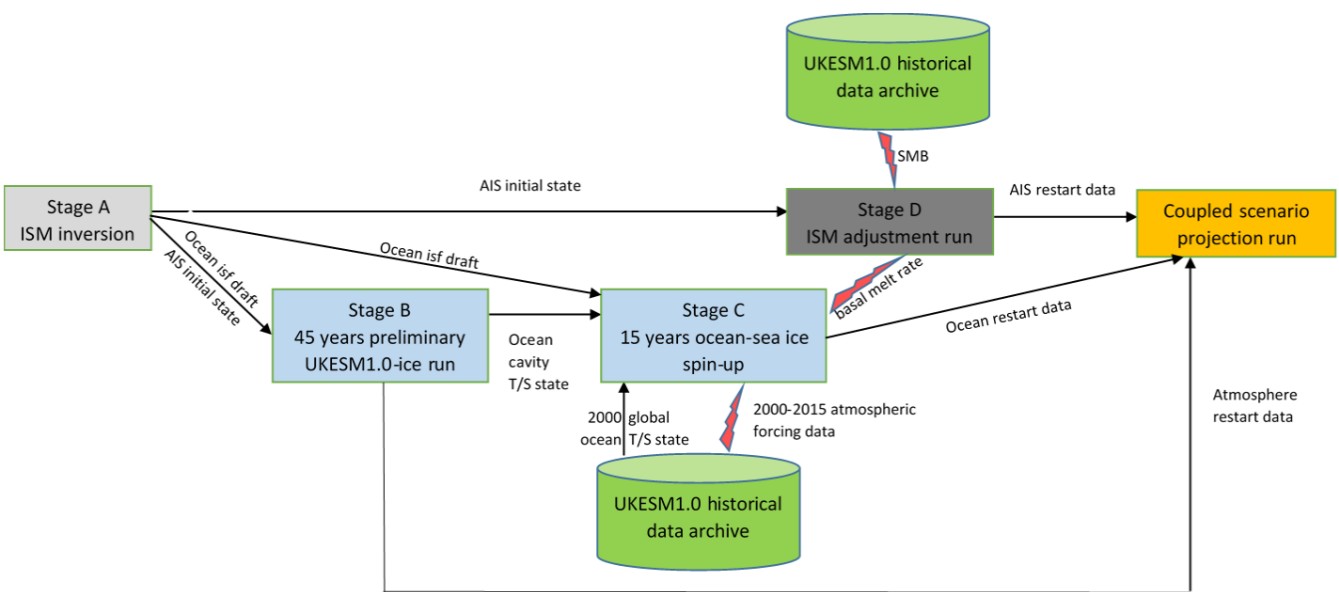

**Figure 1: Stages in developing the initial condition for the coupled UKESM1.0-ice scenario projection. Blue boxes designate the two stages of ocean initialisation. Boxes with grey colours indicate ice sheet model run: the light colour denotes a model inversion whereas**
**the dark colour is an ice sheet model forward run. The black arrows indicate the provision of model states, whereas the red lightning signs indicate the source of forcing data.**

### 2.3.1 Stage A: Ice sheet model inversion

An AIS initial state close to the present day was derived in Cornford et al. (2016). Fields for the basal traction and stiffening coefficients were determined by the solution of an ice sheet inversion procedure (Cornford et al., 2015) such that ice velocities
in BISICLES closely match  surface observations (Rignot et al., 2011). The inversion is done using the stress balance equation keeping the ice sheet geometry fixed, hence basal melting and accumulation play no explicit role in it. An implied mass balance (SMB – basal melting) for this state can be diagnosed from the divergence of the model velocities. The bedrock elevation and ice thickness in this inversion were constructed from the Bedmap2 dataset (Fretwell et al., 2013) and the bedrock was modified (Nias et al., 2016) to avoid a thickening tendency in Pine Island Glacier (Rignot et al., 2014).

This initial state has been used in some stand-alone BISICLES projection simulations (Cornford et al., 2016; Martin et al., 2019) and in a preliminary test simulation of the interactive UKESM1.0-ice (Smith et al., 2021). The latter is exactly the Stage B (Section 2.3.2) of our coupled initialisation scheme. We will also use the initial ice state in the stand-alone BISICLES

relaxation run (Stage D). In addition, we choose this initial state velocity as the reference surface velocity in the evaluation of the early behaviour of our projections in Section 3.1.

### 2.3.2 Ocean Initialisation (Stage B & Stage C)

For CMIP6, UKESM1.0 (which has static ice sheets and vertical ice walls from the surface to the bedrock at the ice front) was spun up to equilibrium over several thousand simulated years with preindustrial forcing (Yool et al, 2021) before an initial condition ensemble of historical simulations was conducted, starting from different points in the preindustrial control run (Sellar et al., 2019). These historical simulations provided the initial states for UKESM1.0 ScenarioMIP projections (Swaminathan et al., 2022), but they do not directly provide states that can be used with UKESM1.0-ice, given their lack of data for the ocean under Antarctic ice shelves.

UKESM1 is a computationally expensive model, and it is not practical to re-run the entire spin-up procedure with open ice shelf cavities. Our strategy to generate initial ocean states for our UKESM1.0-ice scenario simulations is instead to construct approximate modern ocean cavity properties (Stage B), splice them onto UKESM1.0 historical global ocean states and then run short stand-alone ocean simulations to make the water properties in the cavities consistent with the UKESM1.0 historical initial states (Stage C). While there may be numerical shocks and drifts associated with this particular strategy, our experimental design in focusing on the differences between simulations that have been initialised identically significantly mitigates their influence.

**-Stage B: Preliminary UKESM1.0-ice simulation to generate approximate ocean cavity properties**

As noted above, Smith et al. (2021) ran a preliminary UKESM1.0-ice simulation using the ice sheet initial state from Stage A. In order to maintain the conformity between the ocean and ice sheet domain, the ice shelf draft topography for the ocean domain on the NEMO eORCA1 grid was interpolated from the ice shelf geometry from the BISICLES grid. In this simulation, the global ocean was initialised with the EN4 climatology ocean data (Good et al., 2013) which was simply extrapolated into ice shelf cavities around Antarctica. This UKESM1.0-ice configuration was then run for 45 years under a repeated 1970 greenhouse gas forcing.

Although the main aim of this preliminary simulation was to test the computational stability and robustness of UKESM1.0-ice, using the final ocean cavity state of this simulation in our ocean initialisation process (Stage C) provides two advantages. Firstly, there are only small discrepancies in the ocean domain between the preliminary run and Stage C since the ice shelf geometries in both are derived from the inversion in Stage A. Secondly, after being run for 45 years the preliminary simulation has a physically plausible density structure to initialise Stage C with. Since an equilibrated ice-ocean state is not necessary in Stage B, a run of 45 years is adequate.

**-Stage C: Stand-alone ocean-sea ice integration**

To create four initial ocean states for UKESM1.0-ice projection ensembles which are compatible with the final state of UKESM1.0 CMIP6 historical simulations but also include ice shelf cavities, we stitched the ocean fields from the ice shelf

cavities in Stage C onto four UKESM1.0 global ocean states representative of early 21st century; these will be described in more detail below.

Since these stitched-together ocean states contain discontinuities across the ice front and require some time to come into balance, we conduct stand-alone ocean-sea ice integrations starting from each of these four states, bringing the ocean in the ice shelf cavities in line with the early 21st century global ocean. The stand-alone ocean-sea ice configuration for these four

integrations are very similar to the ocean-sea ice component of UKESM1.0 (Sellar et al., 2016) which is based on the Global Ocean (GO6) configuration (Storkey et al., 2018). Since this stand-alone configuration includes static ice shelf cavities which are absent in UKESM1.0, some extra settings (Mathiot et al., 2017) specific to the circulation and melting process in the cavities are added to accommodate the 3-equation implementation.

We chose a period of 15 years for these stand-alone integrations on the assumption that the residence time for water in the ice-

shelf cavities is shorter than a decade (Nicholls and Østerhus 2004; Loose et al. 2009). Although the water in our hybrid global ocean and ice shelf cavity states starts from rest, we find that in practice 15 years is a sufficient length of time to flush the cavities with water from the global ocean. As indicated in Fig. 2, temperature discontinuities across the ice front have settled down within a year in all members.

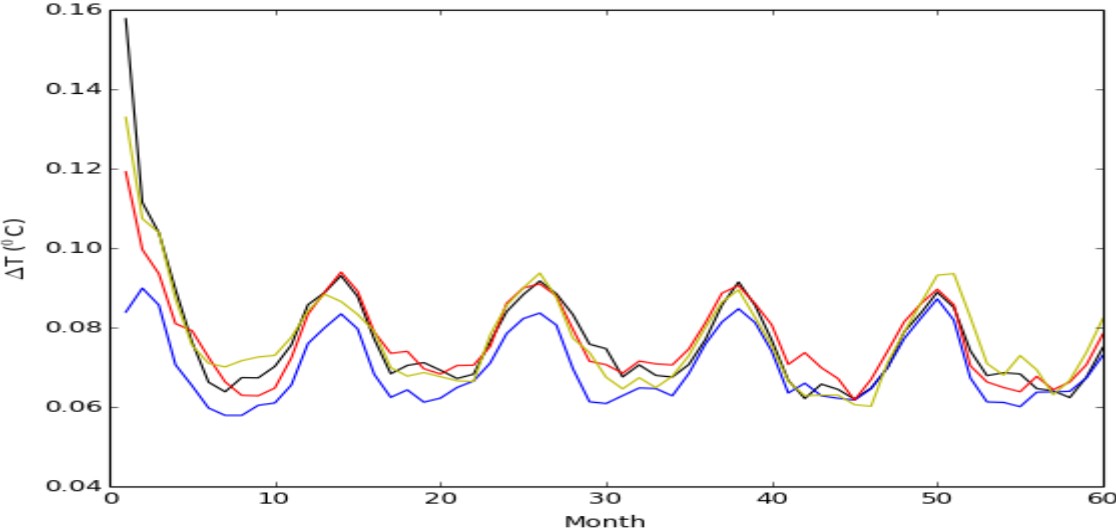


**Figure 2: Monthly time-series of temperature difference across ice fronts averaged over depth and along the entire Antarctic coast in the first 5 years of ocean stand-alone integration (Stage C) for all (four) members. The blue line is the member that was not initialised with the global ocean from the UKESM1.0 CMIP6 historical ensemble.**

As we intend to produce balanced ocean states appropriate for starting projections from the year 2015, these stand-alone ocean-

sea ice simulations are regarded as beginning in the year 2000 and run forward for 15 years. In order to achieve states compatible with the year 2015 in the UKESM1.0 CMIP6 historical ensemble (Yool et al., 2021), three out of our four initial global ocean states to be joined to the cavity data are taken from this ensemble. Our three members were chosen to maximise

the range of ACC strength and Southern Ocean annual average SST across the 19 UKESM1.0 historical ensemble members in the year 2000. The surface forcing for these three simulations are provided by time dependent atmospheric fluxes archived

from the corresponding UKESM1.0 historical member over the 2000-2014 period, with additional restoring of surface temperature and salinity to further prevent drift in the global ocean from the trajectory of the original UKESM1.0 simulations. The global ocean state for our fourth member was taken from the end of the preliminary UKESM1.0-ice simulation (Stage B) and this member was forced with atmospheric fluxes used by one of the first three members. Since Stage B had not been run long enough to drift far from the 1995-2014 average EN4 climatology it was initialised with, it provides an initial ocean state

more representative of the observed modern ocean than the UKESM1.0 historical ensemble, which contains systematic biases characteristic of UKESM1.0. Our UKESM1.0-ice projection initialised with this fourth member can be used in our experimental design to assess if UKESM1.0's systematic biases might have significant impact on the climate and ice sheet evolution we are interested in.

### 2.3.3 Stand-alone Ice sheet relaxation (Stage D)

It is unlikely that the ice sheet state from Stage A will be entirely consistent with the UKESM1.0 climate from the year 2015, and some coupling shock is inevitable when the climate and ice sheet are coupled together. If this shock is too large it might produce unphysical trends and could overwhelm the stability of the ice—climate coupling scheme. We reduce the size of the coupling shock by first conducting short stand-alone ice sheet model relaxation simulations using basal and surface mass balance forcing derived from UKESM1.0 before coupling all the components together interactively. For our projections it

would be inappropriate to bring the ice sheets into equilibrium with the year 2015 climate in UKESM1.0 since in reality we observe the modern ice sheet to be out of balance, and Stage D does not aim to do this, simply to mitigate the worst of the immediate coupling shock and ensure numerical stability of the initial steps of our interactive ice-climate system. Furthermore, it is important for projections that the initial state of the ice sheet is close to that observed in the present day, especially for the positions of the grounding lines. The ice sheet geometry from Stage A is constrained to be close to reality, so it is beneficial

for the relaxation runs in this stage to be as short as practically possible so that the ice sheet starts from a realistic position. Any systematic artificial drift present in our simulated ice sheet evolution due to this procedure will be equally present in the following pairs of SSP5-8.5 and SSP1-1.9 projections and can be accounted for in our analysis by looking at the differences between the two scenarios.

The BISICLES setup for the stand-alone relaxation simulations is the same as within UKESM1.0-ice. We start with the ice

sheet state from Stage A then run 4 ice sheet simulations, using the ice shelf melting diagnosed from each stand-alone ocean run described in Stage C, and an SMB field taken from a UKESM1-ice historical simulation (Smith et al., 2021). Our initial two adjustment runs used an annual average SMB and basal melt from the year 2014 whilst two later runs used the 2010-2015 average. We saw no significant impact from using different time averaging periods for these boundary conditions. For each stand-alone ice sheet run we diagnose the root-mean-square rate of thickness change averaged over the ice sheet which sees

an initial large spike in magnitude when the UKESM1.0 boundary conditions are introduced before reducing to a steady lower

value. Once past this initial spike (up to 1.4m/yr), after about 25-30 years of simulation an approximately steady RMS thickness rate of change (less than 0.4 m/yr) is reached. At this point in the stand-alone ice sheet relaxation runs, some remaining drifts are found in isolated cells near the coast but not in important regions of dynamically evolving glaciers feeding ice shelves.

### 2.3.4 Merging the initial states

The members of our ensemble are initialised at 2015 with one of these ice sheet states from Stage D, the ocean state that produced the ice shelf basal melt rate that corresponds to it from Stage C, and (due to a technical incompatibility between UKESM1.0 and UKESM1.0-ice configurations in the UM) the atmosphere state from the preliminary UKESM1.0-ice simulation in Stage B. The timescales of physical adjustment of the atmosphere and land surface are rapid compared to the ocean and ice sheet and the well-mixed components of the atmospheric composition are specified by concentration in this

configuration of UKESM according to the scenario year, so initialising the atmosphere model with a state appropriate to 1970 does not represent a large inconsistency in the global ESM at 2015.

However, the chemistry scheme in UKESM provides prognostic ozone concentrations, and the 1970 atmospheric state has no history of late 20th century ozone depletion. It thus contains column ozone concentrations that are too high for 2015, especially in Southern Hemisphere spring. However, ozone-depleting substances are specified by concentration in our model as part of

the scenario forcing, and column ozone concentrations reduce to levels that are appropriate for the scenario over the first 5-10 years of the UKESM1.0-ice simulation, as would be expected by the overturning timescale of the Brewer-Dobson circulation controlling stratospheric mixing (Abalos et al. 2021). This temporary overestimate of southern hemisphere ozone would be expected to increase seasonal radiative forcing and affect regional wind patterns in the first few years of our simulations, but we see no evidence of any long term impact of the ozone initialisation.

### 3 Results


In this section we describe the results of the SSP1-1.9 and SSP5-8.5 simulations, concentrating on the factors important for the ice sheet mass balance. Subsection 3.1 evaluates the initial evolution of the ensemble against some modern observational datasets, followed by sections which detail projection of changes to 2100 in the two climate change scenarios. Subsection 3.2 analyses changes in ice shelf basal melting, while subsection 3.3 analyses SMB. Throughout, the terms SSP1-EM, SSP5-EM, and ALL-EM refer to the ensemble means of the SSP1-1.9 ensemble, SSP5-8.5 ensemble, and all simulations, respectively.


### 3.1 Evaluation of the initial model state

We consider the first few years after interactively coupling the ice sheet to the climate model to be potentially biased by further coupling shock, so we choose the mean state over the period from the year 2020 to 2030 to evaluate the early state of the coupled model. The forced responses of many climate variables do not diverge among different scenarios in the first decade

of projections (Abram et al., 2019; Barnes et al., 2014; Bracegirdle et al., 2020) and are less likely to differ significantly from

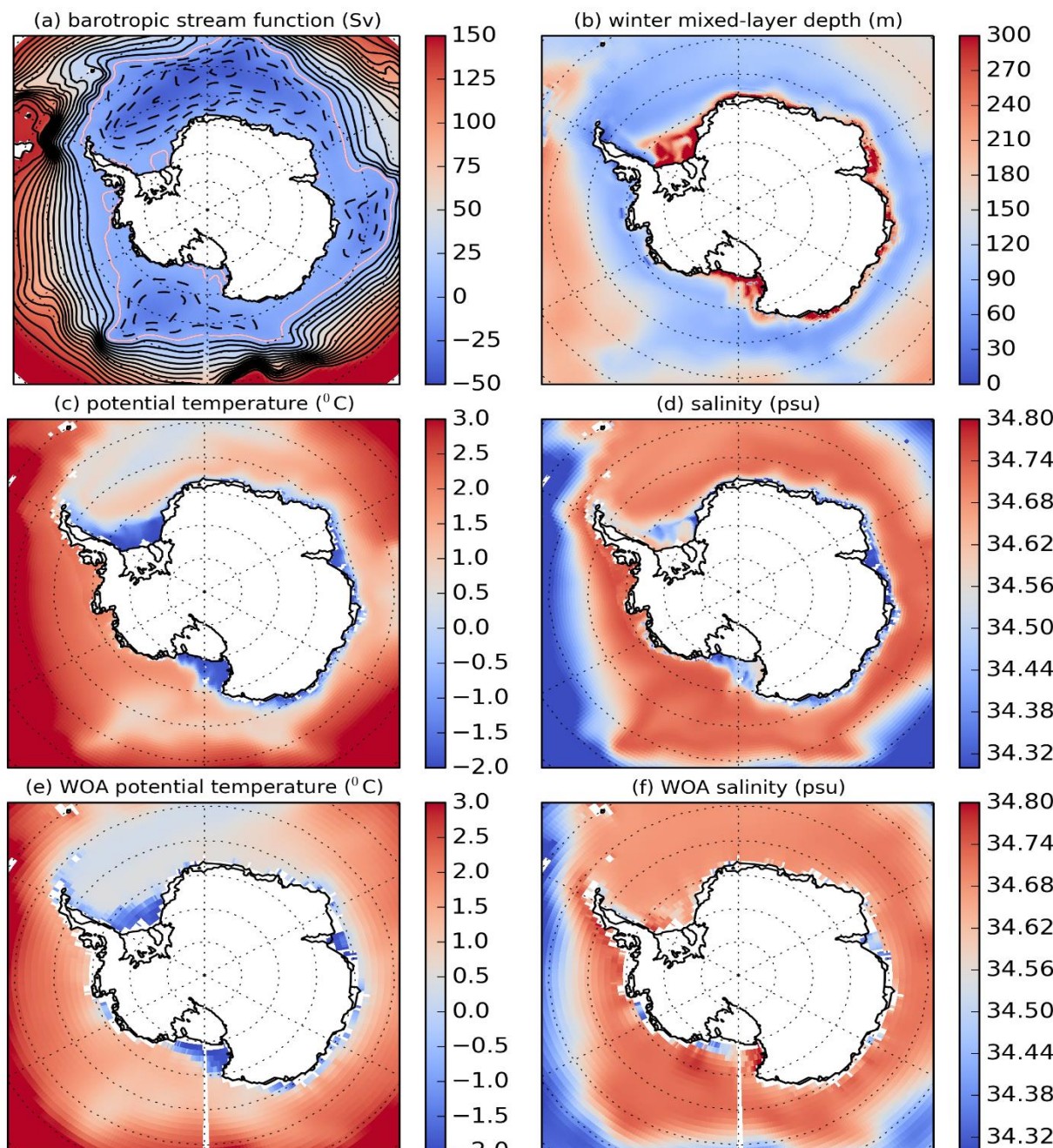

**Figure 3: Mean 2020-2030 ALL-EM of (A) barotropic stream function [contours every 10Sv, pink contour = 0Sv], (B) winter [June-July-August] mixed layer depth (not available in ice shelf cavities), and (C) potential temperature and (D) salinity averaged between 300-1000m. WOA 2013 climatology of averaged 300-1000m of (E) potential temperature (Locarnini et al, 2013), and (F) salinity (Zweng et al, 2013). Ice shelf cavities are masked in all plots. In (C) and (D), a few areas outside the ice sheet with bathymetry < 300m are masked, whereas in (E) and (F) there are bigger masked areas outside the ice sheet extent due to the absence of observations.**


the present day. For this evaluation we use the modelled barotropic stream functions, mixed layer depth, and mean 300-1000m temperature salinity (Figs. 3a-d) where the latter represent the water properties over the depth range important to ice-shelf
melting. While recent observations are broadly available for temperature and salinity (Figs. 3e-f), there are no complete observational data sets for the barotropic stream functions and mixed layer depth around the south polar region.

Figure 3a demonstrates the ALL-EM of the simulated 2020-2030 barotropic streamfunction south of 55 °S, which resembles the observed Antarctic Circumpolar Current path around Antarctica (Sokolov and Rintoul, 2009) and shows that the model simulates the existence and strength of the Ross, Weddell and Australian-Antarctic subpolar gyres as reported by many coupled
climate models (Wang and Meredith, 2008).

The ALL-EM of continental shelf temperature (Fig. 3c) captures the general pattern of cold shelf water in the Ross and Weddell seas and warm water in the Amundsen and Bellinghausen seas compared to the World Ocean Atlas 2009 climatology (Fig. 3e, Locarnini et al, 2013), although with a slight warm bias in the latter seas. The salinity (Fig. 3d) in the Ross and Weddell continental shelves is too fresh compared to observations (Fig. 3f, Zweng et al., 2013), which results from an accumulation of
fresh biases in these continental shelves through the Stage C ocean initialisation period and the following first 15 years of the scenario runs. Since UKESM1.0 historical runs (Sellar et al., 2019) do not suffer from this shortcoming, the fresh biases in the Stage C ocean initialisation stage are most likely due to a combination of different features which are absent in UKESM1.0, such as the choice of initial salinity data in ice shelf cavities, time-evolving Ross Ice Shelf basal melting, and the limitations

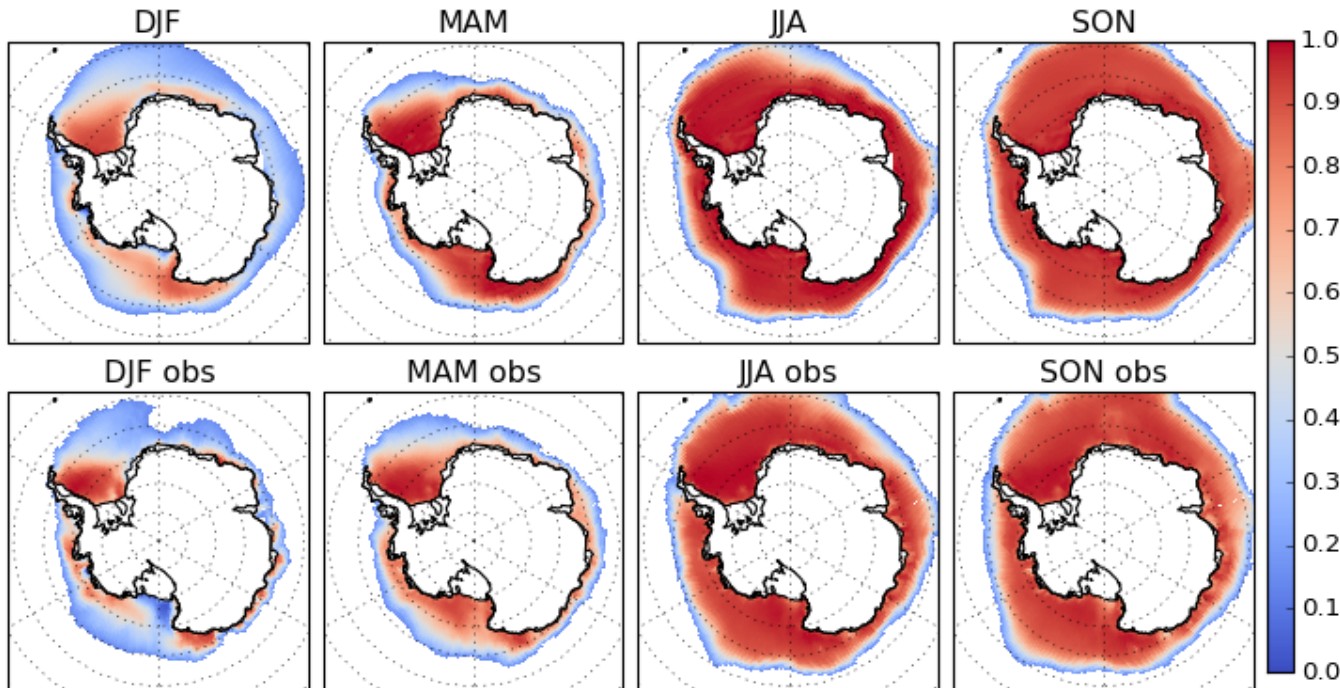

**Figure 4: 2020-2030 ALL-EM (top) and HadISST1 (bottom) fractional sea-ice concentration in each season. The extent is defined by a threshold 0.15 concentration.**

of surface forcing representations. Nevertheless, these fresh biases do not seem to affect ocean temperatures; the deep winter mixed layer depths (MLD) on the Ross and Weddell shelves (Fig. 3b) indicate the continuous presence of cold shelf water that ventilate the seabed, produced by sea-ice formation. The opposite situation occurs on the Amundsen and Bellinghausen

continental shelves, where shallow mixed layers indicate that the relatively shallow surface water allows warm water in the subsurface depth to flood these shelf seas. This subsurface water is slightly warmer and saltier than observations, but the model has very low bias compared to many climate models in this region presented in Heuzé (2020).

Over this period, the ALL-EM of sea ice concentration (Fig. 4) shows good agreement with the HadISST1 observations (Rayner et al., 2003) overall. However, near-shore polynyas in the Ross Sea (DJF season in Fig. 4) are not as extensive in the

model as they appear in the observations. This may not, however, contribute much to the fresh biases on the Ross continental shelf that are present in our configuration, since these polynyas are also underestimated in the UKESM1.0 historical simulation (Sellar et al., 2019), yet that model has a reasonably good bottom salinity in these regions.

The behaviour of the AIS in the 2020-2030 period is shown by the ALL-EM of ice sheet model outputs in Fig. 5. The major ice-dynamical imbalances exhibited by the observed 1992-2017 average rate of elevation change (Fig. 5a, Shepherd et al.,

2019) are also demonstrated by the model (Fig. 5b), where high thinning rates occur in major fast-flowing outlets in the Amundsen Sea region such as Pine Island and Thwaites glaciers. Some thickening is present on the Kamb Ice Stream in the Siple Coast region although at a slightly lower rate than is observed. These features are largely retained from the original BISICLES state in Cornford et al. (2016) where the basal traction coefficients have been tuned to match the modelled ice speed with observations, enabling the model to simulate thinning in places where ice flow has recently accelerated or thickening

where it has decelerated. On the other hand, the observed thinning of Totten Glacier is only partially reproduced in our simulation during the 2020-2030 period. There are speckly patterns of thinning and thickening across the grounded ice sheet, however the magnitude is not large relative to the dynamic thinning signals of interest and these patterns are predominantly associated with the divergence implied by the initial state in Stage A. They appear in both SSP1-1.9 and SSP5-8.5 scenario simulations (Fig. 5c) and disappear in almost all grounded sectors when the simulations are differenced. This means that the

climatically-forced responses that we are focussing can be distinguished from these patterns.

On the grounded area of the ice sheet, also owing to the tuned basal traction coefficients, the ALL-EM of surface velocity (Fig. 5d) in this early part of the simulation generally shows insignificant differences from the reference surface velocity (Cornford et al., (2016) also used in the Stage A initialisation), although with some slight acceleration in the Amundsen Sea ice streams. The flow of ice slows markedly across most ice shelf regions during this 2020-2030 period (Fig. 5d). Reductions of more than

100 m/yr take place on several ice shelves, with the largest difference being in the Larsen region. Most of these reductions occur in the first year of the Stage D ice sheet relaxation, when the ice is adjusting to the UKESM1.0-ice SMB and basal melt forcing, whereas the change of surface velocity from 2015 to 2030 is negligible (not shown). These changes result from a coupling shock arising from discrepancies near the grounding lines between the SMB and basal melt rate in our initial adjustment process and the SMB and basal melting implicit in the inverted reference velocities of Cornford et al. (2016). The

shock is mainly dominated by the basal melt forcing. Since the ocean model cannot accurately represent the very thinnest parts

of the cavities near the grounding lines, no melting is simulated there in Stage C. When this basal melt boundary condition is given to the ice shelf in Stage D, the ice rapidly thickens in these areas leading to small grounding line advances. The increase in drag from these re-grounded areas is instantaneously transmitted through the ice shelves, causing the ice shelves to rapidly decelerate.

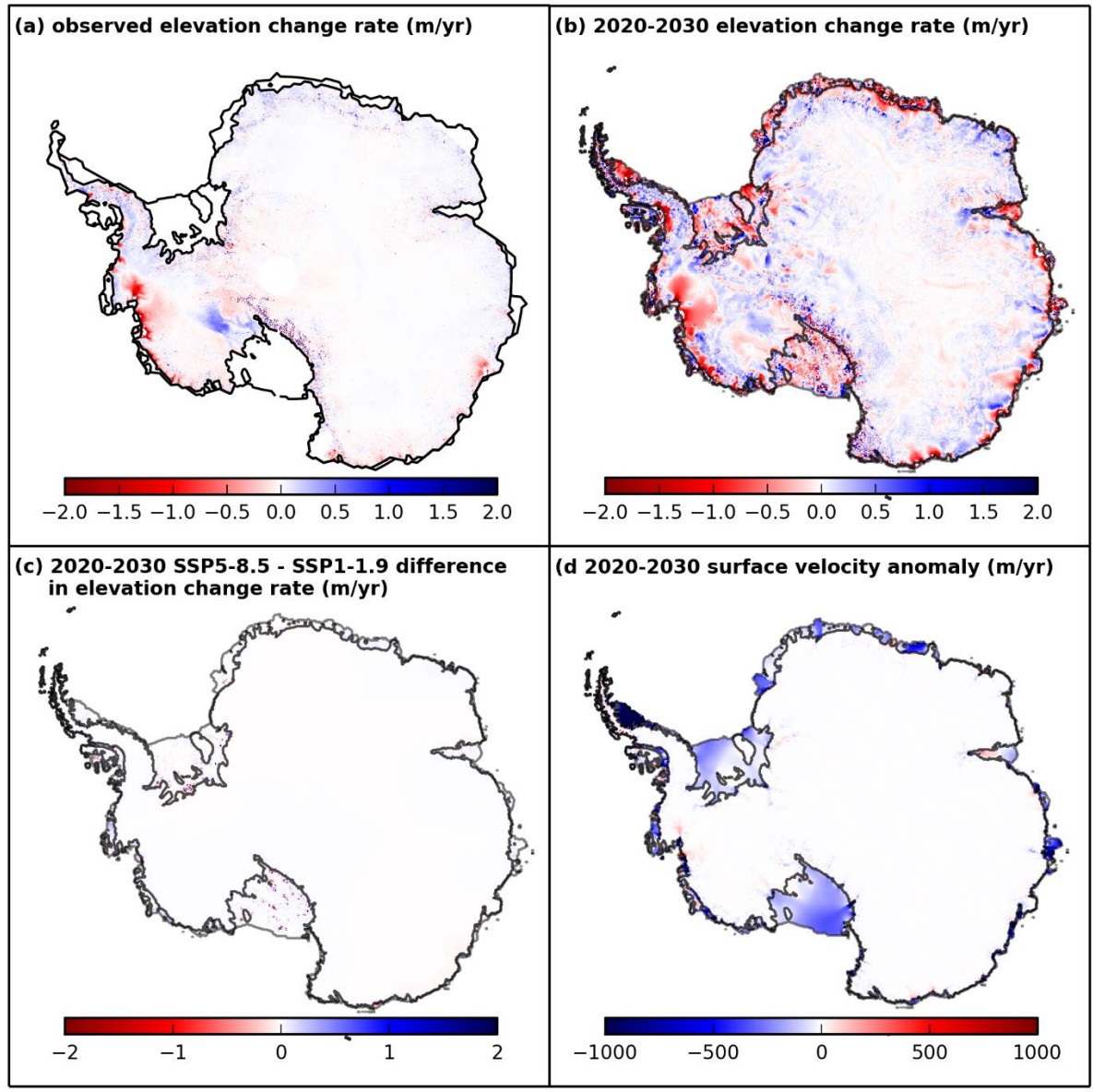


**Figure 5: (a) IMBIE 1992-2017 mean observed elevation change rate (m/yr). Ice shelves and some grounded regions in the Antarctic Peninsula are not covered by the data. (b) 2020-2030 mean modelled elevation change rate (m/yr). (c) 2020-2030 difference of mean elevation change rate (m/yr) between the SSP5-8.5 and SSP1-1.9 scenario. (d) 2020-2030 mean surface velocity anomaly (m/yr) with respect to the reference ice state (Cornford et al., 2016). The black and grey lines in (b), (c) and (d) are the ice sheet grounding lines and ice shelf fronts respectively.**


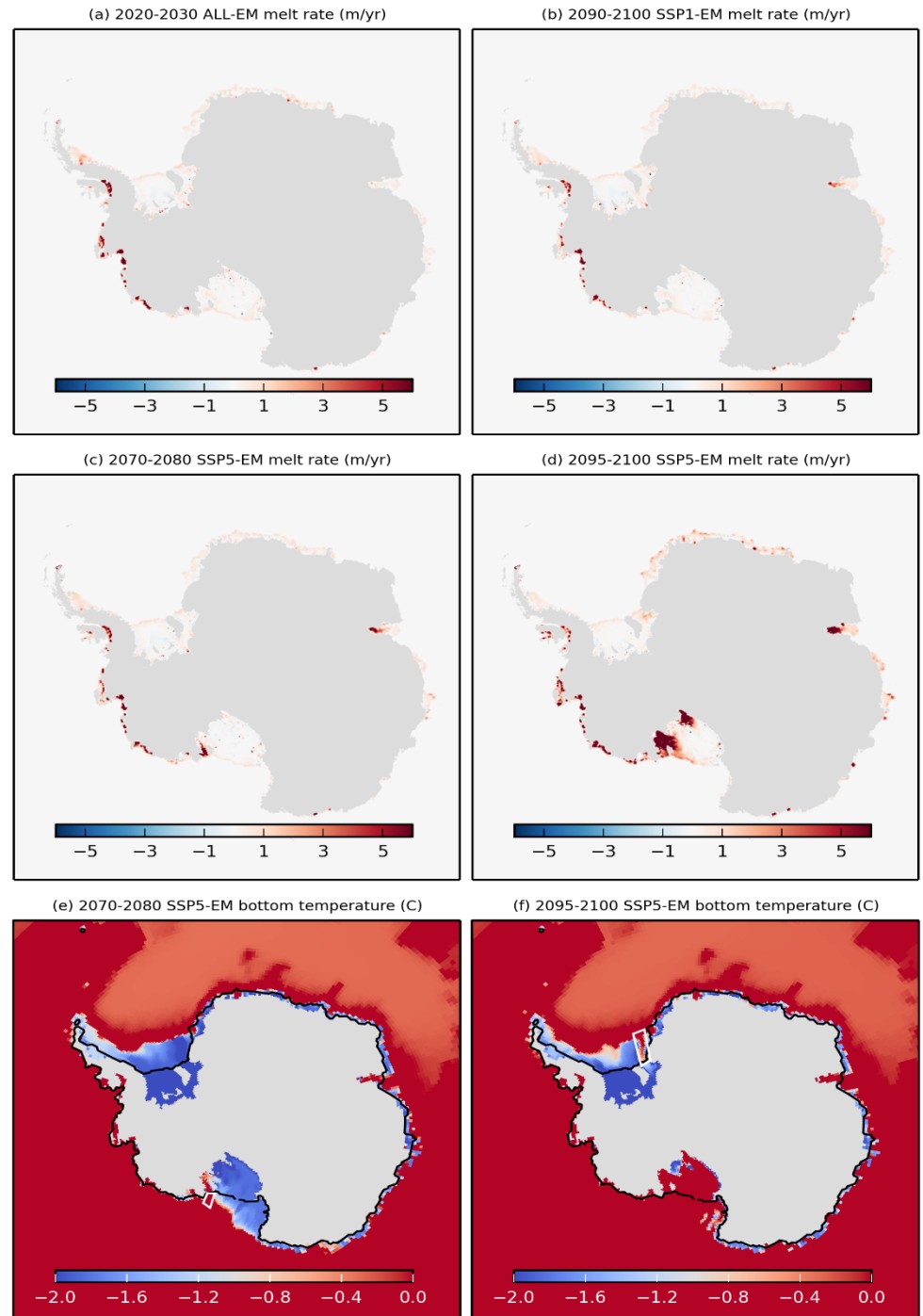

**Figure 6: Antarctic ice shelf melt rates (m/yr) (A) average 2020-2030 ALL-EM, (B) average 2090-2099 SSP1-EM, (C) average 2070-2080 and (D) average 2095-2100 SSP5-EM; Bottom temperature (ºC) (E) average 2070-2080 and (F) average 2095-2100 SSP5-EM. The white boxes in (E) and (F) represent the Little America Basin and Filchner Trough respectively, whereas the black line indicates the ice sheet extent (coastlines).**

Beneath the ice shelves, the ALL-EM of basal melt rates in the 2020-2030 decade (Fig. 6a) shows a similar spatial pattern to present day observations (Rignot et al., 2013; Adusumilli et al., 2020) (the latter is shown in Fig. A1 of Appendix A for visual comparison) under warm ice shelves in West Antarctica as well as cold ice shelves in Ross, Weddell and East Antarctica sectors. However it does not reproduce the high basal melt underneath the Totten Ice Shelf.We do not focus on particular
details of high basal melting in the Amundsen region given the low resolution of the Amundsen cavities, but we further compare the integrated basal melt flux in this early period of the simulations with the estimate from those observations (Table 1) where we refer to the data from Rignot et al. (2013) and Adusumilli et al. (2020) as Obs-Rig and Obs-Ad respectively. Since the uncertainties in the observational datasets are very large and the disagreement between two datasets is also large, we do not perform detailed statistical analysis on them, and compare the interval estimates only. In this loose comparison, the ALL-
EM of area-integral melt flux across Antarctica generally agrees with the observations. The simulated total melt flux over all Antarctic ice shelves (981.8 ± 80.5 Gt/yr) is within the range of Obs-Ad although slightly below the lower end of Obs-Rig. In the warm Amundsen region, the basal melt fluxes under the Pine Island Glacier (PIG) and Thwaites ice shelves coincide with the Obs-Rig and the Obs-Ad range respectively. Under the large Ross and Filchner-Ronne ice shelves, we simulate melt fluxes that fall within the observed ranges of Obs-Ad where their mean values are very close to each other. When compared with the
Obs-Rig, an agreement is only obtained for the Ross Ice Shelf melting, whereas the simulated Filchner-Ronne Ice Shelf melting is much lower than the observed range.

**Table 1: Ice shelf basal melt flux (in Gt yr$^{-1}$) under some selected Antarctic ice shelves from the model and observations. The result from the model is represented by the whole ensemble mean (ALL-EM) which encompasses both scenario members averaged between**
**the year 2020 and 2030. In both observations, the Ross basal melt flux is split into the Ross East and Ross West regions, whereas the Filchner-Ronne melt flux is split into the Filchner and Ronne ice shelves.**

| Ice shelf | ALL-EM (model) | Obs-Rig (Rignot et al., 2013) | Obs-Ad (Adusumilli et al., 2020) |
|---|---|---|---|
| PIG | 103.2 ± 10.7 | 101.2 ± 8.0 | 76.0 ± 8.7 |
| Thwaites | 80.3 ± 10.5 | 97.5 ± 7.0 | 81.1 ± 7.4 |
| Ross | 57.5 ± 4.8 | | |
|    Ross East | | 49.1 ± 14.0 | 31.0 ± 45.3 |
|    Ross West | | −1.4 ± 20.0 | 26.6 ± 69.2 |
| Filchner-Ronne | 51.7 ± 9.6 | | |
|    Filchner | | 41.9 ± 10.0 | 33.5 ± 29.6 |
|    Ronne | | 113.5 ± 35.0 | 21.2 ± 119.9 |
| All ice shelves | 981.8 ± 80.5 | 1325.0 ± 235.0 | 1173.1 ± 148.5 |

**3.2 Projections of shelf oceanography and ice shelf basal melting**

There is a slight decrease of the total basal melt flux in both the SSP1-EM and SSP5-EM from the start of the simulation until the beginning of the 2060s (Fig. 7a) with a decreasing trend of -52 Gt/decade and -57 Gt/decade respectively, but thereafter
the two scenarios diverge. This timescale is less than two decades after the mid 2040s, which Barnes et al. (2014) and Bracegirdle et al. (2020) define as the period for which the responses to radiative forcing scenarios begin to clearly diverge for the westerly jet and some other surface climate variables over the Antarctic and Southern Ocean.

The SSP1-1.9 scenario runs do not show a drastic change of melt rate pattern (Fig. 6b) or area-integral melt flux (Fig. 7) during the 21$^{st}$ century. The only exception is under the Amery Ice Shelf, where the SSP1-EM melt rates become high close to the
grounding line in the last 10 years of the run.

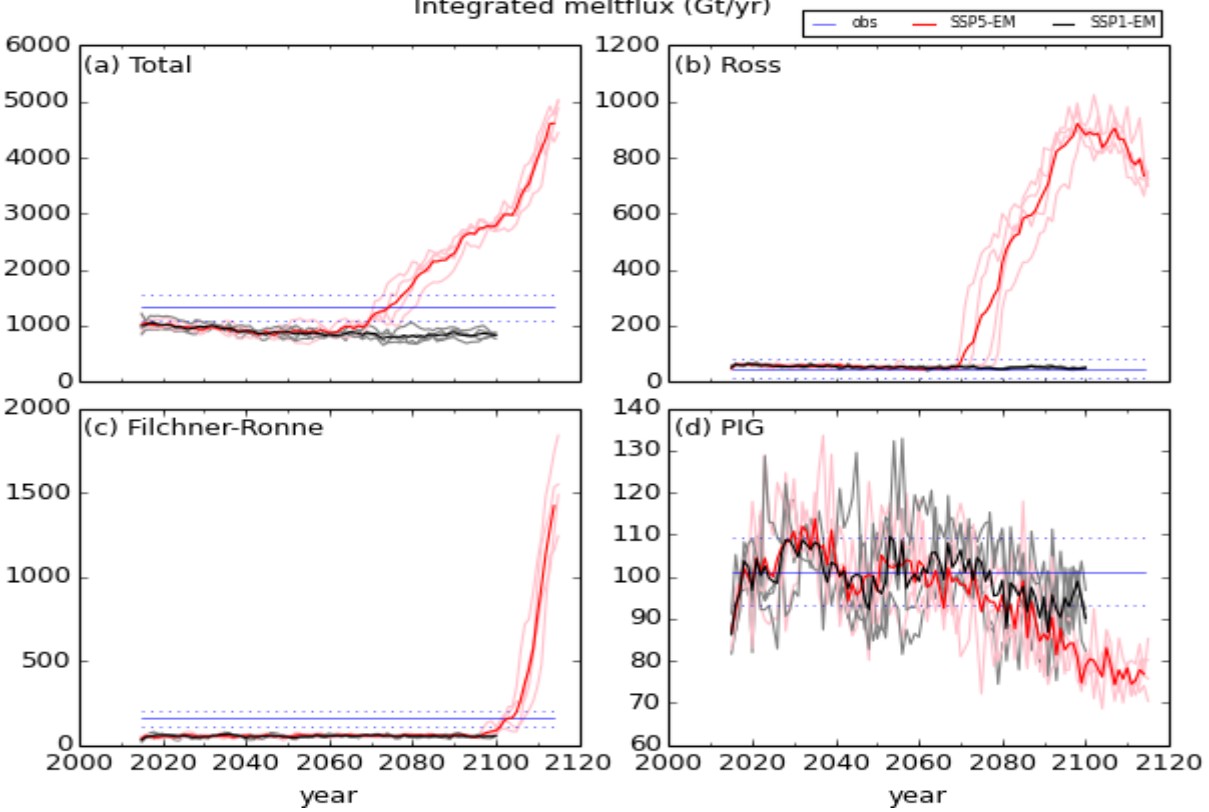

**Figure 7: Time-series of melt flux (Gt/yr) under Antarctic ice shelves. Pink lines are SSP5-8.5 ensemble members and grey lines are SSP1-1.9 members. The red and black lines are the ensemble mean of SSP1-1.9 and SSP5-8.5 respectively, the blue lines are from Rignot et al., (2013) observations with the solid and dashed linestyle representing the mean and uncertainty limit respectively.**

In the high emission SSP5-8.5 scenario runs, there are some notable increases in melting within the second half of the simulation. These are most obvious for the large, cold Ross Ice Shelf (Figs. 6c-d and 7). The sudden rise in Ross Ice Shelf basal melting in the SSP5-8.5 scenario starts around the year 2070 with the incursion of warm water into the eastern Ross Sea area through the Little America Basin (Fig. 6e). There is a time variability of 10 years between the SSP5-8.5 ensemble members

for the onset of this event, with the earliest and the latest being 2068 and 2078 respectively (Fig. 7b). By the end of the 21st

century (Fig. 6f), the area around the southern grounding line of the Ross Ice Shelf is full of this warm water.

A similar strong melting occurs under the Filchner Ice Shelf in the SSP5-8.5 scenario which starts off at the end of the century (Fig. 6d), with a closer time agreement among the ensemble members between the years 2094 and 2097 (Fig. 7c). Here, the Warm Deep Water gains access through the Filchner Trough (Fig. 6f), as found by previous studies (Hellmer et al., 2012; Naughten et al., 2021). As this strong melting only becomes apparent at the end of the 21st century in our simulations, we

extend the SSP5-8.5 projections to the year 2115 in order to verify the persistence of this signal, which is confirmed by the drastic increase of melting under this ice shelf during the 15 years of extension (Fig. 7c). In the Amundsen Sea, despite the increasing total Antarctic melt flux since the 2050s, there is no sign of increase in the basal melting under the Pine Island Glacier Ice Shelf in our SSP5-8.5 simulations (Fig. 7d).

The next subsections will discuss some details of the oceanography related to the basal melting under these ice shelves. The

analyses are taken from the results of one member from each ensemble, which are taken as representative since the members within each ensemble all agree on the overall changes in ice shelf melting.

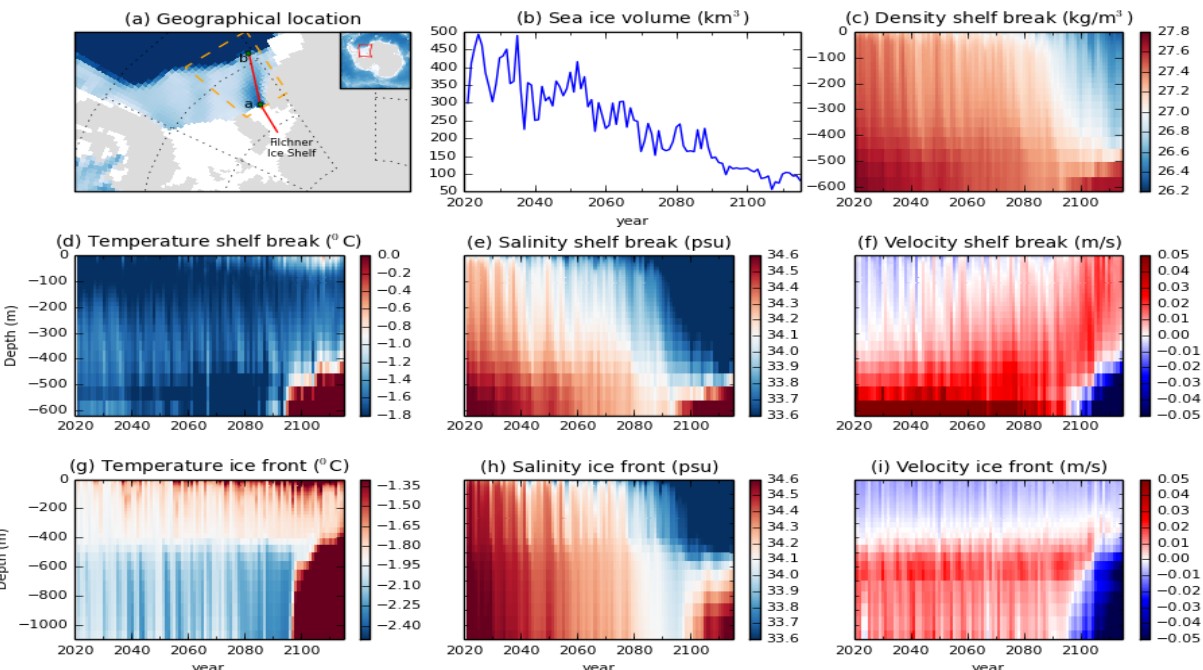

**Figure 8: Filchner Trough profiles in the SSP5-8.5 scenario. (a) The chosen section on the Filchner Trough is indicated by the red**
**line segment between the green dots a and b. The entire red line denotes the meridional section used in Figure 9. The dark and light blue colours represent the deep ocean and the continental shelf in the Weddell Sea, respectively, whereas the grey and white colours represent the grounded ice sheet and the Filchner-Ronne Ice Shelf, respectively. (b) sea ice volume on the area enclosed by the orange dashed lines in (a). (c-f) The timeseries of potential density, temperature, salinity and meridional velocity, respectively at the shelf break indicated by the green dot b in (a). (g-i) The timeseries of temperature, salinity and meridional velocity at the ice front**
**indicated by the green dot a in (a).**

### 3.2.1 Filchner Shelf

The incursion of Warm Deep Water into the Filchner Ice Shelf cavity has been a topic of research since the work of Hellmer et al. (2012), who first simulated this phenomenon in an ocean-sea ice model forced by a climate model projection. Later work (Timmermann and Hellmer, 2013; Hellmer et al., 2017; Daae et al., 2020; Naughten et al., 2021) have investigated this topic
comprehensively in various experimental setups, and our results share many aspects with those works. That being the case, we do not present an analysis of this feature of our simulations and refer readers to those citations for details of this phenomenon. In our SSP5-8.5 projections, this incursion begins in the late 21$^{st}$ century as demonstrated in the previous section. Figure 8 shows the SSP5-8.5 time series profiles for two water columns along the Filchner Trough (on the shelf break and at the ice front, as indicated in Fig. 8a). During this projection the sea ice production over the continental shelf gradually decreases under
climate change (Fig. 8b), and the ocean gradually freshens and gets less dense (Figs. 8c-e). Water on the continental shelf starts off denser than the northern deep ocean at the same depth, but eventually it freshens sufficiently that the deep ocean is denser. The deep flow is therefore northward throughout the simulation, until it changes to southward at the end of the run (Fig. 8f). The changing direction of the bottom flow at the sill depth marks the intrusion of the Warm Deep Water into the Filchner Trough (Fig. 6f). This enables direct access of the warm water to the ice shelf cavity (year 2095-2115 in Figs. 8g-i) which then
leads to a significant increase of basal melting to the south of Filchner ice front (Fig. 7c).

Figure 9 shows a meridional section through the Filchner Trough (red line in Fig. 8a) at the beginning and end of the simulations. It shows that freshening extends to the south into the cavity and slightly to the north at the slope front for both scenarios (Figs. 9b,e,h), with the SSP5-8.5 scenario having the stronger freshening until the incursion starts (Fig. 9h). This agrees with previous studies (Timmermann and Hellmer, 2013; Hellmer et al., 2017; Daae et al., 2020; Naughten et al., 2021)
where continental shelf freshening also precedes the warm water incursion into the Filchner Trough. A comparison of potential density profiles (Figs. 9c,f,i) demonstrates the importance of the meridional density gradient at the sill depth, as the intrusion starts as soon as the density north of the sill is higher than inshore (Fig. 9f). The 15 years of SSP5-8.5 extension run then result in the continental shelf and cavity being filled by the Warm Deep Water (Figs. 9j-l).

Warm Deep Water intrusions into the Filchner Trough in earlier studies (Hellmer et al., 2012; Timmermann and Hellmer,
2013; Hellmer et al., 2017) also occur under a similar high emission scenario forcing (where sea ice volume on the continental shelf decreases significantly). However, the strong basal melting in those studies began earlier (in the 2070s) than in any of our SSP5-8.5 simulations. In addition to the decrease in sea ice volume, the freshening of the continental shelf in the SSP5-8.5 run also receives contributions from basal melting under the Filchner and neighbouring ice shelves (not shown). This is also found by Naughten et al. (2021), who further conclude that the intrusion only starts after a 7 °C global mean surface
warming above the pre-industrial state. In the UKESM1.0 SSP5-8.5 run analysed by Naughten et al. (2021), the global mean surface temperature reaches 7 °C warming at the end of the 21$^{st}$ century, as it does in our SSP5-8.5 UKESM-1.0-ice simulations when the strong melting of the Filchner starts. We also note that the subsurface temperature on the continental shelf initially decreases in our simulations, in agreement with the 'two timescales' of change reported by Naughten et al. (2021).

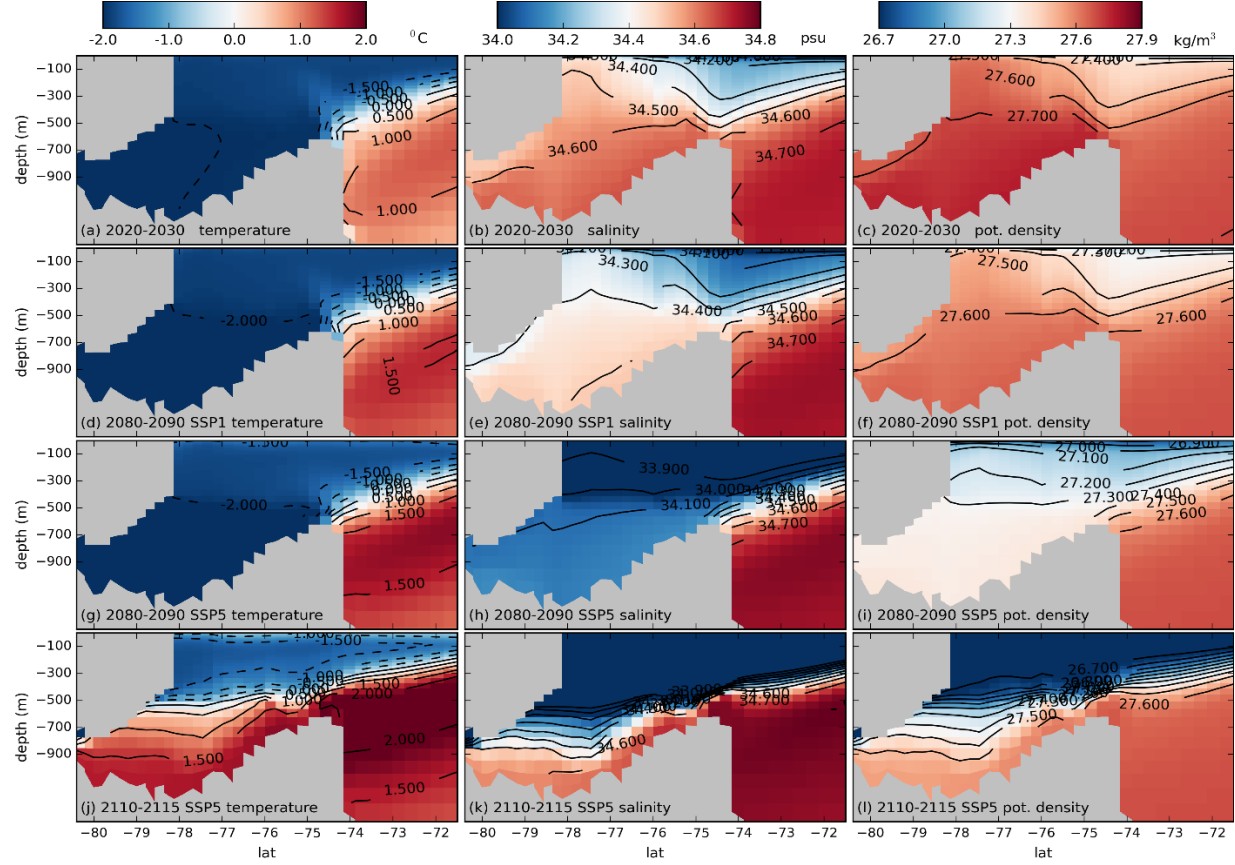

Figure 9: Temperature, salinity and potential density profile in a Filchner Trough section (indicated by the red line in Fig. 8a) for: (a,b,c) SSP5-8.5 year 2020-2030, (d,e,f) SSP1-1.9 year 2080-2090, (g,h,i) SSP5-8.5 year 2080-2090, (j,k,l) SSP5-8.5 year 2110-2115.

### 3.2.2 Ross Shelf

In the SSP5-8.5 simulation, modified Circumpolar Deep Water starts to intrude into the eastern Ross Sea continental shelf in around the year 2070 (Fig. 6e), more than two decades before the Warm Deep Water intrusion into the Filchner region. Unlike in the Filchner Trough however, we are not aware of any previous published studies detailing such pervasive warm-water incursions onto the Ross Sea continental shelf. Circumpolar Deep Water enters the Ross Ice Shelf cavity through the Little America Basin Trough (the green line section between red dots *c* and *d* in Fig. 10a). The observed water mass structure in the eastern Ross Sea is fresher than in the Filchner Trough (Thompson et al., 2018) and this feature is also found in our simulations (Figs. 9a-c & 11a-c), though the model contains fresh biases (Fig. 3d) in both regions. The overall warming mechanism is similar in the Little America Basin, though ocean conditions are slightly different.

In the early period the Little America Basin shelf break is filled with warm, saline water (Figs. 10c-e) while the ice front is mostly cold (Figs. 10g-h). As the SSP5-8.5 simulation progresses, the shelf break is subjected to cold and fresh intrusions while the ice front gradually freshens in response to a decline in sea ice production (Fig. 10b). Eventually, the density gradient

between north and south is so strong (Figs. 11d-f) that warm, saline water floods onto the continental shelf (Figs. 10f,i and
Figs. 11g-i), in the same manner as in the Filchner Trough. The impact of this warm water intrusion on the cavity geometry
over three decades is very evident (Figs. 11g-i), as the ice shelf draft is greatly reduced by the strong ocean-forced basal
melting.

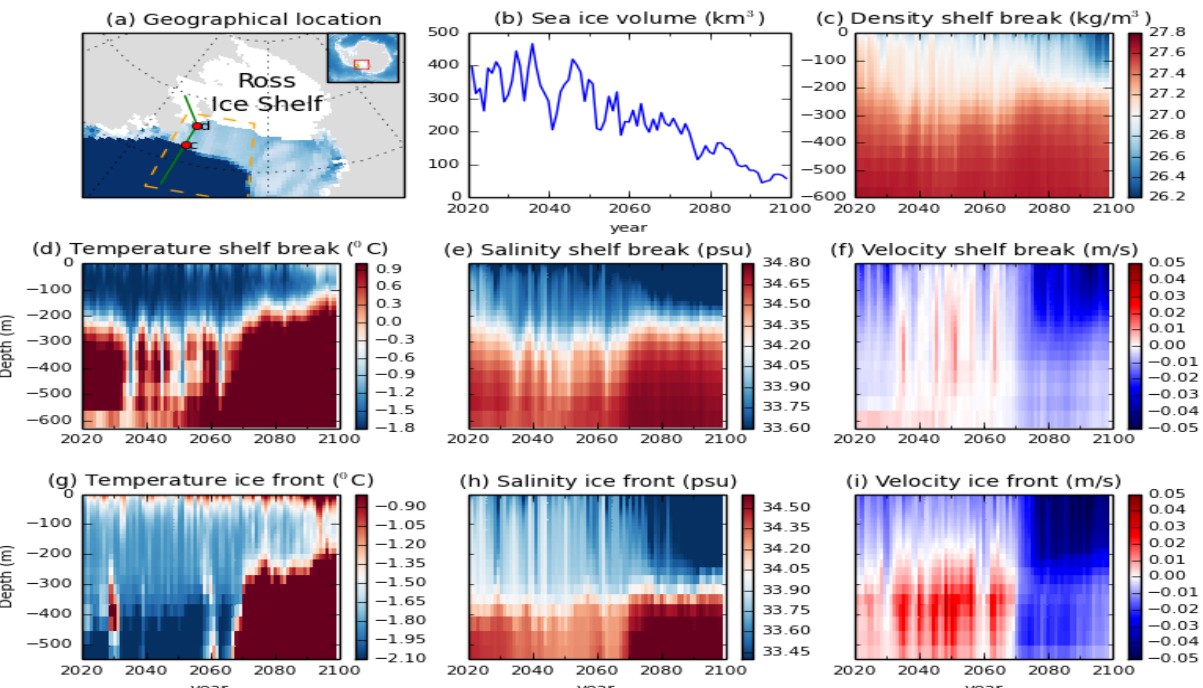

Figure 10: Little America Basin profile in the SSP5-8.5 scenario. (a) The chosen section on the Little America Basin is indicated by
the green line segment between the red dots c and d. The entire green line denotes the meridional section used in Figure 11. The
dark and light blue colours represent the deep ocean and the continental shelf in the Ross Sea, respectively, whereas the grey and
white colours represent the grounded ice sheet and the Ross Ice Shelf, respectively. (b) Sea ice volume on the area enclosed by the
orange dashed lines in (a). (c-f) The timeseries of potential density, temperature, salinity and meridional velocity, respectively at the
shelf break indicated by the red dot c in (a). (g-i) The timeseries of temperature, salinity and meridional velocity at the ice front
indicated by the red dot d in (a).

It is clear from the temperature section in Fig. 11a that early in the simulation the warm Circumpolar Deep Water intrudes onto
the continental shelf over the top of the dense cold shelf water and is then cooled from above by sea-ice formation, as is
observed in the present day (Castagno et al., 2017). After the projected warming (Fig. 11d), the warm water flows along the
seabed and becomes the densest water on shelf, and it flows straight into the Ross Ice Shelf cavity without cooling. In the
SSP1-1.9 simulation, at the end of the projection (Figs. 11j-l) the continental shelf has also freshened, however the freshening
and hence the increase in the meridional density gradient is not as strong as in the SSP5-8.5 simulation (11d-f), and so no
major warm intrusion has occurred.

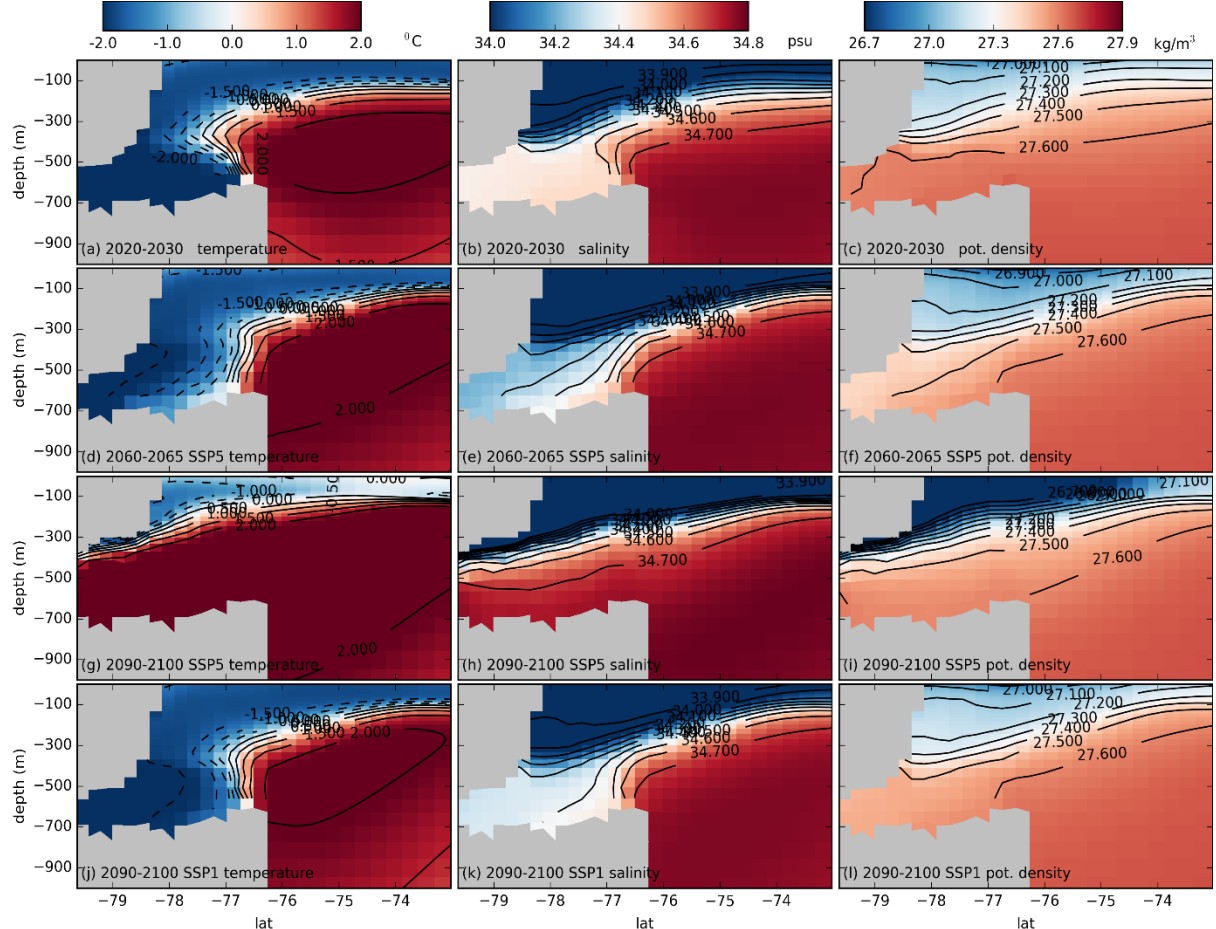

**Figure 11: Temperature, salinity and potential density profile in a Little America Basin section (indicated by the green line in Fig. 10a) for: (a,b,c) SSP5-8.5 year 2020-2030, (d,e,f) SSP5-8.5 year 2060-2065, (g,h,i) SSP5-8.5 year 2090-2100, (j,k,l) SSP1-1.9 year 2090-2100.**

It is interesting to consider why we find a warming on the shelf in the Ross Sea before the Filchner Trough, while previous studies have simulated strong warming of the Filchner Trough without any change (Hellmer et al., 2012) or with only limited warming (Timmermann and Hellmer, 2013) in the Ross Sea. We speculate that the fresh bias found in the UKESM1.0-ice simulations in the Ross Sea may pre-condition this area for the rapid change that we see. We hypothesise that if the same freshening trend were found in a model that were saltier to begin with, there may be a longer delay before the density gradient at the shelf break steepened substantially, and relatively dense (but warm) Circumpolar Deep Water was allowed to flood the shelf. Clearly, improving the initialisation of our model in this region is an important topic for future research. If such a warming were to occur in the coming centuries, the rapid melting of the Siple Coast ice streams would certainly lead to a major reconfiguration of the AIS.

### 3.2.3 Pine Island Bay

While the changing climate in these runs affects the large, cold ice shelves through intrusions of warmer waters, no increase in melt is simulated under the PIG (Fig. 7d) or Thwaites ice shelves, where the warm Circumpolar Deep Water already exists in the present-day climate. Since the ice shelves in the Amundsen Sea are widely seen as vulnerable to climate change (Paolo et al., 2015), it is perhaps surprising that we do not see a significant response here in our simulations. However, the smaller ice shelves are very poorly resolved on the ORCA1 (1° longitude) ocean grid used in this model configuration, which affects

the circulation of warm water under such shelves as well as the performance of the melt parameterisation itself.

    Since the southern ice front of the PIG Ice Shelf (black dot in Fig. 12a) is the open-ocean grid column nearest to most ice-shelf cells with relatively high melting (the southern part of the ice shelf), we examine time series at this location for both the SSP1-1.9 and SSP5-8.5 simulations (Figs. 12c-f). Up to 2060, the temperature and salinity profiles through this whole column do not show significant differences between the SSP1-1.9 (Figs. 12c,d) and SSP5-8.5 (Figs. 12e,f) simulation. The ensemble

mean integrated melt fluxes under this ice shelf are also similar in this period (Fig. 7d).

    Differences between the scenarios in the temperature and salinity responses at the ice front start to appear after 2060. The SSP5-8.5 simulation demonstrates continuous freshening in the entire water column (Fig. 12f), accompanied by a warming trend below 400 m after 2080 (Fig. 12e). Compared to the SSP1-1.9 simulation, in the SSP5-8.5 simulation there is a consistent pattern of colder and fresher waters in the layer between 50 m and 250 m depth after 2060 (Figs. 12g-h), while deep ocean

conditions are warmer.

    In order to find out if this pattern in front of the PIG Ice Shelf originates from the deep ocean, we compare it with the horizontally averaged temperature and salinity difference between the two scenarios (Figs. 12i-j) on the continental shelf (the region bounded by the green box in Fig. 12a) to the northwest of the Pine Island Bay. While the differences between the shelf and ice front are similar between 2060 and 2070, the warmer temperatures at the ice front after 2070 in the SSP5-8.5 simulation

(Fig. 12g) are confined to the deep ocean, with cooling above, in contrast to warmer waters on the continental shelf (Fig. 12i). These opposing temperature patterns are caused by ice shelf melting. In this coarse ocean model, the cavity under Pine Island Glacier is represented by only 11 grid columns (represented by the 11 grid cells with the ice shelf draft colour scale in Fig. 12b).

    In the 2070-2100 period, the ice shelf draft on 9 of those columns is either between 50m and 80m depth or between 190m and

240m depth. These are the same depths at the ice front which show persistently cold temperatures to 2100, implying that the basal melting of the ice shelf has a strong impact on water mass properties at the ice front. In this particular configuration, the averaging of ice shelf draft and melting between the 2km BISICLES grid and the ~1° NEMO horizontal grid leads to many grid columns that have ice shelf base within the same vertical ocean model level. The concentration of strong melting at this level may lead to a large cooling across the whole cavity stratified into a narrow horizontal layer.

We would not expect to accurately resolve ice shelf cavities of this size in our model given the horizontal grid of the ocean we are using. This restricts the scientific questions we are able to investigate with the model, since the behaviour of the glaciers

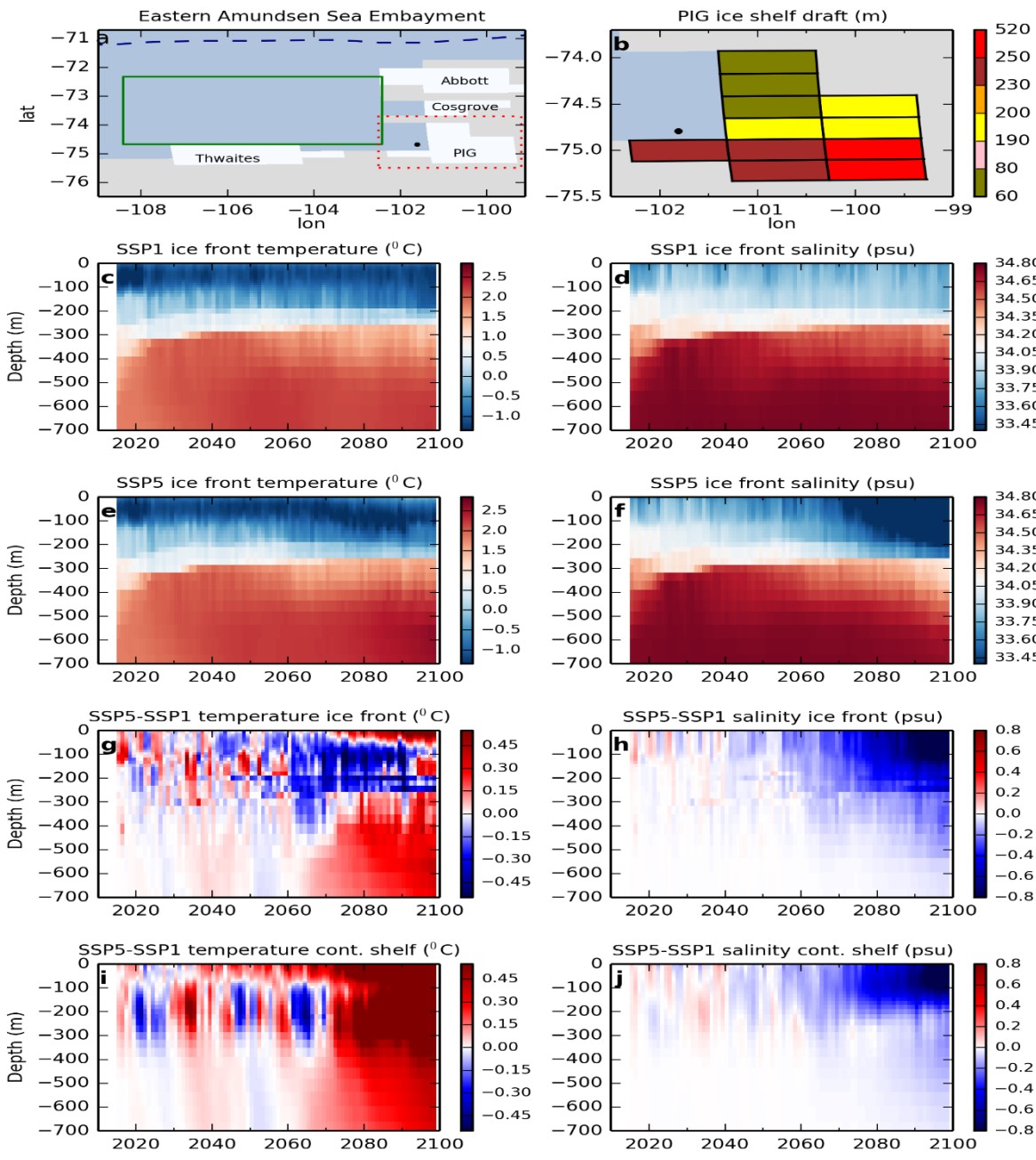

**Figure 12: (a)** The Amundsen Sea Embayment in the ORCA1 NEMO ocean model. Blue, grey, and white colour denote the ocean, grounded ice sheet, and ice shelf region. The dashed purple line marks the 700 m isobaths as the continental shelf edges. The black dot indicates the ice front where the time series profile in (c-h) is located. Enclosed by the green box is the continental shelf region where the horizontal average of time series profile in (i-j) is taken from. **(b)** The enlarged area surrounded by red dotted line in (a), but now with PIG Ice Shelf shaded with ice shelf draft colour scale. The draft is taken from the year 2080 of one SSP5-8.5 ensemble member. **(c-h)** Time-series at the PIG ice front of temperature and salinity: (c-d) SSP1-1.9 ensemble mean; (e-f) SSP5-8.5 ensemble mean; (g-h) anomaly between SSP5-8.5 and SSP1-1.9 ensemble mean. **(i-j)** like (g-h) but for the horizontal average over the area bounded by the green box in (a).

that drain into the Amundsen Sea are crucial to the long-term stability of the West Antarctic Ice Sheet (Pritchard et al., 2012). Improving the performance of our melting parameterisation at low grid resolutions, and modelling the ocean at higher resolution within our framework, are two foci of our ongoing research.

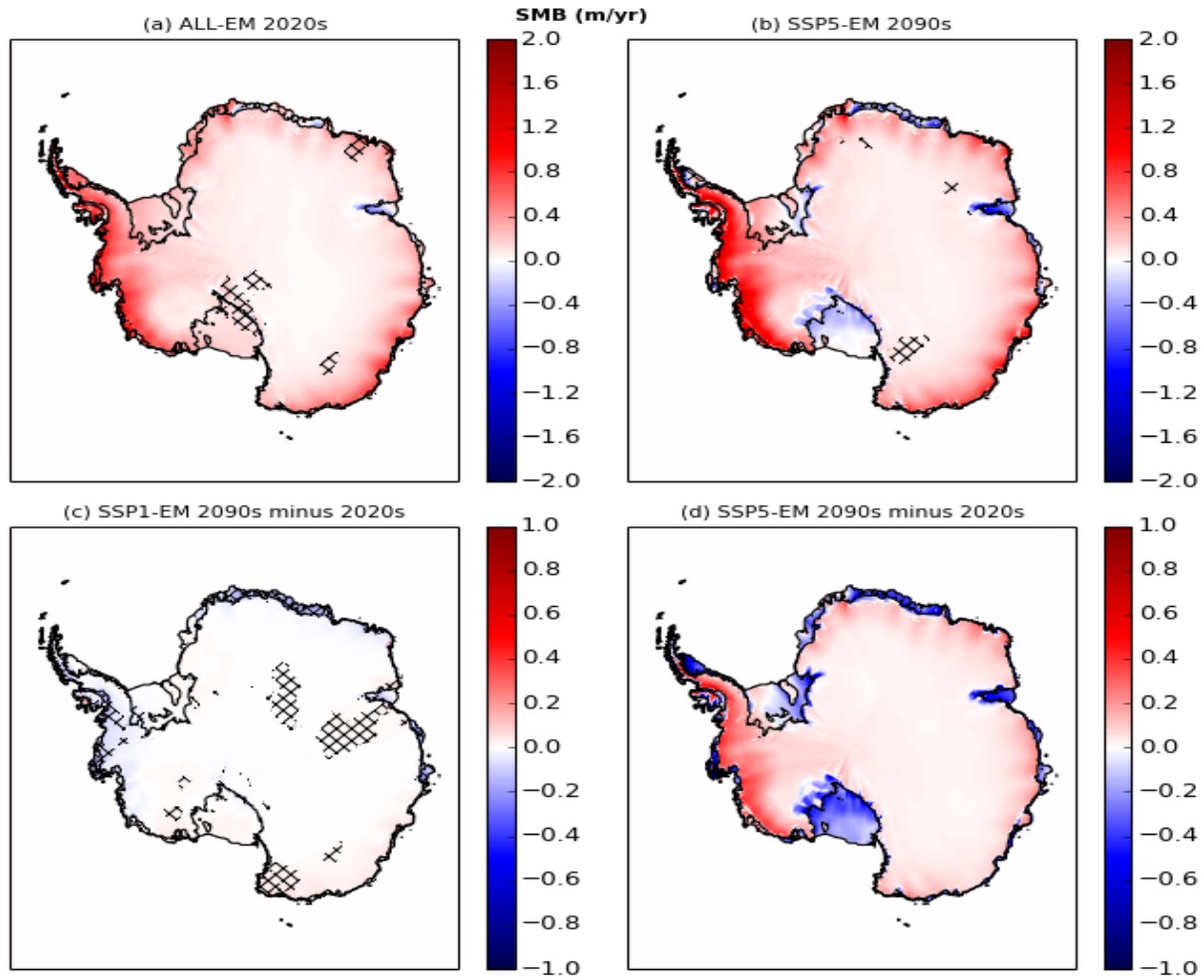


**Figure 13: (a) ALL-EM SMB for period 2020-2030. Locations with statistically significant difference (95% Student's t- confidence interval) between SSP1-EM and SSP5-EM are hatched. (b) SSP5-EM SMB 2090-2100. Locations with statistically significant difference (95% confidence interval with one-way ANOVA) among the ensemble members are hatched. (c) Difference of SSP1-EM SMB between 2090-2100 period and 2020-2030 period. Locations where the difference is statistically significant (95% Student's t- confidence interval) are hatched. (d) Difference of SSP1-EM SMB between 2090-2100 period and 2020-2030 period, almost everywhere is statistically significant. Black and grey lines indicate the ice sheet grounding lines and ice shelf fronts, respectively.**


### 3.3 Projections of Surface Mass Balance

As well as ice shelf basal melting, the other climatic influence on an ice sheet is through the SMB. In the present day the majority of Antarctica is too cold to experience substantial surface melting, but this is expected to change for some climate

warming scenarios (Kittell et al., 2021). Decadal-average differences in SMB between the SSP1-EM and SSP5-EM in general are minor in the early part of our simulations, and in this period statistically significant differences between the two scenarios are found only in a few places such as around the southern tip of the Ross Ice Shelf and a number of ice shelves in the Dronning Maud Land region (Fig. 13a). Likewise, differences in the total area-integrated SMB over this period (Table 2) between the two scenarios are not significant on the grounded area or on the ice shelves.

**Table 2: Ensemble mean, decadal mean area-integrated SMB (Gt yr$^{-1}$) on ice shelves and grounded ice from SSP1-1.9 and SSP5-8.5 simulations for 2020-2030 and 2090-2100. The standard deviation indicates the ensemble spread.**

| SMB | 2020-2030 | | 2090-2100 | |
|---|---|---|---|---|
| | SSP1-1.9 | SSP5-8.5 | SSP1-1.9 | SSP5-8.5 |
| Ice shelves | $335 \pm 6$ | $321 \pm 8$ | $305 \pm 11$ | $-52 \pm 48$ |
| Grounded ice | $1743 \pm 25$ | $1741 \pm 29$ | $1747 \pm 11$ | $2241 \pm 35$ |
| Total | $2078 \pm 31$ | $2062 \pm 31$ | $2052 \pm 14$ | $2188 \pm 83$ |

Since continental-scale observations of Antarctic SMB are not yet available, we evaluate our simulations against data from Arthern et al. (2006), which were interpolated from scarce Antarctic SMB observations. Our ALL-EM of SMB averaged over 650 the 2020-2030 period has a similar pattern and magnitude with those data (Fig. A2 of Appendix A) aside from the Amery Ice Shelf where we simulate negative SMB (Fig. 13a). The highest accumulations are generally simulated in West Antarctica, including the Amundsen and Bellingshausen sectors and the Antarctic Peninsula, with the exception being the interior of Marie Byrd Land, where it is close to zero. Relatively high magnitudes are also found on the periphery of East Antarctica. Compared with regional climate model hindcast simulations reported in Mottram et al. (2021), our total area-integrated SMB over the 655 2020-2030 period (Table 2) lies on the lower end of the intercomparison range and is close to the ERA-Interim reanalysis.

The last decade of the simulations shows a divergence in response to the forcing scenario. In the SSP1-1.9 simulations, statistically significant SMB changes from the 2020s to the 2090s are sparse (hatched area in Fig. 13c), with the most significant differences around the Eastern Amundsen and Bellinghausen sectors with an average SMB decrease of 0.057 m/yr. We also simulate a negligible difference in total integrated SMB between the two periods (Table 2) for both ice shelves and 660 grounded area for the SSP1-1.9 simulations.

In the higher forcing SSP5-8.5 runs, the total integrated SMB over the AIS increases from $2062 \pm 31$ Gt/yr in the 2020-2030 period to $2188 \pm 83$ Gt/yr in the 2090-2100 period (Table 2). The magnitude of change in this ice sheet-wide integrated value however does not reflect the scale of regional changes that occur. There are large changes on the grounded ice sheet and ice shelves, with opposing signs (Table 2). SMB generally increases on the grounded ice whilst most ice shelves see large 665 decreases (Fig. 13d). Decreases of more than 0.6 m/yr are found on many ice shelves across the continent. These large decreases result in negative SMB on many ice shelves such as the Ross and Filchner ice shelves, and the majority of ice shelves in Queen Maud Land (Fig. 13b), and there is a negative total integrated SMB over all ice shelves (Table 2).

Figure 14 illustrates the most significant SMB changes in a representative member of the SSP5-8.5 ensemble. The large negative changes on ice shelves are dominated by increases in surface melting and runoff (Figs. 14b-c) generated by rising temperatures at low-lying elevations. Although this negative SMB does not directly affect sea level, given its location on the floating part of the ice sheet, the future stability of the ice sheet may be affected by the dynamic influence of reduced ice shelf thickness or widespread hydrofracturing (Trusel et al., 2015; DeConto and Pollard, 2016) that could be triggered by large amounts of melt on ice shelves.


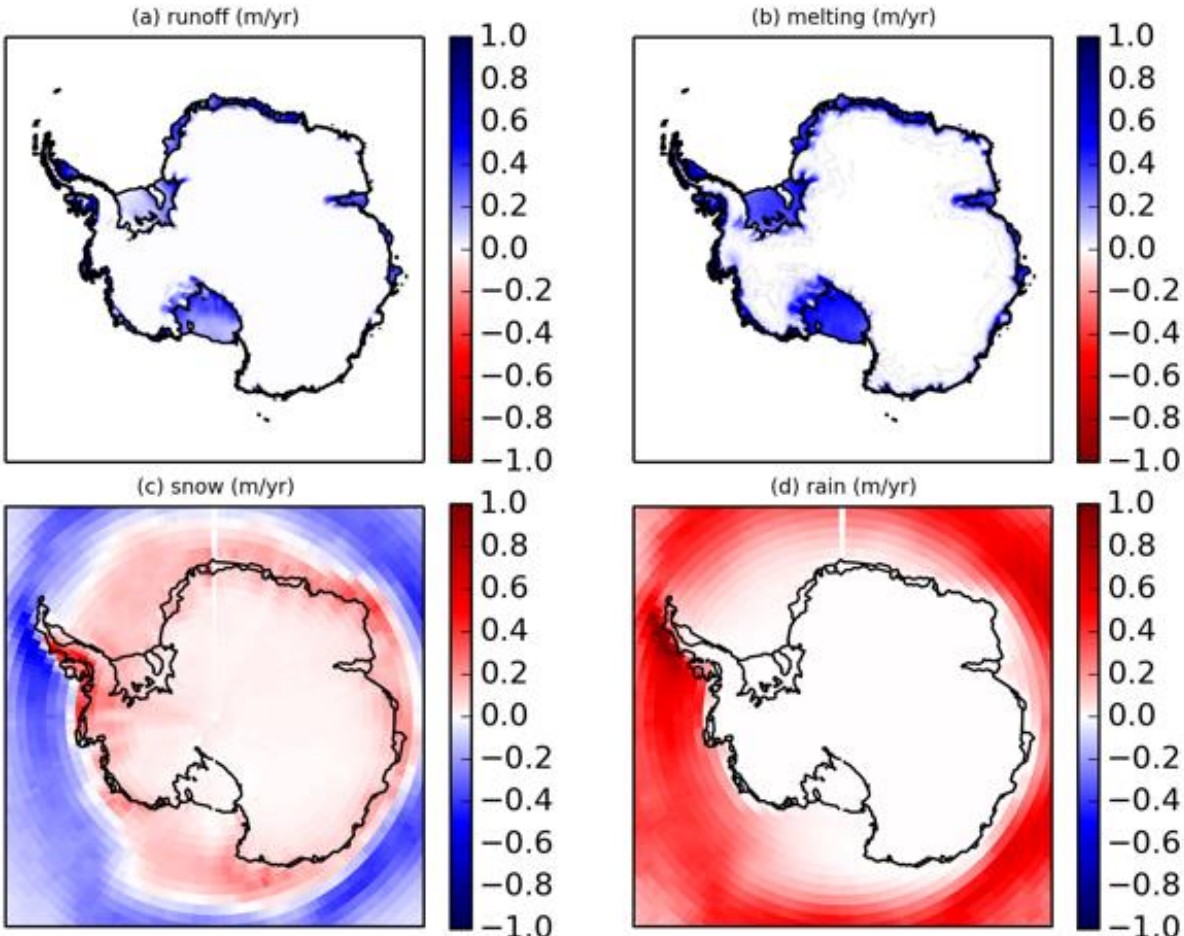


**Figure 14: The difference between average 2090-2100 and average 2020-2030 of SMB components in SSP5-8.5: (a) surface runoff (b) surface melting (c) snowfall and (d) rainfall, all in m/yr. Black lines indicate the grounding lines of the ice sheet.**

On the grounded ice sheet, the area-integrated SMB is projected to increase by almost 30% between the 2020s and the 2090s (Table 2). Significant increases in snowfall are simulated on the periphery of East Antarctica and on most of the West Antarctic Ice Sheet (Fig. 14c). These are also the areas with the highest SMB over the 2020-2030 period. Changes in SMB on the grounded ice in all our simulations are caused by increases in snowfall (Fig. 14c), with rain, surface melting and runoff on


grounded ice remaining rare in general (Figs. 14a,b,d). An increase in precipitation over Antarctica is expected to occur through a number of mechanisms as the climate warms (Dalaiden et al., 2020) and this pattern of SMB decrease on ice shelves and increase on grounded ice is a common feature of climate model simulations (e.g. Kittel et al., 2021).

## 3.4 Projections of ice sheet evolution

This section covers the impact of the basal and surface forcing on the thickness and dynamics of the AIS. Particular focus is given to changes in the volume of ice above flotation which contributes to sea level rise.

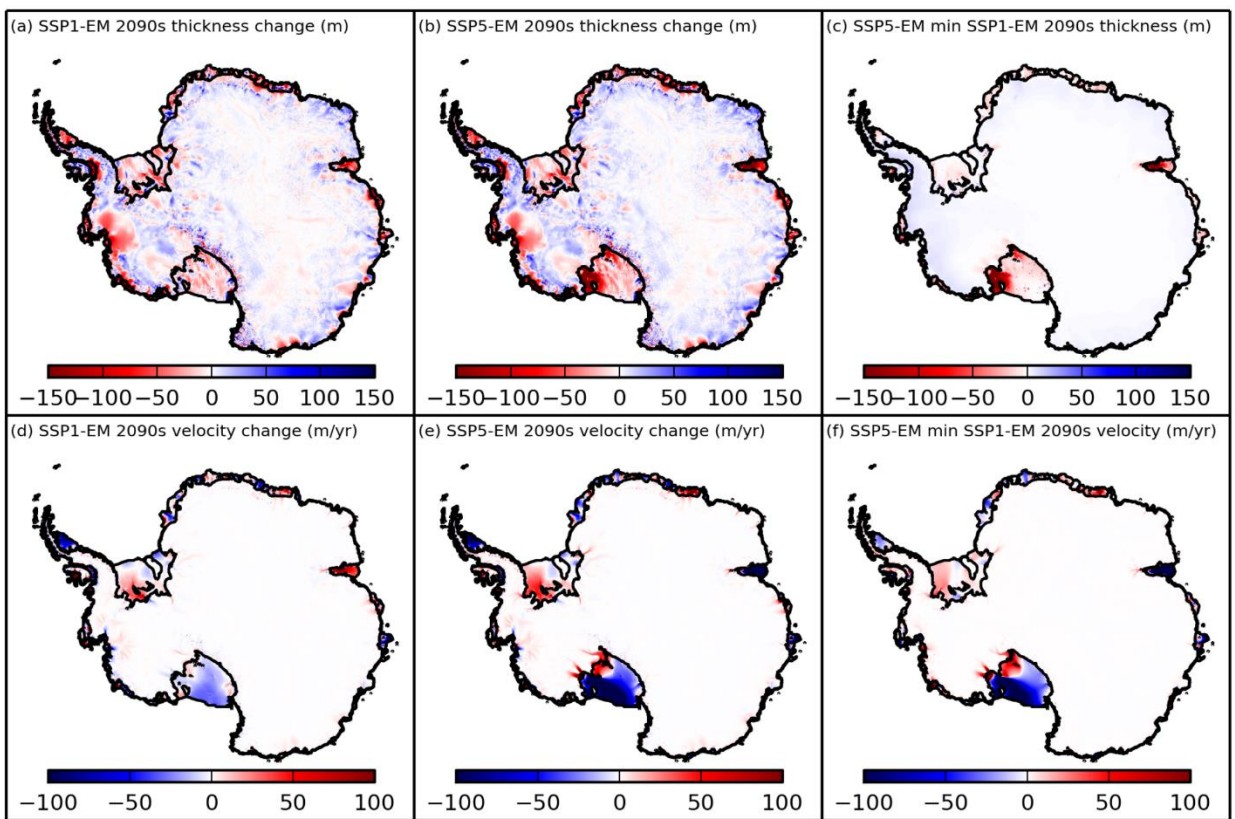

**Figure 15: Change of (top) ice thickness and (bottom) surface velocity magnitude between (Left) 2015 and 2100 in SSP1-1.9 (Middle) 2015 and 2100 in SSP5-8.5. (Right) Difference of (top) ice thickness and (bottom) surface velocity magnitude in the final period between SSP5-8.5 and SSP1-1.9.**

Figures 15a-b show the SSP1-EM and SSP5-EM of simulated ice thickness change during the simulations. Most Antarctic ice shelves thin in both scenarios by the end of the century. In the Amundsen sector, the thinning spreads upstream from Pine Island Glacier and Thwaites Glacier ice shelves and it is slightly less (Fig. 15c) in SSP5-EM than SSP1-EM. This follows from the increase in SMB, which dominates under the higher radiative forcing scenario. However, we do not consider our projections in these two relatively small regions to be very robust given the low resolution of the ocean cavities underneath the ice shelves. The differences between the two scenario ensembles are more obvious for the eastern Ross and Amery ice shelves where

thickness anomalies are significantly larger in the SSP5-8.5 scenario and have also propagated to the ice streams feeding the shelves. In most other regions, the spatial pattern of thickness change is nearly the same for both scenarios (Figs. 15a-b) and

the speckly, low magnitude pattern of thinning and thickening disappears when the difference is taken between the pairs of scenario simulations to highlight the climatically-forced signals (Fig. 15c). In the Filchner Ice Shelf, there is no visible difference in the magnitude of thickness change. This is as expected, since the strong melting under the ice shelf in the SSP5-8.5 scenario has only just started around 2100. Ice in the SSP5-8.5 simulations is slightly thicker than in the SSP1-1.9 simulations on the grounded ice sheet, up to about 12 m thicker in the West Antarctic and up to about 8 m thicker in the East

Antarctic near the coasts. In most sectors across the ice sheet, on the timescales of our simulations the thickness changes predominantly reflect the local changes in SMB and basal melting discussed in previous sections.

There is a spatial heterogeneity in simulated surface velocity changes across the AIS. The noticeable changes (Figs. 15d-f) mostly occur on the shelves, where reductions in speed are associated with the thinning of the ice. Some exceptional cases of acceleration are the Ronne Ice Shelf (in both scenario ensembles) and Amery Ice Shelf (in the SSP1-1.9 ensemble). Pine Island

and Thwaites glaciers show small accelerations in both ensembles. The biggest difference between ensembles is the acceleration of ice streams along the Siple Coast in the SSP5-8.5 simulations.

Figures 16a-c show the evolution of the simulated cumulative ice mass above flotation (MAF) anomaly for all the ensemble members. The total MAF loss (Fig. 16a) is higher in the SSP5-8.5 simulations than in the SSP1-1.9 simulations until the mid-2040s, after which the SSP5-8.5 simulations show an increasingly positive MAF trend. In the year 2100, the SSP5-EM

cumulative total MAF reaches a positive anomaly of 7715 ± 1856 Gt (21.2 ± 5.1 mm GMSL fall equivalent), with the lowest and highest among the members being 5146 and 10370 Gt respectively. The SSP1-EM cumulative total MAF decreases during the entire simulation period, although at a slowing rate. In the year 2100, it reaches a negative mass anomaly of 8069 ± 1026 Gt (22.2 ± 2.8 mm GMSL rise equivalent). The overall SSP1-EM of total MAF loss trend in the entire period is 73 Gt/yr, which is within the present day observed range (IMBIE Team, 2018).

In East Antarctica, simulations from both ensembles experience an increase in MAF from the 2040s until 2100 (Fig. 16b). They differ, however, in the rate of increase: SSP5-EM has an accelerating gain in mass with an average mass gain rate of 207 Gt/yr, whereas the increase in the SSP1-EM remains steady at about 35 Gt/yr.

In West Antarctica (including the Antarctic Peninsula), the loss of MAF remains relatively steady in SSP1-1.9 over the entire period (Fig. 16c) and it exceeds the MAF gain in East Antarctica (Fig. 16b), leading to an overall mass loss for the ice sheet

(Fig. 16a). In the SSP5-8.5 simulations, the decreasing mass trend in West Antarctica reduces after the 2040s and changes sign in the 2060s, making the ensemble mean of West Antarctica MAF in the year 2100 around 1800 Gt lower than its initial value (Fig. 16c). For the SSP5-8.5 simulations, the 2040s is the turning point where acceleration of mass gain occurs in both West and East Antarctica regions (Figs. 16b-c). In the Ross sector, although its MAF is increasing, a significant portion of the eastern Ross Ice Shelf has lost more than 50% of the thickness by 2100, due to strong melting at both the top and bottom boundaries.

Both SSP1-EM and SSP5-EM follow a similar pattern in the total ice discharge across the grounding lines (Fig. 16d) until the end of the 2060s, with a decrease in the rate of discharge starting in the 2040s. The trend toward decreasing discharge remains

in the SSP1-1.9 simulations until 2100 but this is not the case with the SSP5-8.5 simulations. In these simulations there is an increase in discharge after 2080 (Figs. 16d,f) due to the high basal melt rate of the eastern Ross Ice Shelf which starts a decade earlier. Accelerating ice discharge from the Pine Island and Thwaites glaciers has been found in present day observations

(Rignot et al., 2020) and is the main contributor to current WAIS ice mass loss, but there is little indication of this in our simulations. Like our ocean model circulation in the Amundsen region which needs improvement, the development of Amundsen glaciers modelling is another focus of our ongoing research.

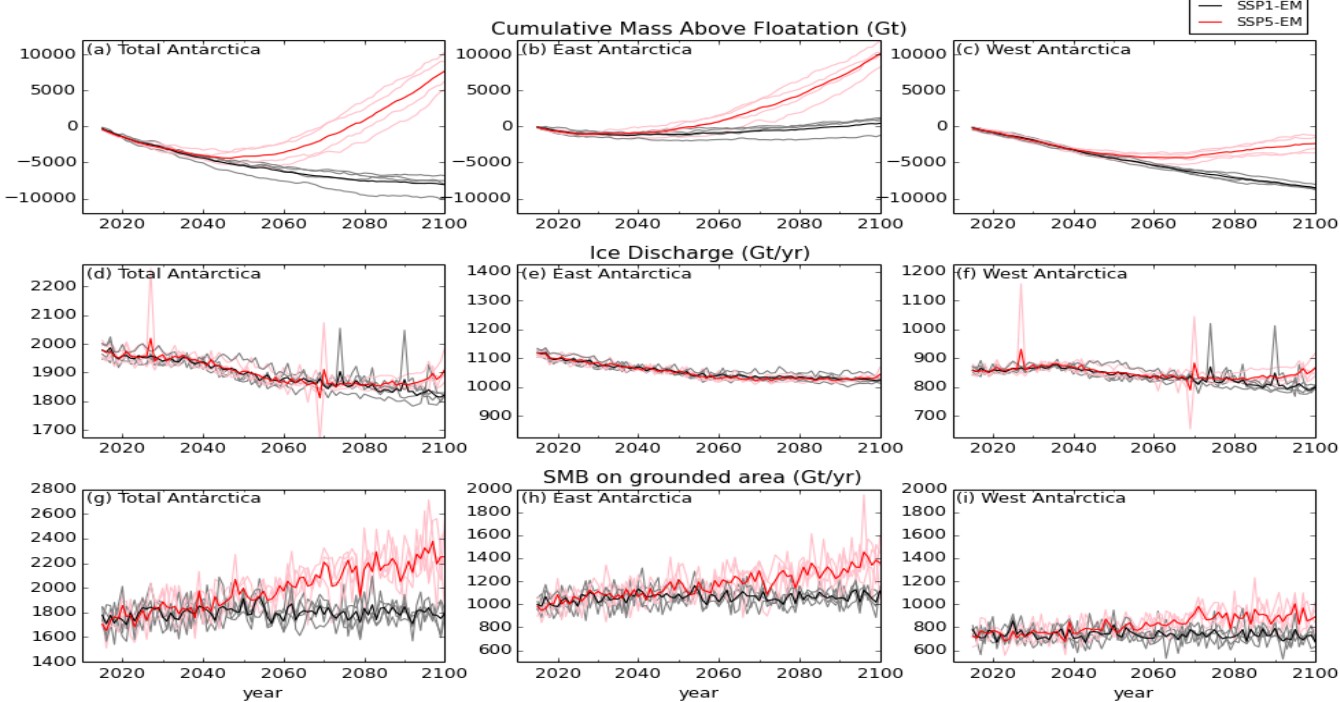

Figure 16: Time-series of ice mass budget. Top row: cumulative anomaly of ice mass above flotation relative to the initial condition;
middle: rate of discharge across grounding lines; bottom row: SMB rate on the grounded ice. The grey, black, pink and red lines represent the SSP1-1.9 ensemble members, SSP1-1.9 ensemble mean, SSP5-8.5 ensemble members and SSP5-8.5 ensemble mean respectively. The Antarctic Peninsula region is considered part of West Antarctica. In each row (budget), the axis range is set the same for all columns (regions).

In the SSP1-1.9 simulations, the area-integrated SMB over the entire grounded ice sheet remains stable at around 1745 Gt/yr
over the course of simulation (Figs. 16g and Table 2), so the MAF loss trend is largely controlled by the rate of ice discharge (Figs. 16d-f). On the other hand, the scale of ice discharge in the SSP5-8.5 simulations is outweighed by the SMB over the grounded area after the 2050s. The accelerating SMB from the 2040s until the end of the simulations (Figs. 16g-i) dominates the grounded ice mass budget in this strong forcing scenario which then results in the increasingly positive MAF trend.

Despite the increasing MAF in the SSP5-8.5 simulations, ice shelves with strong melt rates still show a pattern of grounding
line retreat. Figure 17 illustrates the simulated grounding line changes for major ice shelves with relatively strong basal melt rates. In the Amundsen region, there is no substantial difference in total ice shelf area between the SSP1-1.9 and SSP5-8.5

simulations, and both ensembles display similar patterns of grounding line migration, where the grounding line of Thwaites Glacier retreats southward by up to 40 km and that of Pine Island Glacier southward and westward by a similar distance (Fig. 17a).

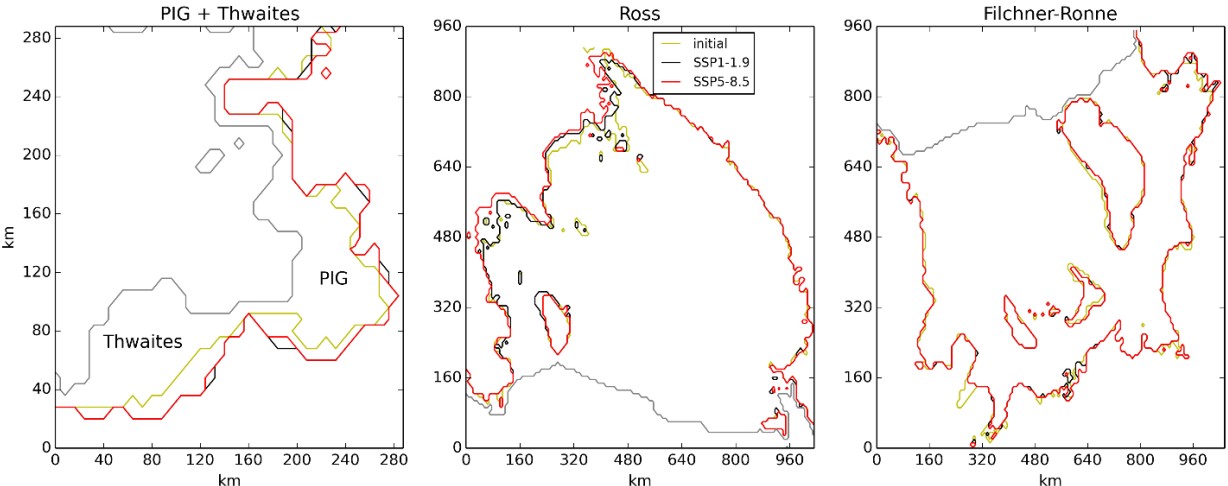

**Figure 17: Grounding line change in (Left) Pine Island & Thwaites Glaciers (Middle) Ross and (Right) Filchner-Ronne ice shelves. Gray=ice front, yellow=initial grounding line, black=grounding line for SSP1-1.9 in 2100, red=grounding line for SSP5-8.5 in 2100.**

Beneath the Filchner-Ronne Ice Shelf, the two ensembles also show similar patterns of retreat and advance in many areas. The main difference is a 30-40 km retreat in northeast Filchner for the SSP5-8.5 simulations following the strong melting which starts in the last decade of the run (Fig 17c). The largest change of grounding line is simulated under the Ross Ice Shelf in the SSP5-8.5 simulations (Fig. 17b), with up to 70 km retreat in the eastern region and 90km in the southern part. The Ross Ice Shelf also ungrounds from several pinning points in its eastern sector in the SSP5-8.5 simulations. Given these impacts from 30 years of strong melting under the Ross Ice Shelf, longer simulations may reveal large grounding line migrations under the Filchner-Ronne Ice Shelf.

# 4 Discussion

## 4.1 Summary

This paper presents 21$^{st}$ century simulations of the Antarctic Ice Sheet (AIS) mass budget using the UKESM1.0-ice Earth System model, which includes an interactive ice sheet component coupled to the ocean circulation under the ice shelves and the snowpack mass balance at the surface. Four-member initial-condition ensembles of future simulations are run under one low (SSP1-1.9) and one extreme (SSP5-8.5) greenhouse gas emission scenario. All our simulations have proven to be computationally stable over the 21$^{st}$ century and show that this model has the capability to assess the future evolution of the AIS in a global ESM that contains direct ice-climate feedbacks between component models.

For each emission scenario, the simulations show similar ice sheet and basal melting responses across the ensemble members regardless of the initial climate state. In the SSP1-1.9 simulations, the total ice shelf basal melting does not show a significant trend and no major changes are simulated in regional climate or ice sheet behaviour. In this scenario the mass of ice above flotation on Antarctica decreases more slowly than is currently observed (IMBIE Team, 2018; Gardner et al., 2018; Rignot et al., 2019), and therefore our model simulates less future GMSL rise for this scenario than other projections (Oppenheimer et al., 2019; Fox-Kemper et al., 2021).

The strongest responses to the forcing from the SSP5-8.5 simulations take place under the Ross and Filchner ice shelves, where sustained warm water intrusions under the ice begin around 2070 and 2100 respectively, driven by freshening on the continental shelf and slope. These intrusions lead to greatly increased basal melt rates under the ice shelves, and in our simulations the grounding line of the eastern Ross Ice Shelf retreats significantly. On the other hand, we simulate only limited changes under ice shelves in the Amundsen Sea, where melting under the Pine Island Glacier and Thwaites ice shelves is no higher than in the SSP1-1.9 simulations. These results may indicate the large potential that Ross/Weddell sectors have in becoming major sea level contributors in future warming scenarios.

The surface mass balance (SMB) of the grounded ice sheet and floating ice shelves show opposing trends in the SSP5-8.5 simulations. A strong increase in SMB on the grounded ice sheet follows the increase in atmospheric greenhouse gas concentrations, with a rapid increase in snowfall and negligible surface melting. On the ice shelves, a large increase in surface melting and runoff dominates the SMB trend, leading to net loss of ice mass from the surface. Ice shelf thinning is particularly significant near the grounding line of the eastern Ross Ice Shelf due to the large amount of both surface and basal melting; in this region we simulate a 50% reduction of the initial ice thickness by 2100.

Despite significant surface warming and strong melting beneath the major ice shelves, our SSP5-8.5 simulations do not produce a significant sea level rise contribution from Antarctica by 2100. This is due to the large increase in snowfall over the grounded ice sheet, which outweighs changes in the discharge of ice across the grounding line. The former dominates because it is a rapid response to climate change, while the latter takes longer to play out. Since ice sheets respond to changes in climate on centennial timescales, our simulations of the 21$^{st}$ century do not capture the full potential impact of the changes triggered in the Antarctic Ice Sheet.

Given the inherent biases in our component models and our rather crude method for initialising coupled climate-ice sheet states, the strength of these model simulations lies in their consistent, coupled evolution of elements of the Earth System that are usually considered separately. This model enables us, for the first time, to study a range of physical interactions between the AIS and a state-of-the-art CMIP6-class climate model. For example, in the Ross Sea, we simulate ocean freshening and warming that is fully coupled to ice shelf melting, thinning, and grounding-line retreat. This also helps us identify a potential climate-driven change in an AIS sector which is not considered a threat from the point of view of modern-day observations. Looking beyond the 21st century, feedbacks between the ice state and the climate would be expected to be even more pronounced, and this sort of model is uniquely capable of simulating climate and ice sheet mass loss contributions to sea level rise that are fully consistent with the climate forcing scenario.

## 4.2 Mean state of projections

Atmospheric greenhouse gas concentrations in the SSP1-1.9 scenario, whilst low, are still above present-day concentrations and therefore one might expect to simulate rates of change of the AIS that are higher than those currently observed (e.g. IMBIE Team, 2018). In fact, both our initialised modern state and our SSP1-1.9 future simulations appear rather more stable than the AIS is observed to be now. Although this brings into question the direct use of these simulations as projections, it may be seen as beneficial in our context. This stability gives us confidence that our coupled climate-ice system is not computationally unstable, and that flaws in our initialisation technique do not lead to simulations with an unacceptable level of numerical drift away from a realistic climate state. We can thus use the SSP1-1.9 projections as a baseline to compare the SSP5-8.5 simulations against, interpreting the differences in behaviour of the climate and ice sheet between those simulations as genuine forced responses of the system. In our simulations, we have identified some clear responses to differences between the SSP5-8.5 and SSP1-1.9 forcing, such as the large increases in basal melting, the diverging SMB responses over ice shelves and grounded ice sheet, and the grounding line retreat of the Ross Ice Shelf.

Although the SSP5-8.5 simulations in UKESM experience a substantial amount of atmosphere and ocean warming and the Ross and Filchner ice shelves transition to become warm water cavities with large amounts of sub-shelf melting, all our SSP5-8.5 simulations produce a negative contribution to GMSL from the AIS by 2100. This contribution would represent a modest offset to the overall global sea level rise which is expected to be dominated by positive contributions from ocean heating and ice mass loss from glaciers and the Greenland ice sheet. Our projected negative contribution from the AIS is outside the likely range assessed in the Sixth Assessment Report of the Intergovernmental Panel on Climate Change (Fox-Kemper et al., 2021), and may seem counterintuitive. However, under an extreme radiative forcing a significant 21[st] century negative GMSL contribution from the AIS is not implausible given the dominance of snow accumulation (a rapid response to the forcing in the atmosphere) on decadal timescales, whilst typical ice sheet velocities mean that discharge increases at a much slower rate. Taken individually, the individual components of the AIS mass balance - SMB, BMB and ice discharge - in our simulations are plausible when compared to a range of estimates in the literature.

Our SSP5-8.5 simulations show a large increase in snowfall, and thus SMB, integrated over the grounded ice sheet during the 21st century. An increase in precipitation would be expected in a scenario with high radiative forcing (Lenaerts et al., 2019) given the positive correlation between the atmospheric temperature and specific humidity. Since even in the SSP5-8.5 scenario the majority of the ice sheet remains too cold for significant surface ablation, it is this increase in accumulation that dominates the SMB contribution to the mass balance. We do simulate significant surface ablation and net negative SMB near the coast due to surface warming, but these areas are mostly floating ice shelves and this mass loss makes no direct contribution to sea level rise. Although UKESM1.0 is one of the most sensitive CMIP6 models in terms of the global mean surface temperature response to greenhouse gas concentrations (Senior et al., 2020), we find that the total area-integrated AIS SMB from UKESM1.0 lies in the middle of the range projected by CMIP6 models (Gorte et al., 2020). This part of our AIS mass balance budget is thus not unusual. The SMB simulated by UKESM1.0 is also not extreme in terms of the range of AIS SMB forcings

used for stand-alone ice sheet models by ISMIP6 (Seroussi et al., 2020), whose multi-model ensemble projections of GMSL have a much wider, yet still physically plausible, range than that published in IPCC reports (Oppenheimer et al., 2019; Fox-Kemper et al., 2021).

Our simulations produce a strong basal melting response to the SSP5-8.5 scenario forcing under the large Filchner and Ross ice shelves, which have low melting under present day conditions (Rignot et al., 2013; Adusumilli et al., 2020) or in our SSP1-

1.9 simulations. While warm water intrusions under the Filchner Ice Shelf have been simulated in projections for high anthropogenic forcing scenarios by previous studies (Hellmer et al., 2012; Timmermann and Hellmer, 2013; Hellmer et al., 2017; Naughten et al., 2021), we are not aware of projections where strong melting is initiated under the Ross Ice Shelf for a similar forcing intensity. The mechanism leading to strong melting of the Ross Ice Shelf in our simulations is, however, plausible, and similar to that already proposed for the Filchner Ice Shelf, with warm water intrusions onto the continental shelf

being preceded by progressive freshening and reductions in sea ice (e.g. Timmermann and Hellmer, 2013; Hellmer et al., 2017; Naughten et al., 2021). An observational record from Jacobs et al. (2022) indicates a persistent Ross Sea freshening in the last 60 years while the study by Timmermann and Hellmer (2013) suggests a future freshening in this region. Biases in our initialised ice shelf-ocean state mean that our simulations do start with overly fresh conditions under both shelves, so we do not consider the exact timing of the incursions we simulate to be reliable. However, under the sustained freshening trends we

simulate for both shelves we would expect warm water intrusions to be triggered at some point in the late 21st or early 22nd century, regardless of the bias in our initial state. Likewise, although the 30-year time difference between the strong melting events in Ross and Filchner in our simulations may not be robust, recent work indicates a stronger warming trend on the Ross continental shelf than in the Weddell continental shelf towards end of the 21st century in the CMIP6 multi-model ensemble (Purich and England, 2021), suggesting that the phasing of these events is plausible. Similar to our SSP5-8.5 runs, highly

increasing basal melt pattern also started around the year 2070 in ISMIP6 RCP8.5 experiments with standardised basal melt parameterisation regardless of their ocean sensitivity parameter values. Even compared to their simulations with the lowest sensitivity parameter value, the trend of mean basal melt over the 2070-2100 period in our SSP5-8.5 ensemble is still lower.

### 4.3 Limitations of our approach and uncertainty of projections

In the Amundsen sector, the basal melting of the Pine Island and Thwaites ice shelves remains within the currently observed

range (Rignot et al., 2013; Depoorter et al., 2013) in all our simulations, regardless of the forcing scenario, and the ice shelf grounding lines in the SSP5-8.5 scenario do not retreat relative to those in the SSP1-1.9 scenario. We do not consider this to be a reliable projection of future behaviour in this sector. Firstly, the horizontal resolution of the eORCA1 ocean is not sufficient to resolve the ocean circulation near the coast or under these small ice shelves. Secondly, the combination of our melt parameterisation and the vertical discretisation of the geometry of the ice shelf cavity in our z* coordinate model may be

very poorly resolving the dynamics of the fresh meltwater next to the base of these small ice shelves.

Our simulated ice discharge for the 21st century does not increase beyond what is currently observed (Gardner et al., 2018; Rignot et al., 2019). On the contrary, the SSP1-1.9 simulations show a slight negative discharge trend. In the SSP5-8.5 runs,

ice discharge across the grounding line of the eastern Ross Ice Shelf increases towards the end of the century, as ice streams along the Siple Coast accelerate in response to the strong basal melting of the shelf that began a decade before. The long response timescales of ice sheet drawdown means that this slow reaction is not surprising, and we would expect to produce higher discharge from the Ross and Weddell basins in response to the SMB and BMB changes if these simulations were continued. Similarly, we do not simulate unstable grounding line retreat of any ice shelves on 21st century timescales, although the eastern Ross Ice Shelf does unground from a number of pinning points by the end of the SSP5-8.5 simulations.

The shortcomings of our ocean model in the Amundsen sector may be a particular drawback in this context, since this is the region considered most vulnerable to unstable shelf retreat in the near future (Joughin and Alley, 2011). However these shortcomings may have been mitigated in our simulations by the continuous presence of strong (although not increasing) melting under the warm ice shelves and consistent (although not large) thinning and acceleration upstream of the grounding lines. Also, the 2 km highest level of mesh refinement we allowed for BISICLES in these simulations may not be sufficient to accurately model the grounding line dynamics in this region (Cornford et al., 2016), although testing suggests that increasing the allowed refinement of the BISICLES to 500 m would not significantly alter our model evolution of the next few decades. Therefore one key point which should be addressed for a future model improvement is whether or not our ice sheet component should have triggered an instability of the Amundsen sector under the continuously strong basal melting throughout the 21$^{st}$ century.

Simulating SMB on the AIS at the spatial scales usually considered by global ESMs is challenging due to the roles that synoptic-scale events and processes like katabatic winds play at the surface. The subgrid-scale elevation tile downscaling used to improve SMB modelling in UKESM1.0 does not function as effectively on Antarctica as on Greenland, since SMB processes are not simply correlated with altitude on the AIS, so it is likely that the use of higher resolution in the atmosphere model will be necessary to make significant improvement to the explicit modelling of AIS SMB in UKESM1.0. The increase in surface melting on the ice shelves expected with future conditions (e.g. Kittel et al., 2021) would be expected to lead both to significantly lower albedo through the formation of melt ponds, and also possible shelf hydrofracture and collapse (DeConto and Pollard, 2016; Lai et al., 2020). These mechanisms are not yet represented in our model and are a focus for future development work as we look to extend our simulations and analysis beyond the 21st century, when ice shelf collapse becomes a more likely prospect.

Our projections of 21st century AIS mass balance do not show a wide range in either the timing or the magnitude of changes we simulate across the members of each ensemble. On these timescales, we expected state uncertainty due to multi-decadal ocean internal variability to be an important factor to sample, and drew our initial conditions from across the phase space of Southern Ocean variability simulated in the larger UKESM1.0 historical ensemble. However, we simulate rather small variations in the timing of the onset of the warm water intrusions across the SSP5-8.5 ensemble members, i.e. variation of 10 years for the Ross Ice Shelf and only 3 years for the Filchner Ice Shelf. This indicates that the forced response of the ocean to the SSP5-8.5 scenario simply overwhelms the variation between initial conditions.

In all SSP5-8.5 runs, warm water intrusion into the Weddell and Ross continental shelves are preceded by progressive freshening. Therefore, although initial fresh biases appear in all our ensemble members in these two regions, the magnitude of the biases may affect the timing of the intrusion. However, finding a suitable metric to represent the averaged biases is not simple given the spatial bias heterogeneity in those continental shelves. In addition to improving the initial biases, evaluation of such a metric will be needed for future work on this topic. This evaluation will also be a helpful tool in analysing the impact of freshening trend on the intrusion.

The very wide range of uncertainty produced by ISMIP6 (Seroussi et al., 2020) suggests that our ensemble spread is much smaller than the systematic uncertainty in ice-sheet modelling to project AIS mass balance for a given climate change scenario. Much of the ISMIP6 spread comes from their multi-model approach, i.e. structural uncertainty, but within UKESM1.0 alone we could include an assessment of parametric uncertainty in our approach. Key parameters to test would be those controlling our ice shelf basal melting parameterisation and the ice sheet dynamics, where uncertainties in ice stiffness and basal sliding are important factors in simulating how fast the ice responds to changes in thickness.

Our simulations have produced large changes in the surface and basal forcing of the AIS at the end of the 21st century, but the slow response of ice sheet drawdown means that much longer simulations are required to see the implications of these changes on the dynamics and overall mass balance of the ice sheet. Major changes of ice sheet shape and extent evolve on centennial timescales (Noble et al., 2020), as do the feedbacks that then occur in the climate that we wish to study with UKESM1.0-ice. UKESM1.0-ice is a computationally expensive model to run, and the formulation of robust experimental protocols to enable such studies are a focus for future work. Extensive thinning of the ice shelves due to basal and surface melting, as well as the effect of hydrology on the shelf, strongly increases the likelihood of hydrofracturing and ice shelf collapse on these time scales (DeConto and Pollard, 2016), which are phenomena we cannot currently simulate in UKESM1.0-ice. Large changes in climate in high emissions scenarios also bring into question our use of fixed internal ice sheet temperatures and basal traction coefficients for longer timescale simulations. Furthermore, other missing physical processes that are likely significant are a dynamic calving front, subglacial hydrology/basal physics and the impact of fracture and damage to the ice sheet rheology. These issues will all be important areas of future work for multi-centennial scenario simulations.

## 5 Conclusions

In this study, we carry out small ensembles of SSP1-1.9 and SSP5-8.5 simulations using the UKESM1.0-ice Earth System Model, which makes these the first simulations with an AOGCM that has two-way coupling between atmosphere and ocean components to dynamic models of the Greenland and Antarctic ice sheets. This enables us to identify important 21st century climate signals around Antarctica, taking explicit account of ice sheet - climate feedbacks. Despite some biases, the 21st century projections are computationally stable throughout the simulations and result in plausible ice sheet mass budgets. This demonstrates promising capabilities that the UKESM1.0-ice has for further research into Earth system projection simulations.

Under the low emission SSP1-1.9 scenario, no major changes to ice shelf basal melting, SMB, or ice discharge in AIS are simulated in any ensemble members. These lead to a projection of $22.2 \pm 2.8$ mm of sea level rise by 2100 from the AIS under this scenario. Under the high emission SSP5-8.5 scenario, all ensemble members simulate the initiation of strong melting under the Ross and Filchner ice shelves before the end of the century, strong negative SMB on ice shelves and increases in accumulation on the grounded ice sheet. The increase in accumulation is the largest contribution to the change in mass above flotation before 2100, such that our SSP5-8.5 simulations project a negative AIS contribution to sea-level rise, reducing the overall global mean sea level rise by $21.2 \pm 5.1$ mm in 2100 under this scenario.

Our simulations highlight important processes that could affect the Antarctic shelf seas in the future. Although the present-day pattern of increasing ice discharge in the Amundsen sector is not simulated by our model due to limitations in some components, our projections do show the potential for major changes under larger ice shelves. Warm water intrusions into the cavities under the Ross and Filchner ice shelves in the SSP5-8.5 scenario, preceded by the freshening of their nearby continental shelves, causes strong basal melting. The initiation of these warm intrusions and consequent strong melting has a timing variability of only 10 years for the Ross and 3 years for the Filchner across our ensemble members. While the Filchner Trough warming is consistent with many previous modelling studies, the Ross Sea shelf warming is a new result from our projections. Due to this strong response, under the SSP5-8.5 scenario the Ross Ice Shelf unpins from Roosevelt Island and other pinning points and an ongoing retreat of grounding lines in this region is triggered, with up to 90 km of retreat by 2100. Both of these potential changes indicate major threats to the Ross Ice Shelf in the 21[st] century under unmitigated climate change.

Results from these simulations hint at some important areas of future work needed to reduce the sea level rise projection uncertainty. Among them are salinity bias reduction in the Ross and Weddell shelves, improvements of ocean model representations in the Amundsen Sea and enhancements of physics and features of the ice sheet model component.

**Appendix A: Extra Figures**

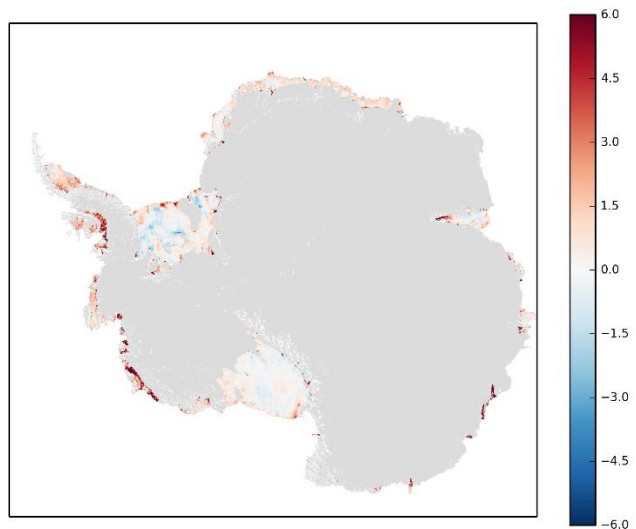

**Figure A1. Ice shelf basal melt rate observation (m/yr) around Antarctica (Adusumilli et al., 2020).**

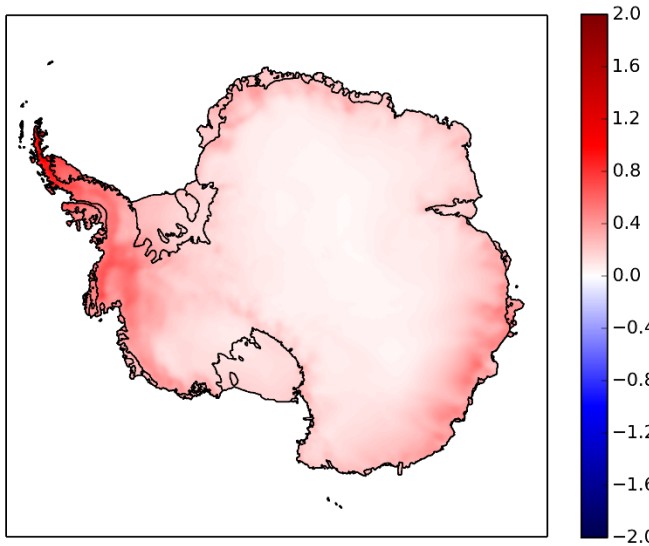

**Figure A2. Surface Mass Balance interpolated from observation (m/yr) around Antarctica (Arthern et al.,2006).**

**Code/Data availability**

UKESM1 source code and model configurations are available via the Met Office Science Repository Service (https://code.metoffice.gov.uk/trac/UKESM), UniCiCles coupling scripts and wrappers via the NCAS-CMS PUMA service (https://puma.nerc.ac.uk/trac/UniCiCles) and BISICLES source from the Los Alamos National Laboratory ANAG repository (https://anagrepo.lbl.gov/svn/BISICLES). The simulation data are archived at the Met Office and are available for research purposes through the JASMIN platform (www.jasmin.ac.uk) maintained by the Centre for Environmental Data Analysis (CEDA); for details please contact UM_collaboration@metoffice.gov.uk

**Author contributions**

AS and RS performed the simulations. AS processed the data. AS, RS, and PH analysed the results, and drafted and revised the manuscript. AJ supervised the ice-ocean interaction work. VL analysed the ice sheet initialisation. JG, VL and PM edited the manuscript. All of the co-authors contributed to scientific discussions throughout the work.

**Competing interests**

The contact author has declared that neither they nor their co-authors have any competing interests.

**Acknowlegements**

The authors would like to acknowledge Stephen Cornford for giving the required knowledge and information about various things related to the BISICLES model, and the UKESM core development team on whose work this model builds. The authors thank Susheel Adusumilli and Rob Arthern for providing the observation data used here for comparison. Model simulations were carried out on the Monsoon and NEXCS collaborative High Performance Computing facilities funded by the Met Office and the Natural Environment Research Council. A. Siahaan, R. S. Smith, P. R. Holland, A. Jenkins, V. Lee, A. J. Payne, and C. G. Jones were funded by the National Environmental Research Council (NERC) national capability grants for the UK Earth System Modeling project, NE/N017978/1 and NE/N01801X/1. C. G. Jones also acknowledges funding from the European Commission under the H2020 Research grant no. 641816 (CRESCENDO). J. Ridley and P. Mathiot were supported by the Met Office Hadley Center Climate Programme funded by BEIS and Defra. The authors would also like to acknowledge the constructive comments and helpful suggestions of our reviewers and editor, who helped to shape and improve this paper.

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
