# Peer review of "The Antarctic contribution to 21st century sea-level rise predicted by the UK Earth System Model with an interactive ice sheet"

_The Cryosphere, 2021_

## Author Comment (AC1)

**Responses to Reviewer #1**

We thank reviewer #1 for their careful and thorough review of the manuscript. Below we copy the referee comments in black and write our responses in blue.

General comments

This paper is a major step forward for coupled ice sheet–climate modeling. It presents results from the first simulations using a complex Earth system model with full two-way coupling of ice sheets to the atmosphere and ocean, for both the Greenland and Antarctic ice sheets (though Antarctica is the focus here). Many earlier studies have argued for the importance of coupling and speculated on what might happen when feedbacks are included. Here, these speculations are put to the test, in ensemble simulations to 2100 for both low-emission and high-emission scenarios. The authors explain why their study is novel, while giving due credit to previous work. The paper is well structured and clearly written, with figures and tables that effectively illustrate the main findings.

The results are both plausible and interesting. For me, the most important findings are (1) the intrusion of warm water into the Ross and Filchner Ice Shelf cavities by the end of this century under a high-emission scenario, with consistent timing across ensemble members; (2) the absence of a strong response in the Amundsen Sea region, where warm water is already present in cavities; (3) the fact that increased snowfall (a near-term response to warming) adds more mass above flotation than the ice sheet dynamic response can remove by 2100, but with the likelihood that the dynamic response would accelerate in the 22nd century. There are many uncertainties – related, for example, to the coarse ocean resolution and the challenges of ice-sheet spin-up. The authors acknowledge these uncertainties and are careful (except for a few minor cases noted below) not to draw conclusions that go beyond the data.

We are grateful to reviewer #1 for recognising that while our study has limitations, it is the first of its kind, and offers several new scientific advances that have not previously been possible. We are grateful to the reviewer for pointing out a few areas where we can better caveat our results, given the acknowledged limitations of this ambitious work.

I have some suggestions to sharpen the text and to guide readers who may be unfamiliar with some of the details, but no major criticisms.

Specific comments

p. 5, l. 127: Here or below, it would be useful to say more about how NEMO computes basal melt rates. It would be good to know, for instance, whether the melt rate is a strictly linear function of the thermal forcing, or if it also depends on the speed of the sub-shelf ocean current.

NEMO computes the melt rates based on the approach described in paper cited (Mathiot et al., 2017) but we understand that readers may not be aware of this, so we will be pleased to expand the associated discussion.

p. 6, l. 162: "ice sheet model projections are typically initialized without any spinup". I would say "often" instead of "typically", because many ice sheet models (roughly half the models in the ISMIP6 projections) are spun up in some way.

We agree and will change this

p. 7, l. 211: Were basal melt rates assumed to be zero in the Cornford (2016) initialization? If so, please state this, since the tuned ice-shelf viscosity could be compensating for missing basal melt.

In the Cornford et al. (2016) initialisation, for a given ice sheet geometry (informed by observations) the model flow equations are instantaneously inverted to find fields of basal friction and stiffening coefficients that minimise the difference between model velocities and those derived from satellite observations. A short relaxation phase follows, where the model is run forward with mass balance boundary conditions provided by surface mass balance and basal mass balance diagnosed to balance the SMB and flow divergence at all points on the ice shelves, and thus maintain their initially prescribed geometry. We will expand the paragraph to describe this.

p. 8, l. 221: What is the magnitude of the steady value where the rms thickness rate settles? In what regions is the remaining drift largest? Did you do multi-century standalone ice sheet runs, continuing with the same forcing? Such runs would increase confidence that the drift is small enough to maintain stable grounding lines.

The approximately steady ice sheet-average RMS dh/dt for the 4 simulations are 0.4, 0.35, 0.38, and 0.34, after 17, 28, 29, and 30 years respectively, and the remaining drift is largest in isolated cells near the coast, and not in regions of dynamically evolving glaciers feeding ice shelves. We will add a discussion of these points to the paper. If we continued these standalone simulations, the grounding lines would continue to retreat because ocean melting is out of balance with the ice sheet in the present day. In any case, our experimental design strongly mitigates any concerns over model drift. Throughout the paper we compare SSP1 to SSP5 projections. Therefore SSP1 serves as a 'control'. To leading order, the effects of model drifts will cancel between SSP1 and SSP5, thus revealing the influences of anthropogenic forcing, which is our main interest.

p. 9, Fig. 1: In general, the figures in the paper are informative and easy to interpret. However, many figures (including this one) use a rainbow colour scale that could be problematic for colour-blind readers. Please consider a different scale.

We will revise the figure colour scales where possible.

p. 11, Fig. 3: In Figs. 3a and 3b, there is good agreement with observations in the thinning of Pine Island and Thwaites Glaciers and thickening of the Kamb Ice Stream. My understanding is that this is largely the result of tuning basal coefficients to match observed ice speeds in the BISICLES spin-up. We would expect the tuned velocities to change the thickness in places where ice flow has recently accelerated (for PIG and Thwaites) or decelerated (for Kamb). This can be inferred from the text, but could be made more explicit for readers who are unfamiliar with tuning strategies.

We will expand the text on lines 289-290 to make this much clearer to the readers.

Another notable feature of Fig. 3 is the general slowing and thinning of ice shelves. The text (p. 12, l. 298) attributes the thinning to the SMB and basal melt forcing from the climate model. Is basal melt primarily responsible for shelf thinning, or does SMB also play an important role?

The thinning is caused by discrepancies in basal and surface forcing between the Cornford et al. (2016) and Martin et al. (2019) initialisation and our relaxation process. Such discrepancies are inevitable without a formal coupled ice and climate initialisation procedure (which is an important but very challenging avenue for future work). There are regions where SMB discrepancy is larger than the basal melt discrepancy, but the basal melt difference is larger around the grounding lines. We will expand the text around line 298 to describe this.

p. 12, l. 297: Why would this slowing mostly occur during the first year of the standalone ice sheet initialization stage? I would expect it to occur more gradually as the shelf thins.

The slowing is caused by the change of basal melt forcing around the grounding lines when we first use melt rates from an ocean model rather than the diagnosed melt rates in the initialisation as described above. The ocean model cannot accurately represent the very thinnest cavities near the grounding line, and so no melting occurs there in the modelled fields. Therefore, the ice generally thickens near the grounding line upon coupling, leading to small

grounding line advances. This is one of the reasons why this ice relaxation period is important. The increase in drag from these re-grounded areas is instantaneously transmitted through the ice shelves, so this causes the ice shelves to rapidly decelerate. We will expand the text to fully explain this point.

p. 12, l. 302: You refer to "the SMB and basal melting implicit in the inverted reference velocities". I am not sure what this means. My understanding is that the SMB and basal melting in the BISICLES spin-up are part of the input forcing, with SMB based on reanalysis and basal melting (possibly?) set to zero. In that case, the shock could be attributed to the fact that the SMB and basal melt rates in the adjustment process (derived from the UKESM historical run and standalone ocean spin-up) are different from the SMB and basal melt rates in the spin-up. It would be helpful to describe or plot the differences.

As described in the above response to the comment on line 211, the SMB is imposed in the initialisation, and dh/dt = 0 is imposed on the ice shelves, and so this means that the implicit basal melting varies as part of the Cornford et al. (2016) initialisation. It is correct however that the shock occurs because the SMB and implicit basal melt rates in this initialisation differ from those derived from the UKESM historical run and standalone ocean spin-up. We will describe these differences in the revised paper.

p. 15, Sect 3.2.1: The abrupt transition to warm water for the Filchner Ice Shelf is a very interesting result in an ESM, consistent with the recent regional studies.

We agree that this is fascinating. This has been previously found in ocean-only and ocean/ice models, but our study is the first to show this in a coupled climate model.

p. 16, Fig. 6: Since the transition occurs near 2100, I suggest extending the x-axis to 2115, if possible.

We will extend the axis.

p. 15, Sect 3.2.2: The transition to a warm Ross Ice Shelf cavity is another very interesting result, notwithstanding the fresh bias in the Ross Sea. Some other studies also show freshening in this region in the beginning or near mid the 21st century.

We feel that this is a key result of our study and we believe the Ross Sea shelf warming is a new result. We are aware of a paper suggesting a freshening in this region - Timmermann and Hellmer (2013); Jacobs et al. (2022), so we will add a discussion of those papers. The region has been freshening for the last 60 years, as noted in one of those papers, but that has not yet led to a transition to a warm state.

p. 29, l. 629: "These results may indicate the bigger potential that Ross/Weddell sectors have in becoming major sea level contributors in future warming scenarios." Here, "bigger" seems to mean "bigger than PIG and Thwaites". It's true that the Ross and Weddell sectors have the potential to become major sea level contributors, but it's also true (based on present-day observations and published simulations) that PIG and especially Thwaites could be major contributors. The Amundsen Sea contribution might not be captured by the model, for the reasons discussed in Section 4.3. So I suggest rewording this claim.

We are grateful to the reviewer for bringing this to our attention and will certainly re-word the claim, to say that the Ross and Weddell sectors have a 'large' potential, rather than 'bigger'.

p. 29, ll. 642ff: This paragraph is a good summary of the novelty and importance of the study.

We thank the reviewer for their supportive comments.

p. 30, l. 666. The SROCC is cited several times, but I couldn't find a citation of AR6. Please add AR6 citations where appropriate. For context, I suggest including the projected GMSL from Antarctica under low and high forcing scenarios, according to AR6.

We will update this text to cite AR6.

p. 30, l. 667: It's plausible that the AIS would have a positive mass balance in this century, but it's misleading to call this a "rapid sea level fall". The snowfall contribution is better described as a modest offset (~2 cm) to a robust global trend of rising sea level (28 to 55 cm by 2100 under SSP1-19, and 63 to 102 cm by 2100 under SSP5-85, according to AR6).

We will expand the text to say that while the AIS contribution to sea level will be significantly negative, this would comprise a modest offset to the overall expected increase. This remains a very important conclusion, however, since the current paradigm is that the AIS could contribute a rapid rise in sea level to 2100.

p. 31, l. 701: "… do not retreat." Doesn't Fig. 15 show some GL retreat for PIG and Thwaites?

We will reword the sentence to clarify that we meant that the GL in SSP5 does not retreat far relative to that in SSP1.

p. 31, l. 705: "Nevertheless, the impact of a future strong climate change in Amundsen Sea cavities is unlikely to be larger than our modelled changes in the Ross/Weddell cavities. This is because the Amundsen continental shelf and ice shelf cavities are already filled with the warm Circumpolar Deep Water and hence there is less potential for further warming and strong ice response." I think this is a bit too strong. It may be true that the ASE cavities have less potential for further warming, but this does not imply less potential for strong ice response. Because of its bed geometry, Thwaites might already be retreating unstably, or might be near a threshold such that it could be tipped into unstable retreat with a small amount of additional warming.

We agree that the sentiment expressed in this sentence is highly uncertain. We will rewrite this section to acknowledge that bed geometry may play an important role in modulating the role of changes in forcing, and that in fact the Amundsen Sea could experience warming, as warmer Circumpolar Deep Water at 2C is present offshore of the shelf break in this region.

p. 32, l. 715: "do not appear to simulate". I suggest "do not simulate".

We will change this.

p. 33, Section 5: Many conclusions already appear in the Discussion section. Since some readers will look at only the Abstract and Conclusions, I suggest adding some content in Section 5. For example:

You say here that these are the first AOGCM runs with full two-way ice-climate coupling; you could add a sentence or two (as in Section 4.1) about why this is important.

You could point out that the Filchner warming is consistent with previous modeling studies, whereas the Ross warming is something new.

You could mention the ASE non-response, with appropriate caveats about uncertainty.

I would not end the paper with a sentence that refers to the Ross Ice Shelf alone.

We will briefly add the above points to the Conclusions, while also trying to keep it short and focussed as far as is possible.

Technical corrections

We thank the reviewer for their careful reading and will address all of the suggested changes below.

p. 1, l. 21: "of the 21st century"

p. 5, l. 137: The phrasing is awkward, with two uses of "along with"

p. 7, l. 189: "integration" without the "s"

p. 7, l. 210: "caving" -> "calving"

p. 13, Fig. 4 caption: "The white boxes"

p. 14, l. 330: "where the SSP1-EM melt rates become"?

p. 14, l. 334: "large cold" -> "large, cold"

p. 16, l. 358: "on the ice front" -> "at the ice front"

p. 16, l. 360: "The shelf" -> "Water on the continental shelf" or something similar. In general, please be specific where the two meanings of "shelf" could be confused.

p. 16, l. 361: "becomes" -> "is" (since the deep water is not becoming denser in an absolute sense, but only relative to the shelf)

p. 16, Fig. 6i: In this panel the x-axis is different from the other panels.

p. 18, Fig. 8a: The dashed lines in this panel are hard to see in the ice shelf and continental shelf regions.

p. 18, l. 415: Add comma after "simulation"

p. 20, l. 441: Delete "is" before "already"

p. 23, l. 501, Fig. 11 caption: Typo in "SSP5-EM"

p. 24, l. 516: "on Queen Maud Land" -> "in Queen Maud Land"

p. 25, l. 539, Fig. 13 caption: It's not accurate to describe the right-hand panels as "changes" like the left and middle panels. Please reword, e.g. using "differences".

p. 26, l. 576: "end of the 2060s"

p. 27, Fig. 14 caption: The descriptions of the middle and bottom rows are reversed. In the fourth line, delete "the" before "West Antarctica". In the last line, "column" -> "columns".

p. 27, l. 588: "area-integrated" with a hyphen; delete "area" after "grounded ice sheet"

p. 27, l. 591: "from the 2040s"

p. 28, l. 598: Change "retreat up to 40 km takes place under southern Thwaites Glacier" to something like "the grounding line of Thwaites Glacier retreats southward by up to 40 km" (since the southern part of Thwaites Glacier lies far in the interior). Similarly for PIG.

p. 28, Fig. 15: It took me a few moments to get my bearings for the left and middle panels, which are rotated with respect to the standard view in a polar stereographic projection (e.g., Figs. 11-13). Maybe rotate back to the standard view. On the left panel, perhaps add labels pointing to the Thwaites and PIG ice shelves.

p. 29, l. 639: Instead of "a while", maybe "longer.

p. 29, l. 647: "modern-day" with a hyphen, or just "modern". Similarly, "present-day" at l. 652.

p. 31, l. 696: "end of the 21st century"

p. 32, l. 726: I can't tell if there is a paragraph break after "century". If not, please add one.

p. 33, l. 757: "brings" -> "bring"

p. 33, l. 759: Add a period.

References: Please check for consistent capitalization in paper titles.

Citation: https://doi.org/10.5194/tc-2021-371-RC1

---

## Author Comment (AC2)

**Responses to Reviewer #2**

We thank reviewer #2 for their careful and thorough reviews of the manuscript. We first give a general response to the main comments and major concerns. Below we copy the referee comments in black and write our responses in blue.

The manuscript "The Antarctic contribution to 21st century sea-level rise predicted by the UK Earth System Model with an interactive ice sheet" by Siahaan et al. presents perhaps the first results of a climate model coupled to an ice-sheet model of the Antarctic Ice Sheet.  The paper presents a protocol for offline data coupling of UKESM and BISICLES and explains the initialization process for this complex configuration.  After this, results are presented for small ensembles of coupled simulations following both SSP1-1.9 and SSP5-8.5 greenhouse gas emission scenarios.  SSP1-1.9 shows small changes from initial conditions, while SSP5-8.5 leads to ice-shelf basal melt regime change beneath Ross and Filchner-Ronne ice shelves, and associated ice-shelf thinning and acceleration.  The manuscript discusses the major terms of the ice-sheet mass balance and considers unique aspects and limitations of the modeling approach presented.

The manuscripts presents a significant achievement of running coupled climate and ice-sheet models for Antarctica.  This has been a community goal for many years, and these results are the first to achieve it that I am aware of, even given the "offline" coupling method employed.  On the other hand, the methodology for achieving it includes a number of questionable choices with an unclear impact on the resulting simulation.  Most significantly, the initialization procedure for both the climate model and ice the ice-sheet model appears a bit ad hoc, with a number of details of the protocol not clearly described.  It appears there may be significant artifacts and model drift associated with the ice-sheet model initialization that have not been quantified.  The authors do a fair job of noting shortcomings, but more work could be done in quantifying the impacts of those choices.

On the whole, this is an impressive modeling achievement, but the scientific utility of the results is questionable without further substantiation.  The authors themselves note these limitations, and I wonder if this paper would not be better suited for a journal like GMD. Below I focus on two major areas that require more work - description of the coupling and description and analysis of the initialization procedure.  I then list a number of smaller issues that require addressing.  Even after significant additional analysis, it may be that the initialization and coupling procedure leave the scientific intepretation of the results ambiguous.  The authors may wish to consider if a model description journal would be a more appropriate venue for this work, where it would be a significant contribution.

We are grateful to the reviewer for their careful and thoughtful review, and we agree with most of the points raised.  We are very pleased that the reviewer recognises this work as a significant achievement, and a major 'first' in our field. However we also fully accept that this ground-breaking work has uncovered a few areas that will require a lot of work to get right - most notably in the initialisation. The reviewer is absolutely right to raise these areas. In order to accomplish this first study, we made a series of choices in these areas which we would like to examine more fully in future. However, we feel that our experimental design means that the present manuscript is a very valuable scientific (not only technical) contribution which will be of interest to an audience outside of the model development context and with conclusions that stand despite these questions. We detail our main arguments below, responding to the main points raised in the reviewer's introductory text. These arguments are fully expanded in the responses to the reviewer's specific points.

**Results are sensitive to initialisation and coupling choices:** There is certainly much work to do in understanding the best way to couple ice sheet and climate models, and how to initialise such a coupled system. However, our guiding philosophy throughout this work

has been that the importance of these choices is strongly mitigated by our experimental design. Throughout the paper we qualitatively compare SSP1 and SSP5 projections. Our conclusions are based on this difference alone, with SSP1 effectively acting as a 'control' simulation. Since the impact of initialisation shocks, model drifts, coupling choices, etc. will be expressed in both scenarios, to leading order their differences are independent of these features. On reflection, this overarching philosophy was not expressed clearly enough in the submitted manuscript, and was not repeated in areas where it is relevant, so we will substantially rewrite the paper to clarify this throughout.

**Initialisation procedure is ad-hoc:** Our ambitious coupled modelling work highlights the important and very challenging area of coupled ice–climate model initialisation. There is a fundamental difficulty that climate models are spun up using pre-industrial forcing, while predictive ice sheet models are commonly initialised in the present day using inversion techniques. Any ice–climate coupling will therefore induce some kind of discontinuity into the simulations, and removing the impact of this is a challenging task that will require years of dedicated effort. In the present study we have devised a strategy that works, involving defensible choices, and with acceptable impacts. Since our experimental design mitigates these impacts even further, we believe that this issue should not delay publication of the substantial scientific advances that we have made with this coupled model. To address the reviewer's comments in this area, we will expand the discussion of the challenges of coupled initialisation, and outline possible future avenues for minimising the resulting impacts.

**Initialisation and coupling procedure are not fully described:** The coupling procedure is very fully described in Smith et al. (2021) and that is quite briefly described here. We will expand this slightly where recommended by the reviewer. The initialisation procedure is complex and not fully described elsewhere, so we will expand the discussion substantially, including the above philosophical points. However, there are limits to this because this is a scientific paper, not a model description paper, as justified below.

**Work may be more suitable to a model description journal:** We considered this point at length, both before the original submission and again in response to the reviews. While this manuscript represents a substantial modelling advance, much of the model development is already described in a technical paper (Smith et al., 2021) and the conclusions of the present study are scientific, not technical, in nature. In particular, our conclusions on warming of the Ross and Weddell seas, and the importance of surface mass balance in creating a negative Antarctic contribution to sea-level at 2100, are scientific advances that will be of wide importance and impact. We will revise the paper by more clearly stating that our intentions are to analyse scientific hypotheses, including at the end of the introduction.

**MAJOR CONCERNS**

**1. Better description of coupling.**

I recognize that the coupling methodology was previously described in the Smith et al. (2021) JAMES paper, but given the importance of the coupling to the science results, inclusion of more information here is warranted. Some specific suggestions are:

142-6: The description of the coupling protocol is vague. Given this is a significant novelty of this work and coupled model results can be very sensitive to the coupling procedure, it should be described in much more detail. Please provide evidence that the results are not sensitive to the chosen coupling interval.

The coupling protocol is exactly that described at length in Smith et al. (2021), but we will expand and tighten the summary of these details in the present paper.

Early in the development of the coupled model we examined the sensitivity to coupling interval and satisfied ourselves that 1 year was adequate for the flows in the resolutions characteristic of the global models we are using. However this was many years ago and unfortunately the results are no longer available. It would be prohibitively difficult to re-run these tests with the present coupled model as it would require substantial development of the coupling suite. However the 1-year coupling step is not controversial for these kind of experiments with slow centennial evolution of the climate forcing. Favier et al., 2019 (Geosci. Model Dev., 12, 2255–2283, 2019) indicates very little sensitivity to coupling periods between 1 month and 1 year, while Zhao et al., 2022 (Geosci. Model Dev., 12, 2255–2283, 2019) indicates very little sensitivity to the coupling time interval between 0.5 days and 3 months. We will modify the paper to discuss this point and cite the papers.

130: This sentence is unclear - please clarify or expand on this. Also, please add a description of how retreat of the ice-sheet grounding line is handled by the ocean model.

We will modify the paper as suggested. This is a complex area as the depth-integrated ocean transport must be preserved as the ice geometry changes, in order to avoid creating 'tsunamis' in the ocean surface. A retreat of the grounding line is actually the simplest case, since the new ocean column that is created can simply be given zero ocean transport. An advance of the grounding line is more complex, as some ocean transport then 'disappears'. The full details are given by Smith et al. (2021), but we will certainly expand the discussion in the present paper.

126-146: Please acknowledge in this section that the calving front restriction and the bilinear remapping prevent the system from conserving mass and heat, though these errors are not likely to be significant for the experiments being conducted.

We agree with this and will add this acknowledgement.

**2. Model initialization and its impacts.**

Initialization of coupled climate and ice-sheet models is a very challenging problem. Still, the procedure described here appears ad hoc, missing key information, and does not include an assessment of key initialization choices on the results. The specific point below detail these concerns:

This is a very complex area and we agree its explanation could be clearer. In response to the comments below we will substantially re-write section 2.3. After some consideration we think we can improve the clarity of this section by clearly labelling the different stages of the simulations, as follows:

A.      standalone ice model inversion (Cornford et al, 2015)
B.      standalone ice model relaxation (Cornford et al., 2015)
C.      generate ocean cavity properties (1 simulation, 45 years)
D.      add ocean cavity properties to global UKESM ocean and relax (4 simulations, 15 years)
E.      relax ice model under UKESM ocean cavity and surface mass balance forcing (4 simulations, 17-30 years)
F.      coupled projections (8 simulations, of which first 5 years is regarded as spinup)

We will rewrite the section to describe each of these stages, and will add a schematic diagram illustrating them.

179: This paragraph is confusing and is missing key information.

The issue under consideration here is that we have the CMIP6 UKESM simulations for historical periods but those have a static ice sheet, and no cavities beneath floating ice. They simply have a coastline at the ice front. Therefore we need to generate initial ocean properties and ice shelf melt rates within the sub-ice cavities from a short run of the coupled model. We will expand and clarify this paragraph.

182: 45 years seems like a short ocean spin-up time. Can you provide justification that the most important transient behavior had reduced to small levels prior to creating the branch runs?

In retrospect, the original manuscript was not sufficiently clear on the purpose of this initial 45-year simulation (stage C above). Its only goal here is to produce some preliminary ocean properties in the ice shelf cavities, and 45 years is more than enough to accomplish this. In stage D, these cavity properties are then joined on to global ocean states from UKESM historical runs, and that is run forward for 15 years in order to flush the cavities with ocean properties from the global UKESM. Thus the convergence of stage D is important to the simulations, but the convergence of stage C is not. This will be fully described in the paper, and the convergence of stage D described (see below).

179-194: After reading the full paragraph, I am more confused. Replace the opening paragraph by clearly stating the initialization process is based on a hybrid state of the ice cavities from UKESM1.0-ice stitched onto the global ocean state of UKESM1.0. Also, a schematic of the various runs and processing, would help communicate this process.

This is a complex area and we will certainly replace the opening paragraph as requested and add the schematic.

186: Emphasize for the reader that UKESM1.0 does not include ice shelves, e.g. add a parenthetical "(without ice shelves)" after "UKESM1.0".

We will add this emphasis, reminding the reader that the suffix 'ice' in UKESM1.0-ice is added to refer to models with ice shelf cavities and an active ice sheet.

187: Which years were chosen? Can you elaborate on what "a range of variability" means? How many ensemble members were in the UKESM1.0 CMIP6 historical ensemble? Also, do you have a reference for that ensemble, or a reference to the dataset on ESGF? What does "end of the 20th century" mean? What specific years were used?

The UKESM1.0 CMIP6 historical ensemble contains 19 members, and is most usefully cited via Yool et al, "Evaluating the physical and biogeochemical state of the global ocean component of UKESM1 in CMIP6 historical simulations" GMD 14, (2021). Three of the initial states were taken directly from this ensemble, all from the year 2000 in their respective members. They were chosen to maximise the range of ACC strength and Southern Ocean annual average SST across the available UKESM ensemble members in that year. An error was made in configuration meaning that the fourth was not, in fact, taken from the UKESM historical ensemble, but from the preliminary UKESM1.0-ice simulation mentioned in the paper. However, since this had not been run long enough to drift far from the 1995-2014 average EN4 climatology it was initialised with, it provides a state more representative of the observed modern ocean than the spun-up UKESM historical ensemble that contains systematic biases characteristic of UKESM, so may show if those biases have a significant impact on our projections. We will add clarification of the source of our initial states to the revision.

189: What does "short" mean?  And what does "balance" mean on line 190?  Please show evidence of balance or behavior that is close to steady state, either in the form of a plot or some statistics.

We will replace 'short' with '15-year', as described in the following paragraph.  We will add some statistics to this section showing that the melting is steady.

196: Please explain why the ocean-sea ice simulations are regarded as beginning in 2000 if they were branched from specific times of UKESM and included a cavity state from perpetual 1970.

We will expand the discussion. As noted above in the comment about line 187, the global ocean-sea ice states were taken either from the year 2000 in one of the members in the UKESM1 historical ensemble, or represent the modern observed ocean (averaged 1995-2014). The ice-shelf cavity information does come from perpetual 1970 forcing UKESM1.0-ice simulations (stage C), but is merged with these global ocean states and run for 15 years with UKESM surface forcing from 2000 to 2015 (stage D). This flushes the cavities and bring them in line with the global ocean, and produces an ocean state representative of year 2015.

197: Are the atmospheric fluxes from the same runs that each was branched from?  Or one common set of atmospheric fluxes?

We will add a note that the atmospheric fluxes were taken from the same ensemble member as the initial global ocean state.

195-202: So the final initialization step is 1) not fully coupled to the atmosphere and 2) has temperature and salinity restoring applied.  Given that, it is unlikely that you can branch into a fully coupled projection without some shock and drift.  Please justify this choice.

We will expand the discussion at this point to reflect two philosophical points and one direct answer to the reviewer's comment.

Firstly, the topic of coupled ice-climate initialisation is a challenging research question that will take much future research to fully solve. Secondly, our experimental design strongly mitigates the influence of any coupling shock. Thirdly, we believe that our strategy for this particular issue is broadly defensible and is hard to improve upon.

Stage D of the initialisation procedure is intended to produce ice-shelf cavity ocean properties that are consistent with the UKESM1.0 ocean state in the wider ocean. There is no perfect way to do this, but we believe our strategy is the best available.  We are constraining ocean surface fluxes to match UKESM1.0, and restoring ocean properties outside the ice-shelf cavities to match UKESM1.0. Thus, we believe that in 2015, after 15 years of this procedure, we have ice-shelf cavity ocean properties that are the best match possible to the wider UKESM1.0 global ocean states.  Thus, when we subsequently run this forward as a coupled model, we do not expect large ocean-state shocks to occur. The 'perfect' solution to this problem would be to have a historical climate model simulation with ice-shelf cavities, but these are not currently available. This is our strategy for the next generation of the UKESM.

195-202: Throughout this section, please clarify if every time you refer to UKESM1.0-ice, prognostic ice-shelf basal melt fluxes are on and what is happening with iceberg fluxes (if anything).

Both ice shelf basal rate and iceberg calving, drift, and melting are always prognostic in UKESM1.0-ice. We will state this as requested.

214: What is the thickness output of the inversion procedure? In the previous sentence it is only said that basal drag coefficients and viscosity are adjusted.

The thickness output is described fully in Cornford et al. (2016). Over the ice shelves it is equal to the observed ice thickness from Bedmap2. For grounded ice, it is this ice thickness after the inversion (stage A) but subject to relaxation (stage B). We will add a discussion to the paper, and a citation to Cornford et al. (2016).

216: Ice-shelf melting from what year(s) of each standalone run?

In order to test the influence of interannual variability, two members use 2014 melting whereas two others use the 2010-2015 average, in common with the SMB forcing noted in line 218. This will be noted.

221: Referring to this 'spike' is vague, as is "about 20 years". Please report statistics or, better, show a time series of this RMS metric.

The approximately steady ice sheet-average RMS dh/dt for the 4 simulations are 0.4, 0.35, 0.38, and 0.34, after 17, 28, 29, and 30 years respectively, and the remaining drift is largest in isolated cells near the coast, and not in regions of dynamically evolving glaciers feeding ice shelves. These statistics will be added to the paper. Ice simulations in stage E were motivated by a wish to remove any immediate coupling shock on introducing UKESM surface and basal mass balance fields to the ice sheet model and ensure numerical stability of the initial steps of the interactive ice-climate system in stage F. Stage E simulations were 30 years long. The ice states taken forward to phase F were chosen to have a continental average RMS dh/dt of less than 0.5m/yr and a maximum magnitude of dh/dt in any location less than 75 m/yr. The state taken at year 17 was taken earlier than for the others to avoid the effect of a significant transient spike in both measures of dh/dt that occurs near the end of that simulation.

204-223: It is entirely clear what year the ice sheet initial condition represents. In line 212, it is stated the ice-sheet state starting the procedure represents "early 21st century", but then it is relaxed for "about 20 years". How does the final state of each ensemble member compare to recent observed thinning rates, which have been highly variable in time?

The ice sheet inversion (stage A) represents the ice thickness from the Bedmap2 dataset and the ice velocity from the Rignot dataset. These data are from a range of dates and so the inverted ice state does not have a clear uniform time-stamp other than 'early 21st century'. In stage E, the ice sheet is then evolved for a further 17-30 years so that this ice sheet state can adjust to the four different realisations of 2014 or 2010-2015 basal melt forcing produced by the UKESM. So the ice model state upon coupling does not have a clear time stamp, other than it is 'early 21st century'. This will be fully described in the revised paper.

Figure 3 and associated ext (286-292): The very large amount of noise (presumably transient flux divergence) in the ice-sheet elevation/thickness change deserves a few sentences of explanation. While this is a well known and common challenge for ice-sheet models, the amount of spurious thickness change after 15 years of integration and 20 years of relaxation (if I followed the protocol correctly - it was confusing - see above) seems unacceptably large. Also, what is the purpose of panel d? It shows nearly the same information as panel b.

We will clarify this in the text in the revised manuscript. We do not feel that the noise in Figure 3b is unacceptably large - it is of order 0.1 m/y. Crucially, as shown in figure 13 and discussed further below, this 'noise' exists in both the SSP1 and SSP5 projections, and is cancelled in their difference. Therefore, despite the 'noise', we are able to determine the influence of climatic forcing. The purpose of figure 3d is to show the total ice thickness change over the ice model relaxation (stage E), while the purpose of figure 3b is to show the initial trend during the projection period.

297: It is not obvious to me why ice shelves would slow so significantly in the first year just due to one year of thickness change coming from surface and basal mass balance, especially if further adjustment after one year is small. Is it possible grounding line positions have shifted or the ice temperature field changed or something else is going on? A more thorough explanation is warranted.

The slowing is caused by the change of basal melt forcing around the grounding lines when we first use melt rates from an ocean model rather than the 'implicit' melt rates in the ice initialisation (stages A and B). The ocean model cannot accurately represent the very thinnest cavities near the grounding line, and so no melting occurs there in the modelled fields. Therefore, the ice generally thickens near the grounding line upon coupling, leading to small grounding line advances. This is one of the reasons why this ice relaxation period is important. The increase in drag from these re-grounded areas is instantaneously transmitted through the ice shelves, so this causes the ice shelves to rapidly decelerate. We realise that this was not well explained in the original manuscript and will expand the text to fully explain this point.

Section 3.4: It would be easier to interpret the changes presented if there was also a standalone BISICLES control run that had constant 2015 forcing. Presumably the speckly pattern of thinning and thickening in Fig. 13 panels a and b is due to unrealistic transient behavior in the initial condition. That is a common problem in ice-sheet models, so it does not necessarily invalidate the results, but it should be clearly identified. I would prefer to see additional results for an unforced control run. Without it, it is difficult to assess what aspect of these results are an effect of the ice-sheet model initialization procedure and what is due to the climate forcing coming from UKESM.

We thank the reviewer for pointing this out. The 'noise' in figure 13 is the same between SSP1 and SSP5 scenarios, so it must be associated with the initial state. However we feel that this 'noise' is not large relative to the dynamic thinning signals of interest, and most crucially of all, it does not appear when we difference SSP1 and SSP5 in figure 13. This means that the climatic signals that we are focussing on are not influenced by the 'noise. This will be fully described in the revised paper. We considered the possibility of conducting a standalone ice control run but this is problematic for two reasons: i) the ice would evolve because ocean melting is out of balance with the ice sheet in the present day and ii) it is not clear what melt rates we would use when the grounding lines retreated.

**Other Comments:**

Abstract: Mentioning Greenland in the abstract is slightly misleading, because the Greenland results are not part of this paper.

We will clarify that while Greenland is coupled into the model, we only analyse Antarctica here.

47: Also cite the only paper that demonstrates this for the observational period that this sentence discusses:

Gudmundsson, G.H., Paolo, F.S., Adusumilli, S., Fricker, H.A., 2019. Instantaneous Antarctic ice sheet mass loss driven by thinning ice shelves. Geophys. Res. Lett. 46, 13903–13909. doi:10.1029/2019GL085027

We will update the references

48-55: There are a lot of other important studies that would be appropriate to reference here, e.g.:

Spence, P., Holmes, R.M., Hogg, A.M., Griffies, S.M., Stewart, K.D., England, M.H., 2017. Localized rapid warming of West Antarctic subsurface waters by remote winds. Nat. Clim. Chang. 7, 595–603. doi:10.1038/nclimate3335

We will update the references

65: CMIP5 and CMIP6

We will change this.

68: One CMIP-class ESM recently published (since this manuscript was submitted) Antarctic subglacial melt rates (but those simulations were not part of CMIP6):

Comeau, D., Asay-Davis, X. S., Begeman, C., Hoffman, M. J., Lin, W., Petersen, M. R., et al. (2022). The DOE E3SM v1.2 Cryosphere Configuration: Description and Simulated Antarctic Ice-Shelf Basal Melting. Journal of Advances in Modeling Earth Systems, 14, e2021MS002468. https://doi.org/10.1029/2021MS002468

We will update the references

69-74: There also is recently published (since this manuscript was submitted) regional model that includes all physical climate components (atmosphere, ocean, sea ice, land, ice sheet):

Pelletier, C., Fichefet, T., Goosse, H., Haubner, K., Helsen, S., Huot, P.-V., Kittel, C., Klein, F., Le clec'h, S., van Lipzig, N.P.M., Marchi, S., Massonnet, F., Mathiot, P., Moravveji, E., Moreno-Chamarro, E., Ortega, P., Pattyn, F., Souverijns, N., Van Achter, G., Vanden Broucke, S., Vanhulle, A., Verfaillie, D., Zipf, L., 2022. PARASO, a circum-Antarctic fully coupled ice-sheet–ocean–sea-ice–atmosphere–land model involving f.ETISh1.7, NEMO3.6, LIM3.6, COSMO5.0 and CLM4.5. Geosci. Model Dev. 15, 553–594. doi:10.5194/gmd-15-553-2022

We will update the references

58-81: Somewhere in here you should also acknowledge the fully coupled configuration of CESM with the Greenland Ice Sheet.

We will update the references

104: Can you report the approximate horizontal resolution of the 1 degree ocean grid at the typical latitude of Antarctic ice shelves?

About 17-22 km.

111: It is worth pointing out that this adaptivity is dynamic in time.

We will write 'time-evolving adaptive' the first time we mention this.

117: Can you briefly summarize the impact of choosing 2 km as your finest resolution instead of 1 km or 0.5 km as is sometimes used for BISICLES?

We will update the text in the discussion section (the 2 km highest level of mesh refinement we allowed for BISICLES in these simulations may not be sufficient to accurately model the grounding line dynamics in this region (Cornford et al., 2016), although testing suggests that increasing the allowed refinement of the BISICLES to 500 m would not significantly alter our model evolution of the next few decades.)

162: I would say most ice-sheet models follow this practice, but it is not 'typically' the case - there are a number of ice-sheet models that do a paleo or steady state spinup.

We will use 'often' instead.

169-171: These comments make me wonder if this manuscript would be more appropriate for GMD or JAMES.

Though the paper describes some technical detail, the main conclusions are all scientific in nature, and so we believe this work is much better suited to The Cryosphere.

Figure 1a,b and associated text in 3.1: It is rather awkward to compare these plots to referenced observational data without showing those observations or model biases relative to them. As presented, these comparisons are not useful.

There are no complete observational data sets of the stream function or mixed-layer depth, so we just show the model results. We will mention this in the text. For temperature and salinity, we compare to observations.

264: Would not surface restoring bring properties closer to observations?

This is correct so we will remove the reference to surface restoring and then refer to the inadequacy of ocean-atmosphere surface flux in the stand alone ocean model.

249-271: This discussion of water mass properties, especially at depth, based on mixed layer depth is obtuse. It would be much better to show T&S diagrams for the regions of interest compared to observations. Many global ocean models struggle with the formation of Dense Shelf Water, even with realistic mixed layer depths, so that in and of itself is not a guarantee of good water mass properties. It would also be quite important to see maps of ocean bottom temperature and salinity, as those matter more for ice-shelf basal melting than surface properties.

In Figure 1 we show the water properties over the depth range important to ice-shelf melting, which is the focus of this study. This shows how well the water masses on the shelf are represented. We refer to mixed layer depths merely to describe why the shelf water masses appear as they are (in the real world and in the model) - it is the presence or absence of cold, salty waters produced by sea ice formation that determines this. We will clarify this in the revised paper.

271-2: Please provide evidence for this statement (e.g. the bias value for UKESM and other CMIP models). Are you basing this statement off of the version of UKESM in Heuze (2020) or the simulations presented here?

We realise that the citation to Heuze was misleading. We meant that the temperatures were broadly accurate (Figure 1) and this was superior to many other models (Heuze). We will change the text accordingly.

278: Do you mean *near-shore* fresh bias here? Over most of the Southern Ocean, Fig. 1 shows a saline bias.

We are referring to the fresh bias in the Ross continental shelf and will change the text to reflect this..

Figure 4: Similar to figure 1, this figure should include the observational references fields (or show an anomaly). Simply saying "shows a similar spatial pattern to present day observations (Rignot et al., 2013; Adusumilli et al., 2020)" and expecting a reader to pull those up and make comparisons across different colourbars is not sufficient. Also the linear colourbar is inadequate for showing the high melt rates in the Amundsen Sea - presumably the ice shelves in that entire sector are well above 5 m/yr. Similarly, the colourbar in panel e and f saturates in areas of interest (e.g., Ross and Larsen ice shelves).

We will use a nonlinear colour bar and add the satellite-derived melting.

Table 1: Presumably in your model analysis you can separate Ross and FRIS into the two halves that the observational data uses.

We can add this.

309: While the modeled melt might be within the range, I suspect a t-test would indicate a significant difference. That is not necessarily unacceptable, but please report a more careful comparison.

The errors in the observational datasets are very large and the disagreement between datasets is large. For this reason we are unsure that detailed statistics are appropriate. We will re-phrase the sentence to say that the model is loosely in agreement.

Figure 5: Please also show present-day observational estimates for reference.

We will add these to the plot

Section 3.2.1: This section demonstrates the melt regime change at FRIS very clearly, but the causal mechanism is left only hinted at. There is a plot and mention of declining sea ice volume and its possible relation to declining density. There also is a mention of increasing freshwater flux. This is already a long, dense paper, but if it were possible to tease out the mechanism(s) leading to WDW increase, that would be a valuable contribution. Have you looked at changes in wind stress? The previous papers you cite also discuss that as a potential mechanism for the WDW intrusion at FRIS.

Our results share many aspects with the cited papers, so we choose to rely on those for FRIS and provide a fuller explanation of the changes in the Ross Sea. This will be described more fully.

424-6: From Figure 9, it looks like a missing piece of this explanation is that Dense Shelf Water (cold and saline) on the continental shelf is present at the start of the simulation, but becomes significantly fresher by 2060 (Fig. 9e). This is consistent with the sea-ice decline mentioned and shown in Fig. 8b to become more substantial around that time. Similar to the

FRIS case, the reduction in the continental shelf density barrier facilitates the intrusion of mCDW. This series of events is alluded to in this paragraph, but the sentences at 426-7 implies that the driving mechanism is warming of the mCDW, which is not apparent in Figure 9. Maybe this just requires some rewording.

We agree with this narrative and will change the wording.

432: As you say, I think the relative model fresh bias in each of these regions is critical. To further illustrate that point, could you follow up with a quantitative metric of the salinity bias in each region at the start of the simulation? (e.g. averaged over the region or at the shelf break analysis point used in each region.

We will add a metric

440: You haven't shown that the regional climate is warming during this period. Maybe reword to "While the changing climate".

We will reword this

441: Remove "is".

We will reword this

Fig. 10: What year and simulation are these draft values from?

Year 2080 from one of the SSP5 members. We will clarify the caption to this figure.

Section 3.2.3: Initially I was skeptical of even discussing results from an ice shelf represented by 11 grid cells, but I appreciate the honest assessment of what is happening in the model here, given the importance of this region. Better to acknowledge the limitations of interpreting these results than ignore it and risk readers reading their own interpretation into it.

We agree

491: Similar to previous comments, simply stating your results look similar to your observational reference is insufficient. Please include a panel in the figure showing the reference dataset (or the difference from it).

We will add the data

Table 2: Please also include a present-day estimate (e.g. from RACMO).

We will add this

525: Another relevant reference here: Trusel, L.D., Frey, K.E., Das, S.B., Karnauskas, K.B., Munneke, P.K., Meijgaard, E. Van, Broeke, M.R. Van Den, 2015. Divergent trajectories of Antarctic surface melt under two twenty-first-century climate scenarios. doi:10.1038/NGEO2563

We will update the references

541: The Thwaites and Pine Island inland thinning goes away when you difference the two scenarios, and there is in fact less thinning in the SSP5 scenario. Please discuss this. Having a control run for context (previous comment) would likely help here.

This is consistent with the results of the paper. The dynamic thinning in the Amundsen region is the same in both projections. We don't place much faith in this, because the ice shelves are not well resolved. SSP5 has a greater SMB than SSP1, hence 'less thinning'. We will clarify this in the text. SSP1 is effectively the control run.

Figure 14: Typo in 'cumulative' in title above panel b.

We will reword this

574: Consider rewording this sentence to avoid the possible interpretation that the thinning of Ross Ice Shelf has a direct impact on MAF.

We will reword this

580: This goes back to my earlier comment about what Thwaites and Pine Island are doing in the control run.

We will reword this

Figure 15: Consider using the same colour scheme for the two scenarios here as in the previous figure.

We will change this

642. 655-8: Maybe GMD/JAMES is a better fit?

The scientific conclusions of our paper are a good fit to The Cryosphere. We are primarily interested in differences between SSP1 and SSP5.

Section 4.2: A short comparison of the results to those of ISMIP6-AIS is warranted, as that set of experiments is perhaps the closest point of reference to this work. In addition to considering the overall behavior of each region, it would be interesting to look at the threshold for surface hydrofracture employed by ISMIP6 and if/when that is passed in your simulations. Similarly, comparing your simulated basal melt rates to the parameterization they employ might help explain differences in response.

ISMIP6: Seroussi, H., Nowicki, S., Payne, A.J., Goelzer, H., Lipscomb, W.H., Abe-Ouchi, A., Agosta, C., Albrecht, T., Asay-Davis, X., Barthel, A., Calov, R., Cullather, R., Dumas, C., Galton-Fenzi, B.K., Gladstone, R., Golledge, N.R., Gregory, J.M., Greve, R., Hattermann, T., Hoffman, M.J., Humbert, A., Huybrechts, P., Jourdain, N.C., Kleiner, T., Larour, E., Leguy, G.R., Lowry, D.P., Little, C.M., Morlighem, M., Pattyn, F., Pelle, T., Price, S.F., Quiquet, A., Reese, R., Schlegel, N.-J., Shepherd, A., Simon, E., Smith, R.S., Straneo, F., Sun, S., Trusel, L.D., Van Breedam, J., van de Wal, R.S.W., Winkelmann, R., Zhao, C., Zhang, T., Zwinger, T., 2020. ISMIP6 Antarctica: a multi-model ensemble of the Antarctic ice sheet evolution over the 21st century. Cryosph. 14, 3033–3070. doi:10.5194/tc-14-3033-2020

We compare our results to ISMIP6 on lines 681-682 and 742-746 and will expand this text.

705-708: This is a very speculative statement. The water in warm cavities can certainly get warmer, as it is modified CDW and not unadulterated CDW. Please remove or rephrase this statement with supporting information.

We will rephrase this

Section 4.3: A major limitation not mentioned is the lack of iceberg calving and dynamic calving front position. Other missing physical processes that might be important are subglacial hydrology/basal physics and the impact on ice rheology of fractures and damage.

We will describe these limitations and our plan to study them in future work.

Section 5: The conclusion would benefit from an additional couple sentences about the technical achievements and limitations of the model.

We will edit the conclusion sections to cover the suggestions of both reviewers.

---

## Author Response (AR1)

We are grateful to the reviewers for their careful review of our manuscript, which raise several important points. In response to these comments we have substantially revised the paper. The changes include:

- Many figures (Figs.1-10 and 14-15 of the old version) have been modified to meet the reviewers' suggestions.

- Four new figures are added into the revised manuscript :

 *The first two new figures become Figure 1 and Figure 2. As a result, Figures 1-15 in the old version now become Figures 3-17 in the revised version.

 *The other two new figures are from observations which are used as comparison. Adding them as new panels in Figs. 6 and 13 (with the same panel size) will force the figures to grow beyond a full page. For this reason, we add these two new figures into a new Appendix A (as Figs. A1 and A2)

- Section 2 has been substantially re-written, including significant additions and editing in Section 2.1 and a major re-organisation of Section 2.3 into four new sub-sections, with additional explanations about the initialisation.

-  Many new details and editing in various parts of the revised manuscript.

Below, we address each of the reviewers' comments.

**Responses to Reviewer #1**

We thank reviewer #1 for their careful and thorough review of the manuscript. Below we copy the referee comments in black and write our responses in blue.

General comments

This paper is a major step forward for coupled ice sheet–climate modeling. It presents results from the first simulations using a complex Earth system model with full two-way coupling of ice sheets to the atmosphere and ocean, for both the Greenland and Antarctic ice sheets (though Antarctica is the focus here). Many earlier studies have argued for the importance of coupling and speculated on what might happen when feedbacks are included. Here, these speculations are put to the test, in ensemble simulations to 2100 for both low-emission and high-emission scenarios. The authors explain why their study is novel, while giving due credit to previous work. The paper is well structured and clearly written, with figures and tables that effectively illustrate the main findings.

The results are both plausible and interesting. For me, the most important findings are (1) the intrusion of warm water into the Ross and Filchner Ice Shelf cavities by the end of this century under a high-emission scenario, with consistent timing across ensemble members; (2) the absence of a strong response in the Amundsen Sea region, where warm water is already present in cavities; (3) the fact that increased snowfall (a near-term response to warming) adds more mass above flotation than the ice sheet dynamic response can remove by 2100, but with the likelihood that the dynamic response would accelerate in the 22nd century. There are many uncertainties – related, for example, to the coarse ocean resolution and the challenges of ice-sheet spin-up. The authors acknowledge these uncertainties and are careful (except for a few minor cases noted below) not to draw conclusions that go beyond the data.

I have some suggestions to sharpen the text and to guide readers who may be unfamiliar with some of the details, but no major criticisms.

We are grateful to reviewer #1 for recognising that while our study has some limitations, it is the first of its kind, and offers several new scientific advances that have not previously been possible. We are grateful to the reviewer for pointing out a few areas where we can better caveat our results, given the acknowledged limitations of this ambitious work.

Specific comments

p. 5, l. 127: Here or below, it would be useful to say more about how NEMO computes basal melt rates. It would be good to know, for instance, whether the melt rate is a strictly linear function of the thermal forcing, or if it also depends on the speed of the sub-shelf ocean current.

NEMO computes the melt rates based on the approach described in paper cited (Mathiot et al., 2017) but we understand that readers may not be aware of this, so we have added details of the 3-equation formulation in Section 2.1 (Ln 131-148).

p. 6, l. 162: "ice sheet model projections are typically initialized without any spinup". I would say "often" instead of "typically", because many ice sheet models (roughly half the models in the ISMIP6 projections) are spun up in some way.

We agree and we have updated this (Ln 207).

p. 7, l. 211: Were basal melt rates assumed to be zero in the Cornford (2016) initialization? If so, please state this, since the tuned ice-shelf viscosity could be compensating for missing basal melt.

In the Cornford et al. (2016) initialisation, for a given ice sheet geometry (from observations), the model flow equations are inverted to find fields of basal friction and stiffening coefficients that minimise the difference between model velocities and those derived from satellite observations. The inversion only uses the stress balance equation and keeps the ice sheet geometry fixed, hence basal melting and accumulation play no explicit role in it. We have now explained this in the new section 2.3.1.

p. 8, l. 221: What is the magnitude of the steady value where the rms thickness rate settles? In what regions is the remaining drift largest? Did you do multi-century standalone ice sheet runs, continuing with the same forcing? Such runs would increase confidence that the drift is small enough to maintain stable grounding lines.

The approximately steady ice sheet-average RMS dh/dt values for the four simulations are 0.4, 0.35, 0.38, and 0.34 m/y, after 17, 28, 29, and 30 years respectively, and the remaining drift is largest in isolated cells near the coast, and not in regions of dynamically evolving glaciers feeding ice shelves. We have now added some description of these points to the paper (Ln 333-338). If we continued these standalone simulations, the grounding lines would continue to retreat because ocean melting is out of balance with the ice sheet in the present day. In any case, we believe that our experimental design strongly mitigates any concerns over model drift. Throughout the paper we compare SSP1 to SSP5 projections. Therefore SSP1 serves as a 'control'. To leading order, the effects of model drifts will cancel between SSP1 and SSP5, thus revealing the influences of anthropogenic forcing, which is our main interest.

p. 9, Fig. 1: In general, the figures in the paper are informative and easy to interpret. However, many figures (including this one) use a rainbow colour scale that could be problematic for colour-blind readers. Please consider a different scale.

The figure colour scales have been revised in the new version of the manuscript, with the rainbow colormap replaced by a red-blue colormap.

p. 11, Fig. 3: In Figs. 3a and 3b, there is good agreement with observations in the thinning of Pine Island and Thwaites Glaciers and thickening of the Kamb Ice Stream. My understanding is

that this is largely the result of tuning basal coefficients to match observed ice speeds in the BISICLES spin-up. We would expect the tuned velocities to change the thickness in places where ice flow has recently accelerated (for PIG and Thwaites) or decelerated (for Kamb). This can be inferred from the text, but could be made more explicit for readers who are unfamiliar with tuning strategies.

We have now added some description of these points (Ln 403-410) in Section 3.1 to make this clearer for readers.

Another notable feature of Fig. 3 is the general slowing and thinning of ice shelves. The text (p. 12, l. 298) attributes the thinning to the SMB and basal melt forcing from the climate model. Is basal melt primarily responsible for shelf thinning, or does SMB also play an important role?

The thinning is caused by discrepancies in basal and surface forcing between the Cornford et al. (2016) and Martin et al. (2019) initialisation and our relaxation process. Such discrepancies are inevitable without a formal coupled ice and climate initialisation procedure (which is an important but very challenging avenue for future work). There are regions where SMB discrepancy is larger than the basal melt discrepancy, but the basal melt difference is larger around the grounding lines. We have expanded a paragraph in Section 3.1 to describe the slowing that occurs (Ln 416-429) – see next comment.

p. 12, l. 297: Why would this slowing mostly occur during the first year of the standalone ice sheet initialization stage? I would expect it to occur more gradually as the shelf thins.

The slowing is caused by the change of basal melting around the grounding lines when we first use melt rates from an ocean model rather than the melt rates that are implicit in the initialised ice fields. As described in a comment above, the ice model initialisation yields an initial ice velocity field for a specified ice thickness field. Combined, these two fields yield a mass flux divergence, and this implies an 'implicit' melt rate, which arises from the initialisation procedure. Upon coupling, the ocean model provides a different melt rate to this 'implicit' rate. The ocean model cannot accurately represent the very thinnest cavities near the grounding line, and so no melting occurs there in the modelled fields. Therefore, the ice generally thickens near the grounding line upon coupling, leading to small grounding line advances. This is one of the reasons why our ice relaxation period is important. The increase in drag from these re-grounded areas is instantaneously transmitted through the ice shelves, so this causes the ice shelves to rapidly decelerate. The expanded paragraph in Section 3.1 (as mentioned in the above point) now describes this issue.

p. 12, l. 302: You refer to "the SMB and basal melting implicit in the inverted reference velocities". I am not sure what this means. My understanding is that the SMB and basal melting in the BISICLES spin-up are part of the input forcing, with SMB based on reanalysis and basal melting (possibly?) set to zero. In that case, the shock could be attributed to the fact that the SMB and basal melt rates in the adjustment process (derived from the UKESM historical run and standalone ocean spin-up) are different from the SMB and basal melt rates in the spin-up. It would be helpful to describe or plot the differences.

As described above, during the initialisation the ice geometry is fixed and the basal drag and ice viscosity are altered until the ice velocities are optimally fit to observations. This means that ice thickness and velocity are initialised, which yields an ice flux divergence in each ice column, and for a given SMB this implies a basal melt rate pattern, by conservation of mass. This basal melt pattern is not zero. A description of the inversion procedure has now been added to a new section (2.3.1). It is correct however that the shock occurs because the SMB and implicit basal melt rates in this inversion process differ from those used in the adjustment process, which are derived from the UKESM historical run and standalone ocean spin-up.

p. 15, Sect 3.2.1: The abrupt transition to warm water for the Filchner Ice Shelf is a very interesting result in an ESM, consistent with the recent regional studies.

We agree that this is fascinating. This has been previously found in ocean-only and ocean/ice models, but our study is the first to show this in a coupled climate model.

p. 16, Fig. 6: Since the transition occurs near 2100, I suggest extending the x-axis to 2115, if possible.

The x-axis has been extended to 2115.

p. 15, Sect 3.2.2: The transition to a warm Ross Ice Shelf cavity is another very interesting result, notwithstanding the fresh bias in the Ross Sea. Some other studies also show freshening in this region in the beginning or near mid the 21st century.

We feel that this is a key result of our study and we believe the Ross Sea shelf warming is a new result. In Ln 851-852, we have cited Jacobs et al. (2022) whose observations indicate that the Ross Sea has been freshening for the last 60 years, as well as Timmermann and Hellmer (2013) which suggests a future freshening in this region. However, neither of those studies discuss the warming.

p. 29, l. 629: "These results may indicate the bigger potential that Ross/Weddell sectors have in becoming major sea level contributors in future warming scenarios." Here, "bigger" seems to mean "bigger than PIG and Thwaites". It's true that the Ross and Weddell sectors have the potential to become major sea level contributors, but it's also true (based on present-day observations and published simulations) that PIG and especially Thwaites could be major contributors. The Amundsen Sea contribution might not be captured by the model, for the reasons discussed in Section 4.3. So I suggest rewording this claim.

We are grateful to the reviewer for bringing this to our attention and have now re-worded (Ln 784) the claim, to say that the Ross and Weddell sectors have a 'large' potential, rather than 'bigger'.

p. 29, ll. 642ff: This paragraph is a good summary of the novelty and importance of the study.

We thank the reviewer for their supportive comments.

p. 30, l. 666. The SROCC is cited several times, but I couldn't find a citation of AR6. Please add AR6 citations where appropriate. For context, I suggest including the projected GMSL from Antarctica under low and high forcing scenarios, according to AR6.

We have updated the text to cite AR6 (Ln 778,824,842).

p. 30, l. 667: It's plausible that the AIS would have a positive mass balance in this century, but it's misleading to call this a "rapid sea level fall". The snowfall contribution is better described as a modest offset (~2 cm) to a robust global trend of rising sea level (28 to 55 cm by 2100 under SSP1-19, and 63 to 102 cm by 2100 under SSP5-85, according to AR6).

We have now expanded the text in Section 4.2 (Ln 821-823) to say that while the AIS contribution to sea level will be significantly negative, this would comprise a modest offset to the overall expected increase. This remains a very important conclusion, however, since the current paradigm is that the AIS could contribute a rapid rise in sea level to 2100.

p. 31, l. 701: "… do not retreat." Doesn't Fig. 15 show some GL retreat for PIG and Thwaites?

We have now reworded the sentence (Ln 751) to clarify that we meant that the GL in SSP5 does not retreat far relative to that in SSP1.

p. 31, l. 705: "Nevertheless, the impact of a future strong climate change in Amundsen Sea cavities is unlikely to be larger than our modelled changes in the Ross/Weddell cavities. This is because the Amundsen continental shelf and ice shelf cavities are already filled with the warm Circumpolar Deep Water and hence there is less potential for further warming and strong ice response." I think this is a bit too strong. It may be true that the ASE cavities have less potential for further warming, but this does not imply less potential for strong ice response. Because of its bed geometry, Thwaites might already be retreating unstably, or might be near a threshold such that it could be tipped into unstable retreat with a small amount of additional warming.

We thank the reviewer for these comments. We have removed this.

p. 32, l. 715: "do not appear to simulate". I suggest "do not simulate".

We have changed this (Ln 877).

p. 33, Section 5: Many conclusions already appear in the Discussion section. Since some readers will look at only the Abstract and Conclusions, I suggest adding some content in Section 5. For example:

You say here that these are the first AOGCM runs with full two-way ice-climate coupling; you could add a sentence or two (as in Section 4.1) about why this is important.

You could point out that the Filchner warming is consistent with previous modeling studies, whereas the Ross warming is something new.

You could mention the ASE non-response, with appropriate caveats about uncertainty.

I would not end the paper with a sentence that refers to the Ross Ice Shelf alone.

We thank the reviewer for the suggestions. We have included these points in the Conclusion.

Technical corrections

We thank the reviewer for their careful reading and have now addressed all of the suggested changes below in various sections of the revised manuscript.

p. 1, l. 21: "of the 21st century"

p. 5, l. 137: The phrasing is awkward, with two uses of "along with"

p. 7, l. 189: "integration" without the "s"

p. 7, l. 210: "caving" -> "calving"

p. 13, Fig. 4 caption: "The white boxes"

p. 14, l. 330: "where the SSP1-EM melt rates become"?

p. 14, l. 334: "large cold" -> "large, cold"

p. 16, l. 358: "on the ice front" -> "at the ice front"

p. 16, l. 360: "The shelf" -> "Water on the continental shelf" or something similar. In general, please be specific where the two meanings of "shelf" could be confused.

p. 16, l. 361: "becomes" -> "is" (since the deep water is not becoming denser in an absolute sense, but only relative to the shelf)

p. 16, Fig. 6i: In this panel the x-axis is different from the other panels.

p. 18, Fig. 8a: The dashed lines in this panel are hard to see in the ice shelf and continental shelf regions.

p. 18, l. 415: Add comma after "simulation"

p. 20, l. 441: Delete "is" before "already"

p. 23, l. 501, Fig. 11 caption: Typo in "SSP5-EM"

p. 24, l. 516: "on Queen Maud Land" -> "in Queen Maud Land"

p. 25, l. 539, Fig. 13 caption: It's not accurate to describe the right-hand panels as "changes" like the left and middle panels. Please reword, e.g. using "differences".

p. 26, l. 576: "end of the 2060s"

p. 27, Fig. 14 caption: The descriptions of the middle and bottom rows are reversed. In the fourth line, delete "the" before "West Antarctica". In the last line, "column" -> "columns".

p. 27, l. 588: "area-integrated" with a hyphen; delete "area" after "grounded ice sheet"

p. 27, l. 591: "from the 2040s"

p. 28, l. 598: Change "retreat up to 40 km takes place under southern Thwaites Glacier" to something like "the grounding line of Thwaites Glacier retreats southward by up to 40 km" (since the southern part of Thwaites Glacier lies far in the interior). Similarly for PIG.

p. 28, Fig. 15: It took me a few moments to get my bearings for the left and middle panels, which are rotated with respect to the standard view in a polar stereographic projection (e.g., Figs. 11-13). Maybe rotate back to the standard view. On the left panel, perhaps add labels pointing to the Thwaites and PIG ice shelves.

p. 29, l. 639: Instead of "a while", maybe "longer.

p. 29, l. 647: "modern-day" with a hyphen, or just "modern". Similarly, "present-day" at l. 652.

p. 31, l. 696: "end of the 21st century"

p. 32, l. 726: I can't tell if there is a paragraph break after "century". If not, please add one.

p. 33, l. 757: "brings" -> "bring"

p. 33, l. 759: Add a period.

References: Please check for consistent capitalization in paper titles.

Citation: https://doi.org/10.5194/tc-2021-371-RC1

**Responses to Reviewer #2**

We thank reviewer #2 for their careful and thorough review of the manuscript. We first give a general response to their main comments and major concerns. Below we copy the referee comments in black and write our responses in blue.

The manuscript "The Antarctic contribution to 21st century sea-level rise predicted by the UK Earth System Model with an interactive ice sheet" by Siahaan et al. presents perhaps the first results of a climate model coupled to an ice-sheet model of the Antarctic Ice Sheet. The paper presents a protocol for offline data coupling of UKESM and BISICLES and explains the initialization process for this complex configuration. After this, results are presented for small ensembles of coupled simulations following both SSP1-1.9 and SSP5-8.5 greenhouse gas emission scenarios. SSP1-1.9 shows small changes from initial conditions, while SSP5-8.5 leads to ice-shelf basal melt regime change beneath Ross and Filchner-Ronne ice shelves, and associated ice-shelf thinning and acceleration. The manuscript discusses the major terms of the ice-sheet mass balance and considers unique aspects and limitations of the modeling approach presented.

The manuscripts presents a significant achievement of running coupled climate and ice-sheet models for Antarctica. This has been a community goal for many years, and these results are the first to achieve it that I am aware of, even given the "offline" coupling method employed. On the other hand, the methodology for achieving it includes a number of questionable choices with an unclear impact on the resulting simulation. Most significantly, the initialization procedure for both the climate model and ice the ice-sheet model appears a bit ad hoc, with a number of details of the protocol not clearly described. It appears there may be significant

artifacts and model drift associated with the ice-sheet model initialization that have not been quantified. The authors do a fair job of noting shortcomings, but more work could be done in quantifying the impacts of those choices.

On the whole, this is an impressive modeling achievement, but the scientific utility of the results is questionable without further substantiation. The authors themselves note these limitations, and I wonder if this paper would not be better suited for a journal like GMD. Below I focus on two major areas that require more work - description of the coupling and description and analysis of the initialization procedure. I then list a number of smaller issues that require addressing. Even after significant additional analysis, it may be that the initialization and coupling procedure leave the scientific intepretation of the results ambiguous. The authors may wish to consider if a model description journal would be a more appropriate venue for this work, where it would be a significant contribution.

We are grateful to the reviewer for their careful and thoughtful review, and we agree with most of the points raised. We are very pleased that the reviewer recognises this work as a significant achievement, and a major 'first' in our field. However we also fully accept that this ground-breaking work has highlighted a few areas that will require substantial future development, most notably in the initialisation. In order to accomplish this first study, we made a series of choices that we would like to examine more fully in future. However, we feel that our experimental design mitigates these issues, so that the present manuscript is a valuable scientific contribution, with conclusions that hold despite the caveats. As a result, it will be of interest to an audience outside of the model development context. We detail our main arguments below, responding to the main points raised in the reviewer's introductory text. These arguments are fully expanded in the responses to the reviewer's specific points.

**Results are sensitive to initialisation and coupling choices:** There is certainly much work to do in understanding the best way to couple ice sheet and climate models, and how to initialise such a coupled system. However, our guiding philosophy throughout this work has been that the importance of these choices is strongly mitigated by our experimental design. Throughout the paper we qualitatively compare SSP1 and SSP5 projections. Our conclusions are based on this difference alone, with SSP1 effectively acting as a 'control' simulation. Since the impact of initialisation shocks, model drifts, coupling choices, etc. will be expressed in both scenarios, to leading order the differences between SSP1 and SSP5 are independent of these features. On reflection, this overarching philosophy was not expressed clearly enough in the submitted manuscript, and was not repeated in areas where it is relevant, so we have substantially rewritten some sections of the paper to clarify this throughout.

**Initialisation procedure is ad-hoc:** Our ambitious coupled modelling work highlights the important and very challenging area of coupled ice–climate model initialisation. There is a fundamental difficulty that climate models are spun up for millennia using pre-industrial radiative forcing, while the best predictive ice sheet models are initialised in the present day using inversion techniques. Any ice–climate coupling will therefore introduce some kind of discontinuity into the simulations when the two are combined, and minimising the impact of this is a challenging task that will require a dedicated research programme. In the present study we have devised an initial strategy that works, involving defensible choices, and with acceptable impacts. Since our experimental design mitigates these impacts even further, we believe that this issue should not delay publication of the notable scientific advances that we have made with this coupled model. To address the reviewer's comments in this area, we have expanded the discussion of the challenges of coupled initialisation and considerably enhanced the descriptions of our initialisation setup.

**Initialisation and coupling procedure are not fully described:** The coupling procedure is fully described in Smith et al. (2021) and so we seek to avoid repetition by only outlining the

main features here. We have expanded this description where recommended by the reviewer. The ice initialisation procedure is complex and not fully described elsewhere, so we have expanded that discussion significantly, as described above.

**Work may be more suitable to a model description journal:** We considered this point at length, both before the original submission and again in response to the reviews. While this manuscript represents a substantial modelling advance, much of the model development is already described in a technical paper (Smith et al., 2021) and the conclusions of the present study are scientific, not technical, in nature. In particular, our conclusions on warming of the Ross and Weddell seas, and the importance of surface mass balance in creating a negative Antarctic contribution to sea-level at 2100, are scientific advances that will be of wide importance and impact. In the revised paper we more clearly state our intention to analyse scientific hypotheses.

MAJOR CONCERNS

**1. Better description of coupling.**

I recognize that the coupling methodology was previously described in the Smith et al. (2021) JAMES paper, but given the importance of the coupling to the science results, inclusion of more information here is warranted. Some specific suggestions are:

142-6: The description of the coupling protocol is vague. Given this is a significant novelty of this work and coupled model results can be very sensitive to the coupling procedure, it should be described in much more detail. Please provide evidence that the results are not sensitive to the chosen coupling interval.

The coupling protocol is described at length in Smith et al. (2021), but we have now expanded the summary of these details in Section 2.1.

Early in the development of the coupled model we examined the sensitivity to coupling interval and satisfied ourselves that 1 year was adequate for the model resolutions characteristic of the global models we are using. Unfortunately those results are no longer available and it would require substantial development of the coupling suite to re-run these tests with the present model. However the 1-year coupling step is not controversial for these kind of experiments, driven by slow centennial evolution of the climate forcing. Favier et al., 2019 (Geosci. Model Dev., 12, 2255–2283, 2019) indicates very little sensitivity to coupling periods between 1 month and 1 year, while Zhao et al., in review (Geosci. Model Dev. Discuss. [preprint], https://doi.org/10.5194/gmd-2022-21, in review, 2022) indicates very little sensitivity to the coupling time interval between 0.5 days and 3 months. We have modified the paper and discussed this point (Ln 183-187) and cited the papers.

130: This sentence is unclear - please clarify or expand on this. Also, please add a description of how retreat of the ice-sheet grounding line is handled by the ocean model.

This is a complex area as the depth-integrated ocean transport must be preserved as the ice geometry changes, in order to avoid creating 'tsunamis' in the ocean free surface. A retreat of the grounding line is actually the simplest case, since the new ocean column that is created can simply be given zero ocean transport. An advance of the grounding line is more complex, as some ocean transport then 'disappears'. The full details are given by Smith et al. (2021), but we have modified the paper to include this (Ln 150-164) as suggested.

126-146: Please acknowledge in this section that the calving front restriction and the bilinear remapping prevent the system from conserving mass and heat, though these errors are not likely to be significant for the experiments being conducted.

We agree with this and have added this acknowledgement (Ln 169-171).

**2. Model initialization and its impacts.**

Initialization of coupled climate and ice-sheet models is a very challenging problem. Still, the procedure described here appears ad hoc, missing key information, and does not include an assessment of key initialization choices on the results. The specific point below detail these concerns:

This is a very complex area and we agree its explanation could be clearer. In response to the comments below we have substantially re-written section 2.3. After some consideration we think we can improve the clarity of this section by clearly labelling the different stages of the simulations, as follows:

A. Standalone ice model inversion (Cornford et al, 2015)

B. Generate ocean cavity properties (Single simulation, 45 years)

C. Add ocean cavity properties to global UKESM ocean and relax (4 simulations, 15 years)

D. Relax ice model under UKESM ocean cavity and surface mass balance forcing (4 simulations, 17-30 years)

We split the section into 4 sub-sections (2.3.1 - 2.3.4) which describe each of these stages, and we added a schematic diagram (the new Fig. 1) illustrating them.

179: This paragraph is confusing and is missing key information.

The issue under consideration here is that we have the CMIP6 UKESM simulations for historical periods but those have a static ice sheet, and no cavities beneath floating ice. They simply have a coastline at the ice front. Therefore we need to generate initial ocean properties and ice shelf melt rates within the sub-ice cavities from a short run of the coupled model. We have expanded and clarified this paragraph in Section 2.3.2.

182: 45 years seems like a short ocean spin-up time. Can you provide justification that the most important transient behavior had reduced to small levels prior to creating the branch runs?

In retrospect, the original manuscript was not sufficiently clear on the purpose of this initial 45-year simulation (stage B above). Its only goal here is to produce some preliminary ocean properties in the ice shelf cavities, and 45 years is more than enough to accomplish this. In stage C, these cavity properties are then joined on to global ocean states from UKESM historical runs, and that is run forward for 15 years in order to flush the cavities with ocean properties from the global UKESM. Thus the convergence of stage C is important to the simulations, but the convergence of stage B is not. This is now described in the prevised aper.

179-194: After reading the full paragraph, I am more confused. Replace the opening paragraph by clearly stating the initialization process is based on a hybrid state of the ice

cavities from UKESM1.0-ice stitched onto the global ocean state of UKESM1.0. Also, a schematic of the various runs and processing, would help communicate this process.

This is a complex area and we have added some sentences (Ln 257-261) in Section 2.3.2 to state this and a schematic in the new Fig. 1.

186: Emphasize for the reader that UKESM1.0 does not include ice shelves, e.g. add a parenthetical "(without ice shelves)" after "UKESM1.0".

We have now included a sentence in the beginning of Section 2.1, reminding the reader that the suffix 'ice' in UKESM1.0-ice is added to refer to models with ice shelf cavities and an active ice sheet.

187: Which years were chosen? Can you elaborate on what "a range of variability" means? How many ensemble members were in the UKESM1.0 CMIP6 historical ensemble? Also, do you have a reference for that ensemble, or a reference to the dataset on ESGF? What does "end of the 20th century" mean? What specific years were used?

The UKESM1.0 CMIP6 historical ensemble contains 19 members, and is most usefully cited via Yool et al, "Evaluating the physical and biogeochemical state of the global ocean component of UKESM1 in CMIP6 historical simulations" GMD 14, (2021). Three of the initial states were taken directly from this ensemble, all from the year 2000 in their respective members. They were chosen to maximise the range of ACC strength and Southern Ocean annual average SST across the available UKESM ensemble members in that year.

The global ocean state for our fourth member was taken from the end of the preliminary UKESM1.0-ice simulation (Stage B). Since Stage B had not been run long enough to drift far from the 1995-2014 average EN4 climatology it was initialised with, it provides an initial ocean state more representative of the observed modern ocean than the UKESM1.0 historical ensemble. We have now added clarification of the source of our initial states in Stage C of the initialisation (Ln 299-312).

189: What does "short" mean? And what does "balance" mean on line 190? Please show evidence of balance or behavior that is close to steady state, either in the form of a plot or some statistics.

We replaced 'short' with '15-year' (Ln 289). We have also added a time-series plot of temperature difference across the ice front in the new Fig. 2 which shows that its seasonal oscillation is steady after the first year of the spin-up.

196: Please explain why the ocean-sea ice simulations are regarded as beginning in 2000 if they were branched from specific times of UKESM and included a cavity state from perpetual 1970.

We expanded the discussion of Stage C of the revised paper. As noted above in the comment about line 187, the global ocean-sea ice states were taken either from the year 2000 in one of the members in the UKESM1 historical ensemble, or represent the modern observed ocean (averaged 1995-2014). The ice-shelf cavity information does come from perpetual 1970 forcing UKESM1.0-ice simulations (stage B), but is merged with these global ocean states and run for 15 years with UKESM surface forcing from 2000 to 2015 (stage C). This flushes the cavities and bring them in line with the global ocean, and produces an ocean state representative of year 2015.

197: Are the atmospheric fluxes from the same runs that each was branched from? Or one common set of atmospheric fluxes?

We have added a note (Ln 305-306) that the atmospheric fluxes for the the first three members were taken from the same ensemble member as the initial global ocean state. We took the surface forcing of one of those three to be used as the surface forcing of the fourth member (Ln 309).

195-202: So the final initialization step is 1) not fully coupled to the atmosphere and 2) has temperature and salinity restoring applied. Given that, it is unlikely that you can branch into a fully coupled projection without some shock and drift. Please justify this choice.

We have expanded the discussion at this point (also in Section 2.3.2). Stage C of the initialisation procedure is intended to produce ice-shelf cavity ocean properties that are consistent with the UKESM1.0 ocean state in the wider ocean. There is no perfect way to do this, but we believe our strategy is the best available. We constrain ocean surface fluxes to match UKESM1.0, and restore ocean properties outside the ice-shelf cavities to match UKESM1.0. Thus, we believe that in 2015, after 15 years of this procedure, we have ice-shelf cavity ocean properties that are the best match possible to the wider UKESM1.0 global ocean states. Thus, when we subsequently run this forward as a coupled model, we do not expect large ocean-state shocks to occur. The 'perfect' solution to this problem would be to have a historical climate model simulation with ice-shelf cavities, but these are not currently available. This will be our strategy for the next generation of the UKESM. As described above, our experimental design of comparing SSP1 and SSP5 scenarios strongly mitigates the influence of any coupling shock that remains.

195-202: Throughout this section, please clarify if every time you refer to UKESM1.0-ice, prognostic ice-shelf basal melt fluxes are on and what is happening with iceberg fluxes (if anything).

Both ice shelf basal melt rate and iceberg calving, drift, and melting are always prognostic in UKESM1.0-ice. We have described these in Section 2 (Ln 128-148,165-169,177-183).

214: What is the thickness output of the inversion procedure? In the previous sentence it is only said that basal drag coefficients and viscosity are adjusted.

The ice thickness does not change in the inversion procedure. This is now clarified in the new Section 2.3.1.

216: Ice-shelf melting from what year(s) of each standalone run?

In order to test the influence of interannual variability, two members use 2014 melting whereas two others use the 2010-2014 average, in common with the SMB forcing. We have now edited the sentence (Ln 331-333) to imply that both SMB and melting are averaged across the same period (2014 for the first two members and 2010-2014 for the other two).

221: Referring to this 'spike' is vague, as is "about 20 years". Please report statistics or, better, show a time series of this RMS metric.

The new section 2.3.3 has been edited to discuss this (Ln 334-338). The approximately steady ice sheet-average RMS dh/dt for the four simulations are 0.4, 0.35, 0.38, and 0.34 m/yr, after 17, 28, 29, and 30 years respectively, and the remaining drift is largest in isolated cells near the coast, and not in regions of dynamically evolving glaciers feeding ice shelves. Ice

simulations in stage D were motivated by a wish to remove any immediate coupling shock on introducing UKESM surface and basal mass balance fields to the ice sheet model and ensure numerical stability of the initial steps of the interactive ice-climate system in the SSP simulations. Stage D simulations were about 30 years long. The ice states taken forward to the projection simulations were chosen to have a continental average RMS dh/dt of less than 0.5 m/yr and a maximum magnitude of dh/dt in any location less than 75 m/yr. The state taken at year 17 was taken earlier than for the others to avoid the effect of a significant transient spike in both measures of dh/dt that occurs near the end of that simulation.

204-223: It is entirely clear what year the ice sheet initial condition represents. In line 212, it is stated the ice-sheet state starting the procedure represents "early 21st century", but then it is relaxed for "about 20 years". How does the final state of each ensemble member compare to recent observed thinning rates, which have been highly variable in time?

The ice sheet inversion (stage A) represents the ice thickness from the Bedmap2 dataset (Fretwell et al., 2013) and the ice velocity from Rignot et al., (2011) dataset. These data are from a range of dates and so the inverted ice state does not have a clear uniform time-stamp other than 'early 21st century'. In stage D, the ice sheet is then evolved for a further 17-30 years so that this ice sheet state can adjust to the four different realisations of 2014 or 2010-2015 basal melt forcing produced by the UKESM. So the ice model state upon coupling does not have a clear time stamp, other than it is 'early 21st century'. The details about the ice sheet inversion are given in the new section 2.3.1.

Figure 3 and associated ext (286-292): The very large amount of noise (presumably transient flux divergence) in the ice-sheet elevation/thickness change deserves a few sentences of explanation. While this is a well known and common challenge for ice-sheet models, the amount of spurious thickness change after 15 years of integration and 20 years of relaxation (if I followed the protocol correctly - it was confusing - see above) seems unacceptably large. Also, what is the purpose of panel d? It shows nearly the same information as panel b.

We hope that the protocol is clearer in the revised paper. We removed the old Figure 3d (ice thickness difference) and added SSP5-SSP1 dh/dt in panel c (now Fig. 5c). We do not feel that the noise in Figure 3b (now Figure 5b) is unacceptably large - it is of order 0.1 m/y. Crucially, as shown in the new panel 5c (and also Figure 15) and discussed further below, this 'noise' exists in both the SSP1 and SSP5 projections, and is cancelled in their difference. Therefore, despite the 'noise', we are able to determine the influence of climatic forcing.

297: It is not obvious to me why ice shelves would slow so significantly in the first year just due to one year of thickness change coming from surface and basal mass balance, especially if further adjustment after one year is small. Is it possible grounding line positions have shifted or the ice temperature field changed or something else is going on? A more thorough explanation is warranted.

We have now added more explanation in Section 3.1 (Ln 416-429). The slowing is caused by the change of basal melt forcing around the grounding lines when we first use melt rates from an ocean model rather than the 'implicit' melt rates arising from the ice initialisation (stage A). The ocean model cannot accurately represent the very thinnest cavities near the grounding line, and so no melting occurs there in the modelled fields. Therefore, the ice generally thickens near the grounding line upon coupling, leading to small grounding line advances. The increase in drag from these re-grounded areas is instantaneously transmitted through the ice shelves, so this causes the ice shelves to rapidly decelerate.

Section 3.4: It would be easier to interpret the changes presented if there was also a standalone BISICLES control run that had constant 2015 forcing. Presumably the speckly pattern of thinning and thickening in Fig. 13 panels a and b is due to unrealistic transient

behavior in the initial condition. That is a common problem in ice-sheet models, so it does not necessarily invalidate the results, but it should be clearly identified. I would prefer to see additional results for an unforced control run. Without it, it is difficult to assess what aspect of these results are an effect of the ice-sheet model initialization procedure and what is due to the climate forcing coming from UKESM.

We have edited the relevant text to explain that the pattern arises from the initial ice state (Ln 699-701). We feel that this 'noise' is not large relative to the dynamic thinning signals of interest and, crucially, it does not appear when we difference SSP1 and SSP5 in the old Figure 13 (Figure 15 in the revised version). This is also pointed out in the current Fig. 5c where the noise is the same between SSP1 and SSP5 scenarios and associated with the initial state. This means that the climatic signals that we are focussing on are not influenced. We considered the possibility of conducting a standalone ice control run but this is problematic for two reasons: i) the ice would still evolve because ocean melting is out of balance with the ice sheet in the present day and ii) it is not clear what melt rates we would use when the grounding lines retreated.

**Other Comments:**

Abstract: Mentioning Greenland in the abstract is slightly misleading, because the Greenland results are not part of this paper.

We have now clarified in Ln 19-20 that while Greenland is coupled into the model, we only analyse Antarctica here.

47: Also cite the only paper that demonstrates this for the observational period that this sentence discusses:

Gudmundsson, G.H., Paolo, F.S., Adusumilli, S., Fricker, H.A., 2019. Instantaneous Antarctic ice sheet mass loss driven by thinning ice shelves. Geophys. Res. Lett. 46, 13903–13909. doi:10.1029/2019GL085027

We have now updated the references and cited it in Ln 47.

48-55: There are a lot of other important studies that would be appropriate to reference here, e.g.:

Spence, P., Holmes, R.M., Hogg, A.M., Griffies, S.M., Stewart, K.D., England, M.H., 2017. Localized rapid warming of West Antarctic subsurface waters by remote winds. Nat. Clim. Chang. 7, 595–603. doi:10.1038/nclimate3335

We have now updated the references and cited it in Ln 50.

65: CMIP5 and CMIP6

We have now changed this in Ln 65.

68: One CMIP-class ESM recently published (since this manuscript was submitted) Antarctic subglacial melt rates (but those simulations were not part of CMIP6):

Comeau, D., Asay-Davis, X. S., Begeman, C., Hoffman, M. J., Lin, W., Petersen, M. R., et al. (2022). The DOE E3SM v1.2 Cryosphere Configuration: Description and Simulated Antarctic

Ice-Shelf Basal Melting. Journal of Advances in Modeling Earth Systems, 14, e2021MS002468. https://doi.org/10.1029/2021MS002468

We do not include the above reference since the context in this paragraph (and also the focus of this study) is on future projection simulations of Antarctica.

69-74: There also is recently published (since this manuscript was submitted) regional model that includes all physical climate components (atmosphere, ocean, sea ice, land, ice sheet):

Pelletier, C., Fichefet, T., Goosse, H., Haubner, K., Helsen, S., Huot, P.-V., Kittel, C., Klein, F., Le clec'h, S., van Lipzig, N.P.M., Marchi, S., Massonnet, F., Mathiot, P., Moravveji, E., Moreno-Chamarro, E., Ortega, P., Pattyn, F., Souverijns, N., Van Achter, G., Vanden Broucke, S., Vanhulle, A., Verfaillie, D., Zipf, L., 2022. PARASO, a circum-Antarctic fully coupled ice-sheet–ocean–sea-ice–atmosphere–land model involving f.ETISh1.7, NEMO3.6, LIM3.6, COSMO5.0 and CLM4.5. Geosci. Model Dev. 15, 553–594. doi:10.5194/gmd-15-553-2022

We do not include the above reference since the context in this paragraph (and also the focus of this study) is on future projection simulations of Antarctica. The above reference is about present day or historical simulations.

58-81: Somewhere in here you should also acknowledge the fully coupled configuration of CESM with the Greenland Ice Sheet.

We do not include the above reference since the context in this paragraph (and also the focus of this study) is on future projection simulations of Antarctica, whereas the above reference is about Greenland Ice Sheet.

104: Can you report the approximate horizontal resolution of the 1 degree ocean grid at the typical latitude of Antarctic ice shelves?

About 17-22 km (Ln 129).

111: It is worth pointing out that this adaptivity is dynamic in time.

We have now written 'time-evolving adaptive' the first time we mention this (Ln 114).

117: Can you briefly summarize the impact of choosing 2 km as your finest resolution instead of 1 km or 0.5 km as is sometimes used for BISICLES?

This is summarised in Ln 883-885 (the discussion section : "the 2 km highest level of mesh refinement we allowed for BISICLES in these simulations may not be sufficient to accurately model the grounding line dynamics in this region (Cornford et al., 2016), although testing suggests that increasing the allowed refinement of the BISICLES to 500 m would not significantly alter our model evolution of the next few decades.")

162: I would say most ice-sheet models follow this practice, but it is not 'typically' the case - there are a number of ice-sheet models that do a paleo or steady state spinup.

We have now changed it to 'often' (Ln 207).

169-171: These comments make me wonder if this manuscript would be more appropriate for GMD or JAMES.

Though the paper describes some technical detail, the main conclusions are all scientific in nature, and so we believe this work is much better suited to The Cryosphere.

Figure 1a,b and associated text in 3.1: It is rather awkward to compare these plots to referenced observational data without showing those observations or model biases relative to them. As presented, these comparisons are not useful.

There are no complete observational data sets of the stream function or mixed-layer depth, so we just show the model results. We mention this in the text (Ln 375-376). For temperature and salinity, we compare to observations.

264: Would not surface restoring bring properties closer to observations?

This is correct so we have removed the reference to surface restoring and referred to the inadequacy of ocean-atmosphere surface flux in the standalone ocean model (Ln 388 & 392).

249-271: This discussion of water mass properties, especially at depth, based on mixed layer depth is obtuse. It would be much better to show T&S diagrams for the regions of interest compared to observations. Many global ocean models struggle with the formation of Dense Shelf Water, even with realistic mixed layer depths, so that in and of itself is not a guarantee of good water mass properties. It would also be quite important to see maps of ocean bottom temperature and salinity, as those matter more for ice-shelf basal melting than surface properties.

In Figure 3 (previously Figure 1 in the old version) we show the water properties over the depth range (300-1000 m) important to ice-shelf melting, which is the focus of this study. This shows how well the water masses on the shelf are represented. We refer to mixed layer depths merely to describe why the shelf water masses appear as they are (in the real world and in the model) - it is the presence or absence of cold, salty waters produced by sea ice formation that determines this. We have re-worded some of the sentences (Ln 381-394) in the revised paper.

271-2: Please provide evidence for this statement (e.g. the bias value for UKESM and other CMIP models). Are you basing this statement off of the version of UKESM in Heuze (2020) or the simulations presented here?

We realise that the citation to Heuze was misleading. We meant that the temperatures were broadly accurate (current Figure 3 or Figure 1 in the old version) and this was superior to many other models (Heuze). We have changed the text accordingly in Ln 396-397.

278: Do you mean *near-shore* fresh bias here? Over most of the Southern Ocean, Fig. 1 shows a saline bias.

We are referring to the fresh bias in the Ross continental shelf and have changed the text (Ln 400-401) to reflect this.

Figure 4: Similar to figure 1, this figure should include the observational references fields (or show an anomaly). Simply saying "shows a similar spatial pattern to present day observations (Rignot et al., 2013; Adusumilli et al., 2020)" and expecting a reader to pull those up and make comparisons across different colourbars is not sufficient. Also the linear colourbar is inadequate for showing the high melt rates in the Amundsen Sea - presumably the ice shelves in that entire sector are well above 5 m/yr. Similarly, the colourbar in panel e and f saturates in areas of interest (e.g., Ross and Larsen ice shelves).

We have now included the map from Adusumilli (2020) in Fig. A1 (Appendix A). Given the low resolution of the Amundsen cavities, we are very reluctant to focus on the particular details of ice shelf melting in this region (Ln 444-445). Therefore, we prefer to keep the colour scale, because firstly it highlights the regions of high and low melting discussed in the section, and secondly it is the same colour scale used by Adusumilli at al., (2020) and Rignot et al., (2013) for their observations.

Table 1: Presumably in your model analysis you can separate Ross and FRIS into the two halves that the observational data uses.

We do not see any clear rationale for these regional splits in Ross and Filchner-Ronne, since any ice shelf could be arbitrarily divided in this way. Therefore, we prefer not to use them. The regional detail of modelled and observationally-derived melt rates is shown in the figure.

309: While the modeled melt might be within the range, I suspect a t-test would indicate a significant difference. That is not necessarily unacceptable, but please report a more careful comparison.

The errors in the observational datasets are very large and the disagreement between different datasets is also large. For this reason we do not feel that detailed statistics are appropriate. We have re-phrased the sentence (Ln 449-451) to say that the model is loosely in agreement.

Figure 5: Please also show present-day observational estimates for reference.

We have added the estimate from Rignot (2013) in Figs. 7a-d, with approximate standard deviations for Ross and Filchner-Ronne.

Section 3.2.1: This section demonstrates the melt regime change at FRIS very clearly, but the causal mechanism is left only hinted at. There is a plot and mention of declining sea ice volume and its possible relation to declining density. There also is a mention of increasing freshwater flux. This is already a long, dense paper, but if it were possible to tease out the mechanism(s) leading to WDW increase, that would be a valuable contribution. Have you looked at changes in wind stress? The previous papers you cite also discuss that as a potential mechanism for the WDW intrusion at FRIS.

Our results share many aspects with the cited papers, so we choose to rely on those for FRIS and focus on providing a fuller explanation of the novel changes in the Ross Sea. We have emphasised this in Ln 504-505.

424-6: From Figure 9, it looks like a missing piece of this explanation is that Dense Shelf Water (cold and saline) on the continental shelf is present at the start of the simulation, but becomes significantly fresher by 2060 (Fig. 9e). This is consistent with the sea-ice decline mentioned and shown in Fig. 8b to become more substantial around that time. Similar to the FRIS case, the reduction in the continental shelf density barrier facilitates the intrusion of mCDW. This series of events is alluded to in this paragraph, but the sentences at 426-7 implies that the driving mechanism is warming of the mCDW, which is not apparent in Figure 9. Maybe this just requires some rewording.

We agree with this narrative and have changed the wording in Ln 564-565.

432: As you say, I think the relative model fresh bias in each of these regions is critical. To further illustrate that point, could you follow up with a quantitative metric of the salinity bias in

each region at the start of the simulation? (e.g. averaged over the region or at the shelf break analysis point used in each region.

The intention of raising this was to point out that further work is needed to reduce the salinity biases in this model, and then re-evaluate the mechanism that is manifest in our SSP projections. As shown in Figure 3 (was Figure 1 in the old version), the salinity biases are highly spatially dependent.  Since we are not sure precisely which area to calculate such a metric over, we prefer to simply point out the spatial heterogeneity of the bias, and the need for future work on this topic (Ln 907-910).

440: You haven't shown that the regional climate is warming during this period. Maybe reword to "While the changing climate".

We have reworded this in Ln 585.

441: Remove "is".

Reworded in Ln 586.

Fig. 10: What year and simulation are these draft values from?

Year 2080, from one of the SSP5 members. We have now clarified this in the caption of Fig. 12 (Fig. 10 in the old version).

Section 3.2.3: Initially I was skeptical of even discussing results from an ice shelf represented by 11 grid cells, but I appreciate the honest assessment of what is happening in the model here, given the importance of this region. Better to acknowledge the limitations of interpreting these results than ignore it and risk readers reading their own interpretation into it.

We agree

491: Similar to previous comments, simply stating your results look similar to your observational reference is insufficient. Please include a panel in the figure showing the reference dataset (or the difference from it).

We have now included this observational reference in the new Fig. A2 (Appendix A).

Table 2: Please also include a present-day estimate (e.g. from RACMO).

Rather than selecting a single model, we have added a sentence (Ln 653-655) comparing our SMB to the intercomparison range in Mottram et al. (2021).

525: Another relevant reference here: Trusel, L.D., Frey, K.E., Das, S.B., Karnauskas, K.B., Munneke, P.K., Meijgaard, E. Van, Broeke, M.R. Van Den, 2015. Divergent trajectories of Antarctic surface melt under two twenty-first-century climate scenarios. doi:10.1038/NGEO2563

We have cited it in Ln 672.

541: The Thwaites and Pine Island inland thinning goes away when you difference the two scenarios, and there is in fact less thinning in the SSP5 scenario. Please discuss this. Having a control run for context (previous comment) would likely help here.

This is consistent with the results of the paper. The dynamic thinning in the Amundsen region is the same in both projections. We don't place much faith in this, because the ice shelves are not well resolved. SSP5 has a greater increase in SMB than SSP1, hence 'less thinning'. As described above, SSP1 is acting as a control run. We have now clarified this in the text (Ln 693-696).

Figure 14: Typo in 'cumulative' in title above panel b.

This is now reworded.

574: Consider rewording this sentence to avoid the possible interpretation that the thinning of Ross Ice Shelf has a direct impact on MAF.

This is now reworded in Ln 728.

580: This goes back to my earlier comment about what Thwaites and Pine Island are doing in the control run.

We have mentioned these limitations in Discussion section, and we have added another sentence in Ln 736-737.

Figure 15: Consider using the same colour scheme for the two scenarios here as in the previous figure.

The color scale is now the same as in the previous figure.

642. 655-8: Maybe GMD/JAMES is a better fit?

The scientific conclusions of our paper are a good fit to The Cryosphere. We are primarily interested in differences between SSP1 and SSP5.

Section 4.2: A short comparison of the results to those of ISMIP6-AIS is warranted, as that set of experiments is perhaps the closest point of reference to this work. In addition to considering the overall behavior of each region, it would be interesting to look at the threshold for surface hydrofracture employed by ISMIP6 and if/when that is passed in your simulations. Similarly, comparing your simulated basal melt rates to the parameterization they employ might help explain differences in response.

ISMIP6: Seroussi, H., Nowicki, S., Payne, A.J., Goelzer, H., Lipscomb, W.H., Abe-Ouchi, A., Agosta, C., Albrecht, T., Asay-Davis, X., Barthel, A., Calov, R., Cullather, R., Dumas, C., Galton-Fenzi, B.K., Gladstone, R., Golledge, N.R., Gregory, J.M., Greve, R., Hattermann, T., Hoffman, M.J., Humbert, A., Huybrechts, P., Jourdain, N.C., Kleiner, T., Larour, E., Leguy, G.R., Lowry, D.P., Little, C.M., Morlighem, M., Pattyn, F., Pelle, T., Price, S.F., Quiquet, A., Reese, R., Schlegel, N.-J., Shepherd, A., Simon, E., Smith, R.S., Straneo, F., Sun, S., Trusel, L.D., Van Breedam, J., van de Wal, R.S.W., Winkelmann, R., Zhao, C., Zhang, T., Zwinger, T., 2020. ISMIP6 Antarctica: a multi-model ensemble of the Antarctic ice sheet evolution over the 21st century. Cryosph. 14, 3033–3070. doi:10.5194/tc-14-3033-2020

Comparison with ISMIP6 is not straightforward since their results use a present-day control projection whereas our experimental design compares SSP1-2.6 to SSP5-8.5. However, we discuss ISMIP6 a number of times (Ln 681-682 and 742-746 of the previous manuscript version). In this new revision, we have added a few sentences in Section 4.2 (Ln 859-862) for basal melting.

705-708: This is a very speculative statement. The water in warm cavities can certainly get warmer, as it is modified CDW and not unadulterated CDW. Please remove or rephrase this statement with supporting information.

We thank the reviewer and agree about this. We have removed the statement.

Section 4.3: A major limitation not mentioned is the lack of iceberg calving and dynamic calving front position. Other missing physical processes that might be important are subglacial hydrology/basal physics and the impact on ice rheology of fractures and damage.

We have now described these limitations and our plan to study them in future work (Ln 927-928).

Section 5: The conclusion would benefit from an additional couple sentences about the technical achievements and limitations of the model.

We have added a few sentences about these in the Conclusion.

---

## Author Response (AR2)

We are grateful to the editor for the suggestions and corrections to the revised manuscript. We have edited it accordingly by correcting the typo mistakes and following the suggestions:

- The resolution of Figures 2 and 7 have been increased from 300dpi into 600 dpi.

- The global surface temperature anomaly at the start of warm water intrusion into Ross Shelf has been added into Section 3.2.2

- The previous reference to the article in review has been updated. It (Zhao et al., 2022) has just been accepted a few days ago, so we have included the DOI.